# BILEVEL OPTIMIZATION UNDER UNBOUNDED SMOOTHNESS: A NEW ALGORITHM AND CONVERGENCE ANALYSIS

**Jie Hao, Xiaochuan Gong, Mingrui Liu**[*]
Department of Computer Science, George Mason University, Fairfax, VA 22030, USA
`{jhao6, xgong2, mingruil}@gmu.edu`

## ABSTRACT

Bilevel optimization is an important formulation for many machine learning problems, such as meta-learning and hyperparameter optimization. Current bilevel optimization algorithms assume that the gradient of the upper-level function is Lipschitz (i.e., the upper-level function has a bounded smoothness parameter). However, recent studies reveal that certain neural networks such as recurrent neural networks (RNNs) and long-short-term memory networks (LSTMs) exhibit potential unbounded smoothness, rendering conventional bilevel optimization algorithms unsuitable for these neural networks. In this paper, we design a new bilevel optimization algorithm, namely BO-REP, to address this challenge. This algorithm updates the upper-level variable using normalized momentum and incorporates two novel techniques for updating the lower-level variable: *initialization refinement* and *periodic updates*. Specifically, once the upper-level variable is initialized, a subroutine is invoked to obtain a refined estimate of the corresponding optimal lower-level variable, and the lower-level variable is updated only after every specific period instead of each iteration. When the upper-level problem is nonconvex and unbounded smooth, and the lower-level problem is strongly convex, we prove that our algorithm requires $\widetilde{O}(1/\epsilon^4)$ [1] iterations to find an $\epsilon$-stationary point in the stochastic setting, where each iteration involves calling a stochastic gradient or Hessian/Jacobian-vector product oracle. Notably, this result matches the state-of-the-art complexity results under the bounded smoothness setting and without mean-squared smoothness of the stochastic gradient, up to logarithmic factors. Our proof relies on novel technical lemmas for the periodically updated lower-level variable, which are of independent interest. Our experiments on hyper-representation learning, hyperparameter optimization, and data hyper-cleaning for text classification tasks demonstrate the effectiveness of our proposed algorithm. The code is available at `https://github.com/MingruiLiu-ML-Lab/Bilevel-Optimization-under-Unbounded-Smoothness`.

## 1 INTRODUCTION

Bilevel optimization refers to an optimization problem where one problem is nested within another (Bracken & McGill, 1973; Dempe, 2002). It receives tremendous attention in various machine learning applications such as meta-learning (Franceschi et al., 2018; Rajeswaran et al., 2019), hyperparameter optimization (Franceschi et al., 2018; Feurer & Hutter, 2019), continual learning (Borsos et al., 2020), reinforcement learning (Konda & Tsitsiklis, 1999; Hong et al., 2023), and neural network architecture search (Liu et al., 2018). The bilevel problem is formulated as the following:

$$\min_{\boldsymbol{x} \in \mathbb{R}^{d_x}} \Phi(\boldsymbol{x}) := f(\boldsymbol{x}, \boldsymbol{y}^*(\boldsymbol{x})), \quad \text{s.t.,} \quad \boldsymbol{y}^*(\boldsymbol{x}) \in \arg\min_{\boldsymbol{y} \in \mathbb{R}^{d_y}} g(\boldsymbol{x}, \boldsymbol{y}), \tag{1}$$

where $f$ and $g$ are referred to as upper and lower-level functions, respectively, and are continuously differentiable. The upper-level variable $\boldsymbol{x}$ directly affects the value of the upper-level function $f$ and indirectly affects the lower-level function $g$ via $\boldsymbol{y}^*(\boldsymbol{x})$. In this paper, we assume the lower-level function $g(\boldsymbol{x}, \boldsymbol{y})$ is strongly-convex in $\boldsymbol{y}$ such that $\boldsymbol{y}^*(\boldsymbol{x})$ is uniquely defined for any $\boldsymbol{x} \in \mathbb{R}^{d_x}$ and $f(\boldsymbol{x}, \boldsymbol{y})$ is potentially nonconvex. One important application under this setting is hyper-representation learning with deep neural networks (Franceschi et al., 2018), where $\boldsymbol{x}$ denotes the shared representation

---

[*]Corresponding Author: Mingrui Liu (`mingruil@gmu.edu`).

[1]Here $\widetilde{O}(\cdot)$ compresses logarithmic factors of $1/\epsilon$ and $1/\delta$, where $\delta \in (0, 1)$ denotes the failure probability.

Table 1: Comparison of stochastic bilevel algorithms for finding an $\epsilon$-stationary point as defined in Definition 2.1. The oracle stands for calls to stochastic gradients and stochastic Hessian/Jacobian-vector products. $\mathcal{C}_L^{a,k}$ denotes $a$-times differentiability with Lipschitz $k$-th order derivatives. "SC" means "strongly-convex". We do not include results with $\widetilde{O}(\epsilon^{-3})$ complexity and with extra mean-squared smooth stochastic gradient assumption (Yang et al., 2021; Khanduri et al., 2021).

| Method | Oracle Complexity | Upper-level $f$ | Lower-level $g$ | Batch Size |
|---|---|---|---|---|
| BSA (Ghadimi & Wang, 2018) | $\widetilde{O}(\epsilon^{-6})$ | $\mathcal{C}_L^{1,1}$ | SC and $\mathcal{C}_L^{2,2}$ | $\widetilde{O}(1)$ |
| StocBio (Ji et al., 2021) | $\widetilde{O}(\epsilon^{-4})$ | $\mathcal{C}_L^{1,1}$ | SC and $\mathcal{C}_L^{2,2}$ | $\widetilde{O}(\epsilon^{-2})$ |
| AmIGO (Arbel & Mairal, 2021) | $\widetilde{O}(\epsilon^{-4})$ | $\mathcal{C}_L^{1,1}$ | SC and $\mathcal{C}_L^{2,2}$ | $O(\epsilon^{-2})$ |
| TTSA (Hong et al., 2023) | $\widetilde{O}(\epsilon^{-5})$ | $\mathcal{C}_L^{1,1}$ | SC and $\mathcal{C}_L^{2,2}$ | $\widetilde{O}(1)$ |
| ALSET (Chen et al., 2021) | $O(\epsilon^{-4})$ | $\mathcal{C}_L^{1,1}$ | SC and $\mathcal{C}_L^{2,2}$ | $O(1)$ |
| F$^2$SA (Kwon et al., 2023a) | $O(\epsilon^{-7})$ | $\mathcal{C}_L^{1,1}$ | SC and $\mathcal{C}_L^{2,2}$ | $O(1)$ |
| SOBA (Dagréou et al., 2022) | $O(\epsilon^{-4})$ | $\mathcal{C}_L^{2,2}$ | SC and $\mathcal{C}_L^{3,3}$ | $O(1)$ |
| MA-SOBA (Chen et al., 2023b) | $O(\epsilon^{-4})$ | $\mathcal{C}_L^{1,1}$ | SC and $\mathcal{C}_L^{2,2}$ | $O(1)$ |
| BO-REP (this work) | $\widetilde{O}(\epsilon^{-4})$ | $(L_{\boldsymbol{x},0}, L_{\boldsymbol{x},1}, L_{\boldsymbol{y},0}, L_{\boldsymbol{y},1})$-smooth | SC and $\mathcal{C}_L^{2,2}$ | $O(1)$ |

layers that are utilized across different tasks, and $\boldsymbol{y}$ denotes the classifier encoded in the last layer. In this paper, we consider the stochastic setting. We only have access to the noisy estimates of $f$ and $g$: $f(\boldsymbol{x},\boldsymbol{y}) = \mathbb{E}_{\zeta \sim \mathcal{D}_f}[F(\boldsymbol{x},\boldsymbol{y};\zeta)]$ and $g(\boldsymbol{x},\boldsymbol{y}) = \mathbb{E}_{\xi \sim \mathcal{D}_g}[G(\boldsymbol{x},\boldsymbol{y};\xi)]$, where $\mathcal{D}_f$ and $\mathcal{D}_g$ are underlying data distributions for $f$ and $g$ respectively.

The convergence analysis of existing bilevel algorithms needs to assume the gradient is Lipschitz (i.e., the function has bounded smoothness parameter) of the upper-level function $f$ (Ghadimi & Wang, 2018; Grazzi et al., 2020; Ji et al., 2021; Hong et al., 2023; Kwon et al., 2023a). However, such an assumption excludes an important class of neural networks such as recurrent neural networks (RNNs) (Elman, 1990), long-short-term memory networks (LSTMs) (Hochreiter & Schmidhuber, 1997) and Transformers (Vaswani et al., 2017) which are shown to have unbounded smoothness (Pascanu et al., 2012; 2013; Zhang et al., 2020b; Crawshaw et al., 2022). For example, Zhang et al. (2020b) proposed a relaxed smoothness assumption that bounds the Hessian by a linear function of the gradient norm. There is a line of work designing algorithms for single-level relaxed smooth functions and showing convergence rates to first-order stationary points (Zhang et al., 2020b;a; Jin et al., 2021; Crawshaw et al., 2022; Li et al., 2023b; Faw et al., 2023; Wang et al., 2023). However, they are only restricted to single-level problems. It remains unclear how to solve bilevel optimization problems when the upper-level function exhibits potential unbounded smoothness (i.e., $(L_{\boldsymbol{x},0}, L_{\boldsymbol{x},1}, L_{\boldsymbol{y},0}, L_{\boldsymbol{y},1})$-smoothness [2]).

Designing efficient bilevel optimization algorithms in the presence of unbounded smooth upper-level problems poses two primary challenges. First, given the upper-level variable, the gradient estimate of the bilevel problem (i.e., the hypergradient estimate) is highly sensitive to the quality of the estimated lower-level optimal solution: an inaccurate lower-level variable will significantly amplify the estimation error of the hypergradient. Second, the bias in the hypergradient estimator depends on both the approximation error of the lower-level solution and the hypergradient itself, which are statistically dependent and difficult to handle. These challenges do not appear in the literature on bilevel optimization with bounded smooth upper-level problems.

In this work, we introduce a new algorithm, namely Bilevel Optimization with lower-level initialization REfinement and Periodic updates (BO-REP), to address these challenges. Compared with the existing bilevel optimization algorithm for nonconvex smooth upper-level problems (Ghadimi & Wang, 2018; Grazzi et al., 2020; Ji et al., 2021; Hong et al., 2023; Kwon et al., 2023a), our algorithm has the following distinct features. Specifically, (1) inspired by the single-level optimization algorithms for unbounded smooth functions (Jin et al., 2021; Crawshaw et al., 2022), our algorithm updates the upper-level variable using normalized momentum to control the effects of stochastic gradient noise and possibly unbounded gradients. (2) The update rule of the lower-level variable relies on two new techniques: *initialization refinement* and *periodic updates*. In particular, when the upper-level variable is initialized, our algorithm invokes a subroutine to run a first-order algorithm for the lower-level variable given the fixed initialized upper-level variable. In addition, the lower-

---

[2]The formal definition of $(L_{\boldsymbol{x},0}, L_{\boldsymbol{x},1}, L_{\boldsymbol{y},0}, L_{\boldsymbol{y},1})$ is illustrated in Assumption 1.

level variable is updated only after every specific period instead of every iteration. This particular treatment for the lower-level variable is due to the difficulty brought by the unbounded smoothness of the upper-level function. Our major contributions are summarized as follows.

- We design a new algorithm named BO-REP, the first algorithm for solving bilevel optimization problems with unbounded smooth upper-level functions. The algorithm design introduces two novel techniques for updating the lower-level variable: initialization refinement and periodic updates. To the best of our knowledge, these techniques are new and not leveraged by the existing literature on bilevel optimization.

- When the upper-level problem is nonconvex and unbounded smooth and the lower-level problem is strongly convex, we prove that BO-REP finds $\epsilon$-stationary points in $\widetilde{O}(1/\epsilon^4)$ iterations, where each iteration invokes a stochastic gradient or Hessian vector product oracle. Notably, this result matches the state-of-the-art complexity results under bounded smoothness setting up to logarithmic factors. The detailed comparison of our algorithm and existing bilevel optimization algorithms (e.g., setting, complexity results) are listed in Table 1. Due to the large body of work on bilevel optimization and limit space, we refer the interested reader to Appendix A, which gives a comprehensive survey of related previous methods that are not covered in Table 1.

- We conduct experiments on hyper-representation learning, hyperparameter optimization, and data hyper-cleaning for text classification tasks. We show that the BO-REP algorithm consistently outperforms other bilevel optimization algorithms.

## 2 PRELIMINARIES AND PROBLEM SETUP

In this paper, we use $\langle \cdot, \cdot \rangle$ and $\| \cdot \|$ to denote the inner product and Euclidean norm. We denote $f$: $\mathbb{R}^{d_x} \times \mathbb{R}^{d_y} \to \mathbb{R}$ as the upper-level function, and $g$: $\mathbb{R}^{d_x} \times \mathbb{R}^{d_y} \to \mathbb{R}$ as the lower-level function. Denote $\nabla \Phi(\boldsymbol{x})$ as the hypergradient, and it is shown in Ghadimi & Wang (2018) that

$$
\begin{aligned}
\nabla \Phi(\boldsymbol{x}) &= \nabla_{\boldsymbol{x}} f(\boldsymbol{x}, \boldsymbol{y}^*(\boldsymbol{x})) - \nabla_{\boldsymbol{x}} \nabla_{\boldsymbol{y}} g(\boldsymbol{x}, \boldsymbol{y}^*(\boldsymbol{x})) [\nabla_{\boldsymbol{y}}^2 g(\boldsymbol{x}, \boldsymbol{y}^*(\boldsymbol{x}))]^{-1} \nabla_{\boldsymbol{y}} f(\boldsymbol{x}, \boldsymbol{y}^*(\boldsymbol{x})) \\
&= \nabla_{\boldsymbol{x}} f(\boldsymbol{x}, \boldsymbol{y}^*(\boldsymbol{x})) - \nabla_{\boldsymbol{x}} \nabla_{\boldsymbol{y}} g(\boldsymbol{x}, \boldsymbol{y}^*(\boldsymbol{x})) \boldsymbol{z}^*(\boldsymbol{x}),
\end{aligned}
\tag{2}
$$

where $\boldsymbol{z}^*(\boldsymbol{x}) = [\nabla_{\boldsymbol{y}}^2 g(\boldsymbol{x}, \boldsymbol{y}^*(\boldsymbol{x}))]^{-1} \nabla_{\boldsymbol{y}} f(\boldsymbol{x}, \boldsymbol{y}^*(\boldsymbol{x}))$ is the solution to the linear system $\boldsymbol{z}^*(\boldsymbol{x}) = \arg\min_{\boldsymbol{z}} \frac{1}{2} \langle \nabla_{\boldsymbol{y}}^2 g(\boldsymbol{x}, \boldsymbol{y}^*(\boldsymbol{x})) \boldsymbol{z}, \boldsymbol{z} \rangle - \langle \nabla_{\boldsymbol{y}} f(\boldsymbol{x}, \boldsymbol{y}^*(\boldsymbol{x})), \boldsymbol{z} \rangle$. We aim to solve the bilevel optimization problem (1) by stochastic methods, where the algorithm can access stochastic gradients and Hessian vector products. We will use the following assumptions.

**Assumption 1** (($L_{\boldsymbol{x},0}, L_{\boldsymbol{x},1}, L_{\boldsymbol{y},0}, L_{\boldsymbol{y},1}$)-smoothness). *Define $\boldsymbol{u} = (\boldsymbol{x}, \boldsymbol{y})$ and $\boldsymbol{u}' = (\boldsymbol{x}', \boldsymbol{y}')$, there exists $L_{\boldsymbol{x},0}, L_{\boldsymbol{x},1}, L_{\boldsymbol{y},0}, L_{\boldsymbol{y},1}$ such that $\|\nabla_{\boldsymbol{x}} f(\boldsymbol{u}) - \nabla_{\boldsymbol{x}} f(\boldsymbol{u}')\| \le (L_{\boldsymbol{x},0} + L_{\boldsymbol{x},1}\|\nabla_{\boldsymbol{x}} f(\boldsymbol{u})\|)\|\boldsymbol{u} - \boldsymbol{u}'\|$ and $\|\nabla_{\boldsymbol{y}} f(\boldsymbol{u}) - \nabla_{\boldsymbol{y}} f(\boldsymbol{u}')\| \le (L_{\boldsymbol{y},0} + L_{\boldsymbol{y},1}\|\nabla_{\boldsymbol{y}} f(\boldsymbol{u})\|)\|\boldsymbol{u} - \boldsymbol{u}'\|$ if $\|\boldsymbol{u} - \boldsymbol{u}'\| \le 1/\sqrt{2(L_{\boldsymbol{x},1}^2 + L_{\boldsymbol{y},1}^2)}$.*

**Remark**: Assumption 1 is a generalization of the relaxed smoothness assumption (Zhang et al., 2020b;a) for a single-level problem (described in Section B.1 and Section B.2 in Appendix). A generalized version of the relaxed smoothness assumption is the coordinate-wise relaxed smoothness assumption (Crawshaw et al., 2022), which is more fine-grained and applies to each coordinate separately. However, these assumptions are designed exclusively for single-level problems. Our $(L_{\boldsymbol{x},0}, L_{\boldsymbol{x},1}, L_{\boldsymbol{y},0}, L_{\boldsymbol{y},1})$-smoothness assumption for the upper-level function $f$ can be regarded as the relaxed smoothness assumption in the bilevel optimization setting, where we need to have different constants to characterize the upper-level variable $\boldsymbol{x}$ and the lower-level variable $\boldsymbol{y}$ respectively. It can recover the standard relaxed smoothness assumption (e.g., Remark 2.3 in Zhang et al. (2020a)) when $L_{\boldsymbol{x},0} = L_{\boldsymbol{y},0} = L_0/2$ and $L_{\boldsymbol{x},1} = L_{\boldsymbol{y},1} = L_1/2$. The details of this derivation are included in Lemma B.3 in Appendix B. Assumption 1 is empirically verified in Appendix G.

**Assumption 2.** *The function $f(\boldsymbol{x}, \boldsymbol{y})$ and $g(\boldsymbol{x}, \boldsymbol{y})$ satisfy the following: (i) There exists $M > 0$ such that for any $\boldsymbol{x}$, $\|\nabla_{\boldsymbol{y}} f(\boldsymbol{x}, \boldsymbol{y}^*(\boldsymbol{x}))\| \le M$; (ii) The derivative $\|\nabla_{\boldsymbol{x}} \nabla_{\boldsymbol{y}} g(\boldsymbol{u})\|$ is bounded, i.e., for any $\boldsymbol{u} = (\boldsymbol{x}, \boldsymbol{y})$, $\|\nabla_{\boldsymbol{x}} \nabla_{\boldsymbol{y}} g(\boldsymbol{u})\| \le C_{g_{xy}}$; (iii) The lower function $g(\boldsymbol{x}, \boldsymbol{y})$ is $\mu$-strongly convex with respect to $\boldsymbol{y}$; (iv) $g(\boldsymbol{u})$ is $L$-smooth, i.e., for any $\boldsymbol{u} = (\boldsymbol{x}, \boldsymbol{y}), \boldsymbol{u}' = (\boldsymbol{x}', \boldsymbol{y}')$, $\|\nabla g(\boldsymbol{u}) - \nabla g(\boldsymbol{u}')\| \le L\|\boldsymbol{u} - \boldsymbol{u}'\|$; (v) The derivatives $\nabla_{\boldsymbol{x}} \nabla_{\boldsymbol{y}} g(\boldsymbol{u})$ and $\nabla_{\boldsymbol{y}}^2 g(\boldsymbol{u})$ are $\tau$- and $\rho$-Lipschitz, i.e., for any $\boldsymbol{u}, \boldsymbol{u}'$, $\|\nabla_{\boldsymbol{x}} \nabla_{\boldsymbol{y}} g(\boldsymbol{u}) - \nabla_{\boldsymbol{x}} \nabla_{\boldsymbol{y}} g(\boldsymbol{u}')\| \le \tau\|\boldsymbol{u} - \boldsymbol{u}'\|, \|\nabla_{\boldsymbol{y}}^2 g(\boldsymbol{u}) - \nabla_{\boldsymbol{y}}^2 g(\boldsymbol{u}')\| \le \rho\|\boldsymbol{u} - \boldsymbol{u}'\|$.*

**Remark**: Assumption 2 is standard in the bilevel optimization literature (Ghadimi & Wang, 2018; Grazzi et al., 2020; Ji et al., 2021; Hong et al., 2023; Kwon et al., 2023a) and we have followed the

same assumptions here. Under Assumption 1 and 2, we can show that the function $\Phi(\boldsymbol{x})$ satisfies standard relaxed smoothness condition: $\|\nabla\Phi(\boldsymbol{x}) - \nabla\Phi(\boldsymbol{x}')\| \leq (K_0 + K_1\|\nabla\Phi(\boldsymbol{x}')\|)\|\boldsymbol{x} - \boldsymbol{x}'\|$ with constant $K_0$ and $K_1$ if $\boldsymbol{x}$ and $\boldsymbol{x}'$ are not far away from each other (i.e., Lemma C.3 in Appendix C).

**Assumption 3.** *We access gradients and Hessian/Jacobian-vector products of the objective functions by unbiased stochastic estimators. The stochastic estimators have the following properties:*

$$\mathbb{E}_{\zeta\sim\mathcal{D}_f}\left[\nabla_{\boldsymbol{x}}F(\boldsymbol{x},\boldsymbol{y};\zeta)\right] = \nabla_{\boldsymbol{x}}f(\boldsymbol{x},\boldsymbol{y}), \quad \mathbb{E}_{\zeta\sim\mathcal{D}_f}\left[\|\nabla_{\boldsymbol{x}}F(\boldsymbol{x},\boldsymbol{y};\zeta) - \nabla_{\boldsymbol{x}}f(\boldsymbol{x},\boldsymbol{y})\|^2\right] \leq \sigma_{f,1}^2,$$

$$\mathbb{E}_{\zeta\sim\mathcal{D}_f}\left[\nabla_{\boldsymbol{y}}F(\boldsymbol{x},\boldsymbol{y};\zeta)\right] = \nabla_{\boldsymbol{y}}f(\boldsymbol{x},\boldsymbol{y}), \quad \mathbb{E}_{\zeta\sim\mathcal{D}_f}\left[\|\nabla_{\boldsymbol{y}}F(\boldsymbol{x},\boldsymbol{y};\zeta) - \nabla_{\boldsymbol{y}}f(\boldsymbol{x},\boldsymbol{y})\|^2\right] \leq \sigma_{f,2}^2,$$

$$\mathbb{E}_{\xi\sim\mathcal{D}_g}\left[\nabla_{\boldsymbol{y}}G(\boldsymbol{x},\boldsymbol{y};\xi)\right] = \nabla_{\boldsymbol{y}}g(\boldsymbol{x},\boldsymbol{y}), \quad \mathbb{E}_{\xi\sim\mathcal{D}_g}\left[\exp\left(\|\nabla_{\boldsymbol{y}}G(\boldsymbol{x},\boldsymbol{y};\xi) - \nabla_{\boldsymbol{y}}g(\boldsymbol{x},\boldsymbol{y})\|^2/\sigma_{g,1}^2\right)\right] \leq \exp(1),$$

$$\mathbb{E}_{\xi\sim\mathcal{D}_g}\left[\nabla_{\boldsymbol{y}}^2G(\boldsymbol{x},\boldsymbol{y};\xi)\right] = \nabla_{\boldsymbol{y}}^2g(\boldsymbol{x},\boldsymbol{y}), \quad \mathbb{E}_{\xi\sim\mathcal{D}_g}\left[\left\|\nabla_{\boldsymbol{y}}^2G(\boldsymbol{x},\boldsymbol{y};\xi) - \nabla_{\boldsymbol{y}}^2g(\boldsymbol{x},\boldsymbol{y})\right\|^2\right] \leq \sigma_{g,2}^2,$$

$$\mathbb{E}_{\xi\sim\mathcal{D}_g}\left[\nabla_{\boldsymbol{x}}\nabla_{\boldsymbol{y}}G(\boldsymbol{x},\boldsymbol{y};\xi)\right] = \nabla_{\boldsymbol{x}}\nabla_{\boldsymbol{y}}g(\boldsymbol{x},\boldsymbol{y}), \quad \mathbb{E}_{\xi\sim\mathcal{D}_g}\left[\|\nabla_{\boldsymbol{x}}\nabla_{\boldsymbol{y}}G(\boldsymbol{x},\boldsymbol{y};\xi) - \nabla_{\boldsymbol{x}}\nabla_{\boldsymbol{y}}g(\boldsymbol{x},\boldsymbol{y})\|^2\right] \leq \sigma_{g,2}^2.$$

**Remark**: Assumption 3 requires the stochastic estimators to be unbiased and have bounded variance and are standard in the literature (Ghadimi & Lan, 2013b; Ghadimi & Wang, 2018). In addition, we need the stochastic gradient noise of function $g$ to be light-tail. It is a technical assumption for high probability analysis for $\boldsymbol{y}$, which is typical for algorithm analysis in the single-level convex and nonconvex optimization problems (Lan, 2012; Hazan & Kale, 2014; Ghadimi & Lan, 2013b).

**Definition 2.1.** $\boldsymbol{x} \in \mathbb{R}^{d_x}$ *is an $\epsilon$-stationary point of the bilevel problem (1) if* $\|\nabla\Phi(\boldsymbol{x})\| \leq \epsilon$.

**Remark**: In nonconvex optimization literature (Ghadimi & Lan, 2013b; Ghadimi & Wang, 2018; Zhang et al., 2020b), the typical goal is to find an $\epsilon$-stationary point since it is NP-hard in general for finding a global minimum in nonconvex optimization (Hillar & Lim, 2013).

## 3 ALGORITHM AND THEORETICAL ANALYSIS

### 3.1 MAIN CHALLENGES AND ALGORITHM DESIGN

**Main Challenges**. We first illustrate why previous bilevel optimization algorithms and analyses cannot solve our problem. The main idea of the convergence analyses of the existing bilevel optimization algorithms (Ghadimi & Wang, 2018; Grazzi et al., 2020; Ji et al., 2021; Hong et al., 2023; Dagréou et al., 2022; Kwon et al., 2023a; Chen et al., 2023b) is approximating hypergradient (2) and employ the approximate hypergradient descent to update the upper-level variable. The hypergradient approximation is required because the optimal lower-level solution $\boldsymbol{y}^*(\boldsymbol{x})$ cannot be easily obtained. The typical approximation scheme requires to approximate $\boldsymbol{y}^*(\boldsymbol{x})$ and also the matrix-inverse vector product $\boldsymbol{z}^*(\boldsymbol{x})$ by solving a linear system approximately. When the upper-level function has a bounded smoothness parameter, these approximation errors cannot blow up, and they can be easily controlled. However, when the upper-level function is $(L_{\boldsymbol{x},0}, L_{\boldsymbol{x},1}, L_{\boldsymbol{y},0}, L_{\boldsymbol{y},1})$-smooth as illustrated in Assumption 1, an inaccurate lower-level variable will significantly amplify the estimation error of upper-level gradient: the estimation error explicitly depends on the magnitude of the gradient of the upper-level problem and it can be arbitrarily large (e.g., gradient explosion problem in RNN (Pascanu et al., 2013)). In addition, in a stochastic optimization setting, the bias in the hypergradient estimator depends on both the approximation error of the lower-level variable and the hypergradient in terms of the upper-level variable, which are statistically dependent and difficult to analyze. Therefore, existing bilevel optimization algorithms cannot be utilized to address our problems where the upper-level problem exhibits unbounded smoothness.

**Algorithm Design**. To address these challenges, our key idea is to update the upper-level variable by the momentum normalization technique and a careful update procedure for the lower-level variable. The normalized momentum update for the upper-level variable has two critical goals. First, it reduces the effects of stochastic gradient noise and also reduces the effects of unbounded smoothness and gradient norms, which can regarded as a generalization of techniques of (Cutkosky & Mehta, 2020; Jin et al., 2021; Crawshaw et al., 2022) under the bilevel optimization setting. The main difference in our case is that we need to explicitly deal with the bias in the hypergradient estimator. Second, the normalized momentum update can ensure that the upper-level iterates move slowly, indicating that the corresponding optimal lower-level solutions move slowly as well due to the Lipschitzness of the mapping $\boldsymbol{y}^*(\boldsymbol{x})$. This important fact enables us to design initialization refinement to obtain an accurate estimate of the optimal lower-level variable for the initialization, and the slowly changing optimal lower-level solutions allow us to perform periodic updates for updating the lower-level variable. As a result, we can obtain accurate estimates for $\boldsymbol{y}^*(\boldsymbol{x})$ at every iteration.

---

**Algorithm 1:** BO-REP

---

**Input:** $\boldsymbol{x}_0, \boldsymbol{y}_0', \boldsymbol{z}_0, \boldsymbol{m}_0 = \boldsymbol{0}$; $\beta, \eta, \nu, \gamma, \alpha_0$

1   $\boldsymbol{y}_0 = \texttt{Epoch-SGD}(\boldsymbol{x}_0, \boldsymbol{y}_0', \alpha_0)$                  # initialization refinement

2   **for** $k = 0, 1, \ldots, K - 1$ **do**

3     $\boldsymbol{y}_{k+1} = \texttt{UpdateLower}(\boldsymbol{x}_k, \boldsymbol{y}_k, \gamma; \tilde{\xi}_k)^{\,a}$            # periodic updates

4     $\boldsymbol{z}_{k+1} = \boldsymbol{z}_k - \nu(\nabla_{\boldsymbol{y}}^2 G(\boldsymbol{x}_k, \boldsymbol{y}_k; \xi_k)\boldsymbol{z}_k - \nabla_{\boldsymbol{y}} F(\boldsymbol{x}_k, \boldsymbol{y}_k; \zeta_k))$

5     $\boldsymbol{m}_{k+1} = \beta \boldsymbol{m}_k + (1 - \beta)\left[\nabla_{\boldsymbol{x}} F(\boldsymbol{x}_k, \boldsymbol{y}_k; \zeta_k) - \nabla_{\boldsymbol{x}} \nabla_{\boldsymbol{y}} G(\boldsymbol{x}_k, \boldsymbol{y}_k; \xi_k)\boldsymbol{z}_k\right]$

6     $\boldsymbol{x}_{k+1} = \boldsymbol{x}_k - \eta \frac{\boldsymbol{m}_{k+1}}{\|\boldsymbol{m}_{k+1}\|}$

7   **end**

---

    $^a$We only sample $\tilde{\xi}_k = \cup_{t=0}^{N-1}\{\tilde{\xi}_k^t\}$ when $k = jI$, where $1 \leq j \leq \lfloor \frac{K}{I} \rfloor$ and $I$ denotes the update frequency for $\boldsymbol{y}_k$ in Algorithm 2; we do not update $\boldsymbol{y}_k$ and hence do not sample $\tilde{\xi}_k$ when $k \neq jI$ (i.e., $\tilde{\xi}_k = \emptyset$ for $k \neq jI$ and $j \geq 1$).

---

**Algorithm 2:** UpdateLower

---

**Input:** $\boldsymbol{x}, \boldsymbol{y}_k, \gamma, \tilde{\xi}_k$

1   **if** $k > 0$ *and* $k$ *is a multiple of* $I$ **then**

2     $\boldsymbol{y}_k^0 = \boldsymbol{y}_k$

3     **for** $t = 0, ..., N - 1$ **do**

4       $\boldsymbol{y}_k^{t+1} =$

        $\Pi_{\mathcal{B}(\boldsymbol{y}_k^0, R)}\left(\boldsymbol{y}_k^t - \gamma \nabla_{\boldsymbol{y}} G(\boldsymbol{x}, \boldsymbol{y}_k^t; \tilde{\xi}_k^t)\right)$

5       $\bar{\boldsymbol{y}}_k^{t+1} = \frac{t}{t+1}\bar{\boldsymbol{y}}_k^t + \frac{1}{t+1}\boldsymbol{y}_k^{t+1}$

6     $\boldsymbol{y}_{k+1} = \bar{\boldsymbol{y}}_k^N$

7   **else**

8     $\boldsymbol{y}_{k+1} = \boldsymbol{y}_k$

9   **return** $y_{k+1}$

---

**Algorithm 3:** Epoch-SGD

---

**Input:** $\boldsymbol{x}_0, \boldsymbol{y}_0^{0,0}, \alpha_0$

1   **Initialize:** $\mathcal{B}_0, T_0, k^\dagger; s = 0$

2   **for** $s = 0, 1, \ldots, k^\dagger - 1$ **do**

3     **for** $t = 0, ..., T_s - 1$ **do**

4       $\boldsymbol{y}_0^{s,t+1} =$

        $\Pi_{\mathcal{B}_s}\left(\boldsymbol{y}_0^{s,t} - \alpha_s \nabla_{\boldsymbol{y}} G(\boldsymbol{x}_0, \boldsymbol{y}_0^{s,t}; \tilde{\xi}_0^{s,t})\right)$

5     **end**

6     $\boldsymbol{y}_0^{s+1,0} = \frac{1}{T_s}\sum_{t=1}^{T_s} \boldsymbol{y}_0^{s,t}$

7     Update $\mathcal{B}_s, T_s, \alpha_s$ via (28), (29), (30)

8   **end**

9   **return** $\boldsymbol{y}_0^{k^\dagger,0}$

---

The detailed framework is described in Algorithm 1. Specifically, we first run a variant of epoch SGD for the smooth and strongly convex lower-level problem (Ghadimi & Lan, 2013a; Hazan & Kale, 2014) (line 1) to get an initialization refinement. Once the upper-level variable $\boldsymbol{x}_0$ is initialized, we need to get an $\boldsymbol{y}_0$ close enough to $\boldsymbol{y}^*(\boldsymbol{x}_0)$. Then, the algorithm updates the upper-level variable $\boldsymbol{x}$, lower-level variable $\boldsymbol{y}$, and the approximate linear system solution $\boldsymbol{z}$ within a loop (line 2∼7). In particular, we keep a momentum buffer to store the moving average of the history hypergradient estimators (line 5), and use normalized momentum updates for $\boldsymbol{x}$ (line 6), stochastic gradient descent update for $\boldsymbol{z}$ (line 4) and periodic stochastic gradient descent with projection updates for $\boldsymbol{y}$ (line 3). Note that $\mathcal{B}(\hat{\boldsymbol{y}}, R) := \left\{\boldsymbol{y} \in \mathbb{R}^{d_y} : \|\boldsymbol{y} - \hat{\boldsymbol{y}}\| \leq R\right\}$ denotes a ball centered at $\hat{\boldsymbol{y}}$ with radius $R$, $\Pi$ is denoted as the projection operator.

### 3.2   MAIN RESULTS

We will first define a few concepts. Let $\mathcal{F}_k$ denote the filtration of the random variables for updating $\boldsymbol{z}_k, \boldsymbol{m}_k$ and $\boldsymbol{x}_k$ before iteration $k$, i.e., $\mathcal{F}_k := \sigma\left\{\xi_0, \ldots, \xi_{k-1}, \zeta_0, \ldots, \zeta_{k-1}\right\}$ for any $k \geq 1$, where $\sigma\{\cdot\}$ denotes the $\sigma$-algebra generated by the random variables. Let $\widetilde{\mathcal{F}}_0^{s,t}$ denote the filtration of the random variables for updating lower-level variable $\boldsymbol{y}_0$ starting at the $s$-th epoch before iteration $t$ in Algorithm 3, i.e., $\widetilde{\mathcal{F}}_0^{s,t} := \sigma\{\tilde{\xi}_0^{s,0}, \ldots, \tilde{\xi}_0^{s,t-1}\}$ for $1 \leq t \leq T_s$ and $0 \leq s \leq k^\dagger - 1$, which contains all randomness in Algorithm 3. Let $\widetilde{\mathcal{F}}_k^t$ denote the filtration of the random variables for updating lower-level variable $\boldsymbol{y}_k$ ($k \geq 1$) before iteration $t$ in Algorithm 2, i.e., $\widetilde{\mathcal{F}}_k^t := \sigma\{\tilde{\xi}_k^0, \ldots, \tilde{\xi}_k^{t-1}\}$ for $1 \leq t \leq N$ and $k = jI$, where $1 \leq j \leq \lfloor \frac{K}{I} \rfloor$ and $I$ denotes the update frequency for $\boldsymbol{y}_k$ in Algorithm 2. Let $\widetilde{\mathcal{F}}_K$ denote the filtration of all random variables for updating lower-level variable $\boldsymbol{y}_k$ ($k \geq 0$), i.e., $\widetilde{\mathcal{F}}_K := \sigma\left\{\left(\cup_{s=0}^{k^\dagger-1} \widetilde{\mathcal{F}}_0^{s,T_s}\right) \cup \left(\cup_{k=1}^{K-1} \widetilde{\mathcal{F}}_k^N\right)\right\}$. For the overview of notations used in this paper, please check our Table 2 in Appendix.

**Theorem 3.1.** *Suppose Assumptions 1, 2 and 3 hold. Run Algorithm 1 for $K$ iterations and let $\{\boldsymbol{x}_k\}_{k\geq 0}$ be the sequence produced by Algorithm 1. For $\epsilon \leq \min\left( \frac{K_0}{K_1}, \sqrt{\frac{\sigma_{f,1}^2 + \frac{2M^2}{\mu^2}\sigma_{g,2}^2}{\min\left(1, \frac{\mu^2}{32C_{g_{xy}}^2}\right)}} \right)$ and given $\delta \in (0,1)$, if we choose $\alpha_s$ as (30), $\gamma$ as (44), $N$ as (45) and $I = \frac{\sigma_{g,1}^2 K_0^2}{\mu^2 \epsilon^2}$, $1 - \beta = \min\left( \frac{\epsilon^2}{\sigma_{f,1}^2 + \frac{2M^2}{\mu^2}\sigma_{g,2}^2} \min\left(1, \frac{\mu^2}{32C_{g_{xy}}^2}\right), \frac{C_{g_{xy}}^2}{8\sigma_{g,2}^2}, \frac{\mu^2}{16\sigma_{g,2}^2}, \frac{1}{4} \right)$, $\nu = \frac{1}{\mu}(1-\beta)$,*

$$\eta = \min\left( \frac{1}{8}\min\left( \frac{1}{K_1}, \frac{\epsilon}{K_0}, \frac{\Delta}{\|\nabla\Phi(\boldsymbol{x}_0)\|}, \frac{\epsilon\Delta}{C_{g_{xy}}^2 \Delta_{\boldsymbol{z},0}} \right)(1-\beta), \frac{1}{\sqrt{2\left(1 + \frac{C_{g_{xy}}^2}{\mu^2}\right)(L_{\boldsymbol{x},1}^2 + L_{\boldsymbol{y},1}^2)}}, \frac{\mu\epsilon}{8K_0 I C_{g_{xy}}} \right),$$

*where $\Delta := \Phi(\boldsymbol{x}_0) - \inf_{\boldsymbol{x}\in\mathbb{R}^{d_x}} \Phi(\boldsymbol{x})$ and $\Delta_{\boldsymbol{z},0} := \|\boldsymbol{z}_0 - \boldsymbol{z}^*(\boldsymbol{x}_0)\|^2$, then with probability at least $1-\delta$ over the randomness in $\widetilde{\mathcal{F}}_K$, Algorithm 1 guarantees $\frac{1}{K}\sum_{k=0}^{K-1} \mathbb{E}\|\nabla\Phi(\boldsymbol{x}_k)\| \leq 30\epsilon$ as long as $K = \frac{4\Delta}{\eta\epsilon}$, where the expectation is taken over the randomness in $\mathcal{F}_K$. In addition, the number of oracle calls for updating lower-level variable $\boldsymbol{y}$ (in Algorithm 2 and Algorithm 3 is at most $\widetilde{O}\left(\frac{\Delta}{\eta\epsilon}\right)$.*

**Remark**: Theorem 3.1 indicates that Algorithm 1 requires a total $\widetilde{O}(\epsilon^{-4})$ oracle complexity for finding an $\epsilon$-stationary point in expectation, which matches the state-of-the-art complexity results in non-convex bilevel optimization with bounded smooth upper-level problem (Dagréou et al., 2022; Chen et al., 2023b) and without mean-squared smoothness assumption on the stochastic oracle [3]. Please note that our complexity is optimal up to logarithmic factors due to the $\Omega(\epsilon^{-4})$ complexity lower bounds in the nonconvex stochastic single-level optimization for finding $\epsilon$-stationary points (Arjevani et al., 2023). More detailed statement of the optimality is described in Appendix L.

## 3.3 SKETCH OF THE PROOF

In this section, we present the sketch of the proof of Theorem 3.1. The detailed proof can be found in Appendix E. Define $\boldsymbol{y}_k^* = \boldsymbol{y}^*(\boldsymbol{x}_k)$, $\boldsymbol{z}_k^* = \boldsymbol{z}^*(\boldsymbol{x}_k)$ and $\widehat{\nabla}\Phi(\boldsymbol{x}_k, \boldsymbol{y}_k, \boldsymbol{z}_k; \zeta_k, \xi_k) = \nabla_{\boldsymbol{x}}F(\boldsymbol{x}_k, \boldsymbol{y}_k; \zeta_k) - \nabla_{\boldsymbol{x}}\nabla_{\boldsymbol{y}}G(\boldsymbol{x}_k, \boldsymbol{y}_k; \xi_k)\boldsymbol{z}_k$. We use $\mathbb{E}_k$, $\mathbb{E}_{\mathcal{F}_k}$ and $\mathbb{E}$ to denote the conditional expectation $\mathbb{E}\left[\cdot \mid \mathcal{F}_k\right]$, the expectation on $\mathcal{F}_k$ and the total expectation over all randomness in $\mathcal{F}_K$, respectively. The main difficulty comes from the bias term $\|\mathbb{E}_k[\widehat{\nabla}\Phi(\boldsymbol{x}_k, \boldsymbol{y}_k, \boldsymbol{z}_k; \zeta_k, \xi_k)] - \nabla\Phi(\boldsymbol{x}_k)\|$ of the hypergradient estimator, whose upper bound depends on $L_{\boldsymbol{x},1}\|\boldsymbol{y}_k - \boldsymbol{y}_k^*\|\|\nabla\Phi(\boldsymbol{x}_k)\|$ by Assumption 1. This quantity is difficult to handle because (i) $\|\nabla\Phi(\boldsymbol{x}_k)\|$ can be large; (ii) both $\|\boldsymbol{y}_k - \boldsymbol{y}_k^*\|$ and $\|\nabla\Phi(\boldsymbol{x}_k)\|$ are measurable with respect to $\mathcal{F}_{k-1}$ and it is difficult to decouple them when taking total expectation.

To address these issues, we introduce Lemma 3.2, 3.3 and 3.4 to control the lower-level error, and hence control the bias in the hypergradient estimator *with high probability* over the randomness in $\widetilde{\mathcal{F}}_K$ as illustrated in Lemma 3.5. Then we analyze the expected hypergradient error between the moving-average estimator (i.e., momentum) and the hypergradient, as illustrated in Lemma 3.6, where the expectation is taken over the randomness in $\mathcal{F}_K$. Lastly we plug in these lemmas to the descent lemma for $(L_{\boldsymbol{x},0}, L_{\boldsymbol{x},1}, L_{\boldsymbol{y},0}, L_{\boldsymbol{y},1})$-smooth functions and obtain the main theorem.

**Lemma 3.2** (Initialization Refinement). *Given $\delta \in (0,1)$ and $\epsilon > 0$, set the parameter $k^\dagger = \left\lceil \log_2(128K_0^2 \mathcal{V}_0/\mu^2\epsilon^2) \right\rceil$, where $\mathcal{V}_0$ is defined in (27). If we run Algorithm 3 for $k^\dagger$ epochs with output $\boldsymbol{y}_0$, with projection ball $\mathcal{B}_s$, the number of iterations $T_s$ and the fixed step-size $\alpha_s$ at each epoch defined as (28), (29) and (30). Then with probability at least $1 - \delta/2$ over randomness in $\sigma\left\{\cup_{s=0}^{k^\dagger -1}\widetilde{\mathcal{F}}_0^{s,T_s}\right\}$ (this event is denoted as $\mathcal{E}_0$), we have $\|\boldsymbol{y}_0 - \boldsymbol{y}^*(\boldsymbol{x}_0)\| \leq \epsilon/8K_0$ in $\widetilde{O}\left(\sigma_{g,1}^2 K_0^2/\mu^2\epsilon^2\right)$ iterations.*

**Remark.** More detailed statement of Lemma 3.2 can be found in Appendix D.2. Lemma 3.2 provides a complexity result for getting a good estimate of $\boldsymbol{y}^*(\boldsymbol{x}_0)$ with high probablity. The next lemma (i.e., Lemma 3.3) is built upon this lemma.

---

[3]Note that there are a few works which achieve $\widetilde{O}(\epsilon^{-3})$ oracle complexity when the stochastic function is mean-squared smooth (Yang et al., 2021; Guo et al., 2021; Khanduri et al., 2021), but our paper does not make such an assumption.

**Lemma 3.3** (Periodic Updates). *Given $\delta \in (0,1)$ and $\epsilon > 0$, choose $R = \frac{\epsilon}{4K_0}$. Under $\mathcal{E}_0$, for any fixed sequences $\{\widetilde{\boldsymbol{x}}_k\}_{k=1}^K$ such that $\widetilde{\boldsymbol{x}}_0 = \boldsymbol{x}_0$ and $\|\widetilde{\boldsymbol{x}}_{k+1} - \widetilde{\boldsymbol{x}}_k\| = \eta$, where $\eta \leq \frac{\mu\epsilon}{8K_0 I C_{g_{xy}}}$, if we run Algorithm 2 with input $\{\widetilde{\boldsymbol{x}}_k\}_{k=1}^K$ and generate outputs $\{\widetilde{\boldsymbol{y}}_k\}_{k=1}^K$, and step-size $\gamma = O(\mu\epsilon^2/K_0^2\sigma_{g,1}^2)$ for $N = \widetilde{O}(\sigma_{g,1}^2 K_0^2/\mu^2\epsilon^2)$ iterations in each update period (the exact formula of $\gamma$ and $N$ are (44) and (45)), then with probability at least $1 - \delta/2$ over randomness in $\sigma\left\{\cup_{k=1}^{K-1}\widetilde{\mathcal{F}}_k^N\right\}$ (this event is denoted as $\mathcal{E}_1$), we have $\|\boldsymbol{y}^*(\widetilde{\boldsymbol{x}}_k) - \widetilde{\boldsymbol{y}}_k\| \leq \epsilon/4K_0$ for any $k \geq 1$ in $\widetilde{O}(K\sigma_{g,1}^2 K_0^2/I\mu^2\epsilon^2)$ iterations.*

**Remark.** More detailed statement of Lemma 3.3 can be found in Appendix D.3. Lemma 3.3 unveils the following important fact: as long as the learning rate $\eta$ is small, the upper-level solution moves slowly, then the corresponding lower-level optimal solution also moves slowly. Therefore, as long as we have a good lower-level variable estimate at the very beginning (e.g., under the event $\mathcal{E}_0$), we do not need to update it every iteration: periodic updates schedule is sufficient to obtain an accurate lower-level solution with high probability at every iteration. In addition, this fact does not depend on any randomness from the upper-level problem, it holds over *any fixed sequence* $\{\widetilde{\boldsymbol{x}}_k\}_{k=1}^K$ as long as $\|\widetilde{\boldsymbol{x}}_{k+1} - \widetilde{\boldsymbol{x}}_k\| = \eta$.

**Lemma 3.4** (Error Control for the Lower-level Problem). *Under event $\mathcal{E} = \mathcal{E}_0 \cap \mathcal{E}_1$, we have $\|\boldsymbol{y}^*(\widetilde{\boldsymbol{x}}_k) - \widetilde{\boldsymbol{y}}_k\| \leq \epsilon/4K_0$ for any $k \geq 0$ and $Pr(\mathcal{E}) \geq 1 - \delta$ and the probability is taken over randomness in $\widetilde{\mathcal{F}}_K$.*

**Remark.** Lemma 3.4 is a direct corollary of Lemma 3.2 and 3.3. It provides a high probability guarantee for the output sequence $\{\widetilde{\boldsymbol{y}}_k\}_{k=1}^K$ in terms of any given input sequence $\{\widetilde{\boldsymbol{x}}_t\}_{t=1}^K$ as long as $\|\widetilde{\boldsymbol{x}}_{k+1} - \widetilde{\boldsymbol{x}}_k\| = \eta$. Note that the event $\mathcal{E} \in \widetilde{\mathcal{F}}_K$ and is independent of the rest randomness in the Algorithm 1 (i.e., $\mathcal{F}_K$). This important aspect of this lemma enables us to plug in the actual sequence $\{\boldsymbol{x}_k\}_{k=1}^K$ in Algorithm 1 without affecting the high probability result. In particular, we will show that under the event $\mathcal{E}$ (which holds with high probability in terms of $\widetilde{\mathcal{F}}_K$), Algorithm 1 will converge to $\epsilon$-stationary point in expectation, where the expectation is taken over randomness in $\mathcal{F}_K$ as illustrated in Theorem 3.1.

**Lemma 3.5** (Bias of the Hypergradient Estimator). *Suppose Assumptions 1, 2 and 3 hold. Then under event $\mathcal{E}$, we have*

$$\left\|\mathbb{E}_k[\widehat{\nabla}\Phi(\boldsymbol{x}_k, \boldsymbol{y}_k, \boldsymbol{z}_k; \zeta_k, \xi_k)] - \nabla\Phi(\boldsymbol{x}_k)\right\| \leq \frac{L_{\boldsymbol{x},1}\epsilon}{4K_0}\|\nabla\Phi(\boldsymbol{x}_k)\| + \left(L_{\boldsymbol{x},0} + L_{\boldsymbol{x},1}\frac{C_{g_{xy}}M}{\mu} + \frac{\tau M}{\mu}\right)\frac{\epsilon}{4K_0} + C_{g_{xy}}\|\boldsymbol{z}_k - \boldsymbol{z}_k^*\|.$$

**Remark.** Lemma 3.5 controls the bias in the hypergradient estimator under the event $\mathcal{E}$. Note that the good event $\mathcal{E}$ make sure that the bias is almost negligible since it depends on small quantities $\epsilon$ and $\|\boldsymbol{z}_k - \boldsymbol{z}_k^*\|$ (Lemma D.5 in Appendix D.5 shows that $\mathbb{E}\|\boldsymbol{z}_k - \boldsymbol{z}_k^*\|$ is small on average).

**Lemma 3.6** (Expected Error of the Moving-Average Hypergradient Estimator). *Suppose Assumptions 1, 2 and 3 hold. Define $\boldsymbol{\delta}_k := \boldsymbol{m}_{k+1} - \nabla\Phi(\boldsymbol{x}_k)$ to be the moving-average estimation error. Then under event $\mathcal{E}$, we have $\mathbb{E}\left[\sum_{k=0}^{K-1}\|\boldsymbol{\delta}_k\|\right] \leq Err_1 + Err_2$, where $Err_1$ and $Err_2$ are defined as*

$$Err_1 := \frac{L_{\boldsymbol{x},1}\epsilon}{4K_0}\sum_{k=0}^{K-1}\|\nabla\Phi(\boldsymbol{x}_k)\| + K\left(L_{\boldsymbol{x},0} + L_{\boldsymbol{x},1}\frac{C_{g_{xy}}M}{\mu} + \frac{\tau M}{\mu}\right)\frac{\epsilon}{4K_0} + C_{g_{xy}}\sqrt{K}\sqrt{\sum_{k=0}^{K-1}\mathbb{E}\left[\|\boldsymbol{z}_k - \boldsymbol{z}_k^*\|^2\right]},$$

$$Err_2 := K\sqrt{1-\beta}\sqrt{\sigma_{f,1}^2 + \frac{2M^2}{\mu^2}\sigma_{g,2}^2} + \sqrt{2}\sigma_{g,2}\sqrt{1-\beta}\sqrt{K}\sqrt{\sum_{k=0}^{K-1}\mathbb{E}\left[\|\boldsymbol{z}_k - \boldsymbol{z}_k^*\|^2\right]}$$

$$+ \frac{K_1\eta\beta}{1-\beta}\sum_{k=0}^{K-1}\|\nabla\Phi(\boldsymbol{x}_k)\| + \frac{K_0 K\eta\beta}{1-\beta} + \frac{\beta}{1-\beta}\|\boldsymbol{m}_0 - \nabla\Phi(\boldsymbol{x}_0)\|.$$

**Remark.** Lemma 3.6 shows that, under the event $\mathcal{E}$, the error can be decomposed as two parts. The $Err_1$ and $Err_2$ represent the error from bias and variance respectively. As long as $1 - \beta$ is small (as chosen in Theorem 3.1), then the accumulated expected error of the moving-average hypergradient estimator grows only with a sublinear rate in $K$, where $K$ is the number of iterations. This fact helps us establish the convergence rate of Algorithm 1. Lemma 3.6 can be seen as a generalization of the normalized momentum update lemma from single-level optimization (e.g., Theorem C.7 in Jin et al. (2021)) to bilevel optimization.

## 4 EXPERIMENTS

### 4.1 HYPER-REPRESENTATION LEARNING FOR TEXT CLASSIFICATION

We conduct experiments on the hyper-representation learning task (i.e., meta-learning) for text classification. The goal is to learn a hyper-representation that can be used for various tasks by simply adjusting task-specific parameters. There are two main components during the learning process: a base learner and a meta learner. The meta learner learns from several tasks in sequence to improve the base learner's performance across tasks (Bertinetto et al., 2018).

The meta-learning contains $m$ tasks $\{\mathcal{T}_i, i = 1, ..., m\}$ sampled from certain distribution $P_{\mathcal{T}}$. The loss function of each task is $\mathcal{L}(\boldsymbol{w}, \boldsymbol{\theta}_i, \xi)$, where $\boldsymbol{w}$ is the hyper-representation (meta learner) which extracts the data features across all the tasks and $\xi$ is the data. $\boldsymbol{\theta}_i$ is the task-specific parameter of a base learner for $i$-th task. The objective is to find the best $\boldsymbol{w}$ to represent the shared feature representation, such that each base learner can quickly adapt its parameter $\boldsymbol{\theta}_i$ to unseen tasks.

This task can be formulated as a bilevel problem (Ji et al., 2021; Hong et al., 2023). In the lower level, the goal of the base learner is to find the minimizer $\boldsymbol{\theta}_i^*$ of its regularized loss on the support set $\mathcal{S}_i$ upon the hyper-representation $\boldsymbol{w}$. In the upper level, the meta learner evaluates all the $\boldsymbol{\theta}_i^*, i = 1, .., m$ on the corresponding query set $\mathcal{Q}_i$, and optimizes the hyperpresentation $\boldsymbol{w}$. Let $\boldsymbol{\theta} = (\boldsymbol{\theta}_1, ..., \boldsymbol{\theta}_m)$ be all the task-specific parameters, the objective function is the following:

$$\min_{\boldsymbol{w}} \frac{1}{m} \sum_{i=1}^{m} \frac{1}{|\mathcal{Q}_i|} \sum_{\xi \in \mathcal{Q}_i} \mathcal{L}(\boldsymbol{w}, \boldsymbol{\theta}_i^*(\boldsymbol{w}); \xi) \text{ s.t. } \boldsymbol{\theta}^*(\boldsymbol{w}) = \arg\min_{\boldsymbol{\theta}} \frac{1}{m} \sum_{i=1}^{m} \frac{1}{|\mathcal{S}_i|} \sum_{\zeta \in \mathcal{S}_i} \mathcal{L}(\boldsymbol{w}, \boldsymbol{\theta}_i; \zeta) + \frac{\mu}{2} \|\boldsymbol{\theta}_i\|^2, \quad (3)$$

where $\mathcal{S}_i$ and $\mathcal{Q}_i$ come from the task $\mathcal{T}_i$. In our experiment, $\boldsymbol{\theta}_i$ is the parameter of the last linear layer of a neural network for classification, and $\boldsymbol{w}$ represents the parameter of a 2-layer recurrent neural network except for the last layer. Therefore, the lower-level function is $\mu$-strongly convex for any given $\boldsymbol{w}$, and the upper-level function is nonconvex in $\boldsymbol{w}$ and has potential unbounded smoothness.

Hyper-representation experiment is conducted over Amazon Review Dataset, consisting of two types of reviews across 25 different products. We compare our algorithm with classical meta-learning algorithms and bilevel optimization algorithms, including MAML (Rajeswaran et al., 2019), ANIL (Raghu et al., 2019), StocBio (Ji et al., 2021), TTSA (Hong et al., 2023), F²SA (Kwon et al., 2023a), SOBA (Dagréou et al., 2022), and MA-SOBA (Chen et al., 2023b). We report both training and test losses. The results are presented in Figure 1(a) and Figure 2(a) (in Appendix F.1), which show the learning process over 20 epochs on the training data and evaluating process on testing data. Our method (i.e., the green curve) significantly outperforms baselines. More experimental details are described in Appendix F.1.

### 4.2 HYPERPARAMETER OPTIMIZATION FOR TEXT CLASSIFICATION

We conduct hyperparameter optimization (Franceschi et al., 2018; Ji et al., 2021) experiments for text classification to demonstrate the effectiveness of our algorithm. Hyperparameter optimization aims to find a suitable regularization parameter $\lambda$ to minimize the loss evaluated over the best model parameter $\boldsymbol{w}^*$ from the lower-level function. The hyperparameter optimization problem can be formulated as:

$$\min_{\lambda} \frac{1}{|\mathcal{D}_{\text{val}}|} \sum_{\xi \in \mathcal{D}_{\text{val}}} \mathcal{L}(\boldsymbol{w}^*(\lambda); \xi), \text{ s.t. } \boldsymbol{w}^*(\lambda) = \arg\min_{\boldsymbol{w}} \frac{1}{|\mathcal{D}_{\text{tr}}|} \sum_{\zeta \in \mathcal{D}_{\text{tr}}} \left( \mathcal{L}(\boldsymbol{w}; \zeta) + \frac{\lambda}{2} \|\boldsymbol{w}\|^2 \right), \quad (4)$$

where $\mathcal{L}(\boldsymbol{w}; \xi)$ is the loss function, $\boldsymbol{w}$ is the model parameter, and $\lambda$ denotes the regularization parameter. $\mathcal{D}_{\text{val}}$ and $\mathcal{D}_{\text{tr}}$ denote validation and training sets respectively. The text classification experiment is performed over the Amazon Review dataset. In our experiment, we compare our algorithm with stochastic bilevel algorithms, including StocBio (Ji et al., 2021), TTSA (Hong et al., 2023), F²SA (Kwon et al., 2023a), SOBA (Dagréou et al., 2022), MA-SOBA (Chen et al., 2023b). As shown in Figure 1(b) and Figure 2(b) (in Appendix F.2), BO-REP achieves the fastest convergence rate and the best performance compared with other bilevel algorithms. More details about hyperparameter settings are described in Appendix F.2.

### 4.3 DATA HYPER-CLEANING FOR TEXT CLASSIFICATION

Consider a noisy training set $\mathcal{D}_{\text{tr}} := \{(x_i, \tilde{y}_i)\}_{i=1}^n$ with label $\tilde{y}_i$ being randomly corrupted with probability $p < 1$ (i.e., corruption rate). The goal of the data hyper-cleaning (Franceschi et al.,

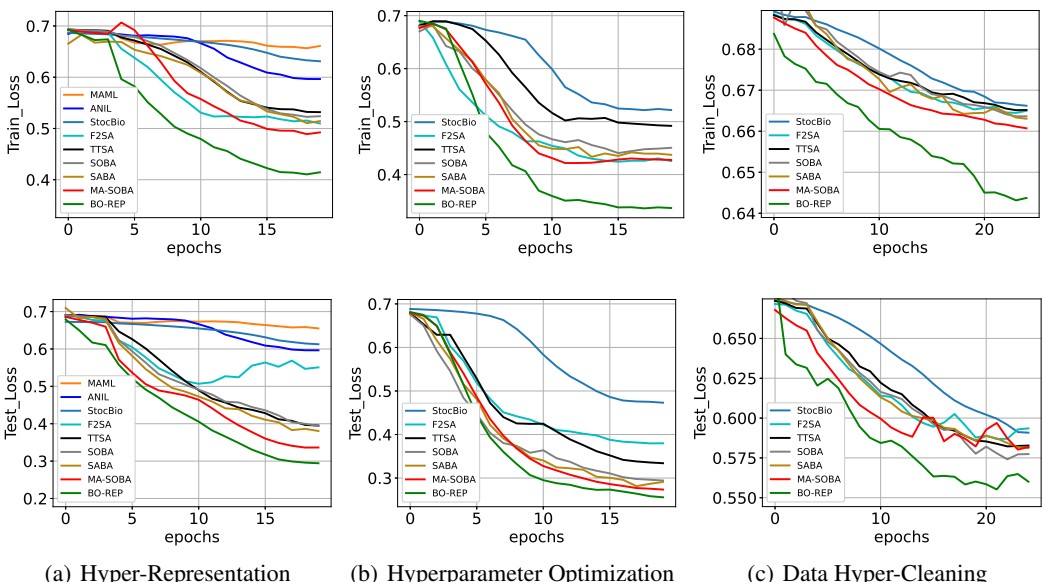

Figure 1: Comparison of various bilevel optimization algorithms on three applications: (a) results of Hyper-representation on Amazon Review Dataset. (b) results of hyperparameter optimization on Amazon Review Dataset. (c) results of data hyper-cleaning on Sentiment140 Dataset with noise rate $p = 0.3$.

2018; Shaban et al., 2019) task is to assign suitable weights $\lambda_i$ to each training sample such that the model trained on such weighted training set can achieve a good performance on the uncorrupted validation set $\mathcal{D}_{\text{val}}$. The hyper-cleaning problem can be formulated as follows:

$$\min_{\boldsymbol{\lambda}} \frac{1}{|\mathcal{D}_{\text{val}}|} \sum_{\xi \in \mathcal{D}_{\text{val}}} \mathcal{L}(\boldsymbol{w}^*(\boldsymbol{\lambda}); \xi), \text{ s.t. } \boldsymbol{w}^*(\boldsymbol{\lambda}) \in \arg\min_{\boldsymbol{w}} \frac{1}{|\mathcal{D}_{\text{tr}}|} \sum_{\zeta_i \in \mathcal{D}_{\text{tr}}} \sigma(\lambda_i)\mathcal{L}(\boldsymbol{w}; \zeta_i) + c\|\boldsymbol{w}\|^2, \quad (5)$$

where $\sigma(\cdot)$ is the sigmoid function, and $\mathcal{L}(\boldsymbol{w}; \zeta)$ is the lower level loss function induced by the model parameter $\boldsymbol{w}$ and corrupted sample $\zeta$, and $c > 0$ is a regularization parameter.

The hyper-cleaning experiments are conducted over the Sentiment140 dataset (Go et al., 2009) for text classification, where data samples consist of two types of emotions for Twitter messages. For each data sample in the training set, we replace its label with a random class number with probability $p$, meanwhile keeping the validation set intact. We compare our proposed BO-REP algorithm with other baselines StocBio (Ji et al., 2021), TTSA (Hong et al., 2023), F$^2$SA (Kwon et al., 2023a), SOBA (Dagréou et al., 2022), and MA-SOBA (Chen et al., 2023b). Figure 1(c) and Figure 2(c) (in Appendix) show the training and evaluation results with corruption rate $p = 0.3$, and Figure 3 (in the Appendix F.3) show the results with $p = 0.1$. BO-REP demonstrates a faster convergence rate and higher performance than other baselines on both noise settings, which is consistent with our theoretical results. We provide more experimental details and discussion in Appendix F.3.

## 5 CONCLUSION

In this paper, we design a new algorithm named BO-REP, to solve bilevel optimization problems where the upper-level problem has potential unbounded smoothness. The algorithm requires access to stochastic gradient or stochastic Hessian/Jacobian-vector product oracles in each iteration, and we have showed that BO-REP algorithm achieves $\widetilde{O}(1/\epsilon^4)$ oracle complexity to find an $\epsilon$-stationary point. It matches the state-of-the-art complexity results under the bounded smoothness setting and without mean-squared smoothness of the stochastic gradient, up to logarithmic factors. We have conducted experiments for various machine learning problems with bilevel formulations for text classification tasks, and our proposed algorithm shows superior performance over strong baselines. In the future, we plan to design more practical variants of this algorithm (e.g., single-loop and Hessian-free algorithms).

ACKNOWLEDGEMENTS

We would like to thank the anonymous reviewers for their helpful comments. This work has been supported by a grant from George Mason University, the Presidential Scholarship from George Mason University, a ORIEI seed funding from George Mason University, and a Cisco Faculty Research Award. The Computations were run on ARGO, a research computing cluster provided by the Office of Research Computing at George Mason University (URL: `https://orc.gmu.edu`).

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

| Symbol | Description |
|---|---|
| $L_{\boldsymbol{x},0}, L_{\boldsymbol{x},1}$ | Relaxed smoothness constants for $f$ with respect to $\boldsymbol{x}$ |
| $L_{\boldsymbol{y},0}, L_{\boldsymbol{y},1}$ | Relaxed smoothness constants for $f$ with respect to $\boldsymbol{y}$ |
| $K_0, K_1$ | Relaxed smoothness constants for function $\Phi$ |
| $M$ | Bound for $\|\nabla_{\boldsymbol{y}} f(\boldsymbol{x}, \boldsymbol{y}^*(\boldsymbol{x}))\|$ |
| $C_{g_{xy}}$ | Bound for $\|\nabla_{\boldsymbol{x}} \nabla_{\boldsymbol{y}} g(\boldsymbol{u})\|$ |
| $L$ | Lipschitz constant for $\nabla g(\boldsymbol{u})$ |
| $\mu$ | Strong-convexity constant for $g$ with respect to $\boldsymbol{y}$ |
| $\tau$ | Lipschitz constant for $\nabla_{\boldsymbol{x}} \nabla_{\boldsymbol{y}} g(\boldsymbol{u})$ |
| $\rho$ | Lipschitz constant for $\nabla_{\boldsymbol{y}}^2 g(\boldsymbol{u})$ |
| $\eta, \nu$ | Step-sizes for updating $\boldsymbol{x}, \boldsymbol{z}$ in BO-REP (i.e., Algorithm 1) |
| $\gamma$ | Step-size for updating $\boldsymbol{y}$ in UpdateLower (i.e., Algorithm 2) |
| $\beta$ | Momentum parameter for $\boldsymbol{m}$ in BO-REP (i.e., Algorithm 1) |
| $\mathcal{V}_0$ | Upper bound for $g(\boldsymbol{x}_0, \boldsymbol{y}_0^{0,0}) - g(\boldsymbol{x}_0, \boldsymbol{y}_0^*)$ in Epoch-SGD (i.e., Algorithm 3) |
| $\mathcal{B}_s$ | Projection ball for $s$-th epoch in Epoch-SGD (i.e., Algorithm 3) |
| $T_s$ | Number of iterations for $s$-th epoch in Epoch-SGD (i.e., Algorithm 3) |
| $\alpha_s$ | Step-size for $s$-th epoch in Epoch-SGD (i.e., Algorithm 3) |
| $k^\dagger$ | Number of epochs for Epoch-SGD (i.e., Algorithm 3) |
| $K$ | Number of iterations for updating $\boldsymbol{x}$ in BO-REP (i.e., Algorithm 1) |
| $N$ | Number of iterations for updating $\boldsymbol{y}$ in UpdateLower (i.e., Algorithm 2) |
| $I$ | Update period for $\boldsymbol{y}$ in UpdateLower (i.e., Algorithm 2) |
| $R$ | Radius of projection for UpdateLower (i.e., Algorithm 2) |

Table 2: Description of Notations

## A  RELATED WORK

**Bilevel Optimization** Bilevel optimization is used to model nested structure in the decision-making process (Bracken & McGill, 1973). Due to its broad applications in machine learning, there is a wave of studies on designing stochastic bilevel optimization algorithms for nonconvex smooth upper-level functions and strongly convex lower level functions. Ghadimi & Wang (2018) initiated the study of Bilevel Stochastic Approximation (BSA) method based on implicit gradient descent, and proved an $O(\epsilon^{-6})$ complexity to $\epsilon$-stationary point. The complexity result was later improved by a series of studies under the framework of automatic implicit differentiation (AID) (Hong et al., 2023; Chen et al., 2022; Ji et al., 2021; Khanduri et al., 2021; Chen et al., 2021; Dagréou et al., 2022; Guo et al., 2021; Yang et al., 2021; Chen et al., 2023b), which requires estimating Hessian inverse directly or approximating it by Hessian vector products. Another class of algorithms fall into the category of iterative differentiation (ITD) (Maclaurin et al., 2015; Franceschi et al., 2017; Finn et al., 2017; Franceschi et al., 2018; Shaban et al., 2019; Pedregosa, 2016; Li et al., 2022; Yang et al., 2021; Ji et al., 2021; Grazzi et al., 2023), which construct a computational graph of updating lower-level variables through gradient descent and compute the hypergradient via backpropagation. There are a few works which use fully first-order methods to solve bilevel optimization problems (Liu et al., 2022a; Kwon et al., 2023a). There is a line of work designing algorithms in the case where the lower-level problem is not strongly convex and have multiple minima (Sabach & Shtern, 2017; Sow et al., 2022; Liu et al., 2020; 2021a;b; 2022a; Shen & Chen, 2023; Kwon et al., 2023b; Chen et al., 2023a). However, all of these works need to assume the upper-level function is convex or has Lipschitz gradient and hence are not applicable to our bilevel optimization problem with an unbounded smooth upper-level function.

**Unbounded Smoothness** Zhang et al. (2020b) proposed the relaxed smooth condition and analyzed gradient clipping/normalization under this condition. This analysis was further improved by the works of (Zhang et al., 2020a; Jin et al., 2021). Crawshaw et al. (2022) considered a coordinate-wise relaxed smooth condition and proved the convergence for the generalized signSGD method. Faw et al. (2023); Wang et al. (2023) studied Adagrad-type algorithms for relaxed smooth functions. Li et al. (2023a); Wang et al. (2022) analyzed the convergence of Adam under relaxed smooth assumptions. Li et al. (2023b) analyzed gradient-based methods under a generalized smoothness condition. Chen et al. (2023c) proposed a new notion of $\alpha$-symmetric generalized smoothness and analyzed normalized gradient descent algorithms. Reisizadeh et al. (2023) considered variance-reduced gradient clipping when the function satisfies an averaged relaxed smooth condition. There is also a line of work focusing on federated optimization for unbounded smooth functions (Liu et al., 2022b; Crawshaw et al., 2023a;b). However all of these works only focus on single-level problems and cannot be applied to the bilevel optimization problem as considered in our paper.

# B  PROPERTIES OF ASSUMPTION 1

## B.1  DEFINITIONS OF RELAXED SMOOTHNESS

The standard relaxed smoothness assumption in Zhang et al. (2020b) is defined in Definition B.1.

**Definition B.1** ((Zhang et al., 2020b, Definition 1)). *A twice differentiable function $f$ is $(L_0, L_1)$-smooth if $\|\nabla^2 f(\boldsymbol{u})\| \le L_0 + L_1 \|\nabla f(\boldsymbol{u})\|$.*

Definition B.2 is an alternative definition for the $(L_0, L_1)$-smoothness. It is strictly weaker than Definition B.1 because it does not require the function $f$ to be twice differentiable.

**Definition B.2** ((Zhang et al., 2020a, Remark 2.3)). *A differentiable function $f$ is $(L_0, L_1)$-smooth if $\|\nabla f(\boldsymbol{u}) - \nabla f(\boldsymbol{u}')\| \le (L_0 + L_1 \|\nabla f(\boldsymbol{u})\|)\|\boldsymbol{u} - \boldsymbol{u}'\|$ for any $\|\boldsymbol{u} - \boldsymbol{u}'\| \le 1/L_1$.*

## B.2  RELATIONSHIP BETWEEN ASSUMPTION 1 AND STANDARD RELAXED SMOOTHNESS

The following lemma shows that our proposed $(L_{\boldsymbol{x},0}, L_{\boldsymbol{x},1}, L_{\boldsymbol{y},0}, L_{\boldsymbol{y},1})$ can recover the standard relaxed smoothness (e.g., Definition B.2) when the upper-level variable $\boldsymbol{x}$ and the lower-level variable $\boldsymbol{y}$ have the same smoothness constants.

**Lemma B.3.** *When $L_{\boldsymbol{x},0} = L_{\boldsymbol{y},0} = L_0/2$ and $L_{\boldsymbol{x},1} = L_{\boldsymbol{y},1} = L_1/2$, Assumption 1 implies that for any $\boldsymbol{u}, \boldsymbol{u}'$ such that $\|\boldsymbol{u} - \boldsymbol{u}'\| \le 1/L_1$, we have*

$$\|\nabla_{\boldsymbol{u}} f(\boldsymbol{u}) - \nabla_{\boldsymbol{u}} f(\boldsymbol{u}')\| \le (L_0 + L_1 \|\nabla_{\boldsymbol{u}} f(\boldsymbol{u})\|)\|\boldsymbol{u} - \boldsymbol{u}'\|. \tag{6}$$

*In other words, $(L_{\boldsymbol{x},0}, L_{\boldsymbol{x},1}, L_{\boldsymbol{y},0}, L_{\boldsymbol{y},1})$-smoothness assumption can recover the standard relaxed smoothness assumption.*

***Proof of Lemma B.3.*** When $L_{\boldsymbol{x},0} = L_{\boldsymbol{y},0} = L_0/2$ and $L_{\boldsymbol{x},1} = L_{\boldsymbol{y},1} = L_1/2$, by Assumption 1 we have for any $\boldsymbol{u}, \boldsymbol{u}'$,

$$\|\boldsymbol{u} - \boldsymbol{u}'\| \le \frac{1}{\sqrt{2\left(L_{\boldsymbol{x},1}^2 + L_{\boldsymbol{y},1}^2\right)}} = \frac{1}{L_1}.$$

Moreover, we have

$$\|\nabla_{\boldsymbol{u}} f(\boldsymbol{u}) - \nabla_{\boldsymbol{u}} f(\boldsymbol{u}')\| = \sqrt{\|\nabla_{\boldsymbol{x}} f(\boldsymbol{u}) - \nabla_{\boldsymbol{x}} f(\boldsymbol{u}')\|^2 + \|\nabla_{\boldsymbol{y}} f(\boldsymbol{u}) - \nabla_{\boldsymbol{y}} f(\boldsymbol{u}')\|^2}$$

$$\le \sqrt{\frac{1}{4}(L_0 + L_1 \|\nabla_{\boldsymbol{x}} f(\boldsymbol{u})\|)^2 \|\boldsymbol{u} - \boldsymbol{u}'\|^2 + \frac{1}{4}(L_0 + L_1 \|\nabla_{\boldsymbol{y}} f(\boldsymbol{u})\|)^2 \|\boldsymbol{u} - \boldsymbol{u}'\|^2}$$

$$\le \sqrt{(L_0^2 + L_1^2 \|\nabla_{\boldsymbol{u}} f(\boldsymbol{u})\|^2)\|\boldsymbol{u} - \boldsymbol{u}'\|^2}$$

$$\le (L_0 + L_1 \|\nabla_{\boldsymbol{u}} f(\boldsymbol{u})\|)\|\boldsymbol{u} - \boldsymbol{u}'\|,$$

which means that the function $f$ is $(L_0, L_1)$-smooth in terms of $\boldsymbol{u} = (\boldsymbol{x}, \boldsymbol{y})$ when $L_{\boldsymbol{x},0} = L_{\boldsymbol{y},0} = L_0/2$ and $L_{\boldsymbol{x},1} = L_{\boldsymbol{y},1} = L_1/2$. □

## C   TECHNICAL LEMMAS

In this section we provide some technical lemmas which are useful for our following proof. This section mainly provides useful properties of the considered bilevel problem under our assumptions.

**Lemma C.1.** *The hypergradient $\nabla\Phi(x)$ takes the forms of*

$$
\begin{aligned}
\nabla\Phi(x) &= \nabla_x f(x, y^*(x)) + \frac{\partial y^*(x)}{\partial x}\nabla_y f(x, y^*(x)) \\
&= \nabla_x f(x, y^*(x)) - \nabla_x\nabla_y g(x, y^*(x))[\nabla_y^2 g(x, y^*(x))]^{-1}\nabla_y f(x, y^*(x)) \\
&= \nabla_x f(x, y^*(x)) - \nabla_x\nabla_y g(x, y^*(x))z^*(x).
\end{aligned}
\tag{7}
$$

*where $z^*(x)$ is the solution of the linear system:*

$$
z^*(x) = [\nabla_y^2 g(x, y^*(x))]^{-1}\nabla_y f(x, y^*(x)).
\tag{8}
$$

***Proof of Lemma C.1.*** Using the chain rule over the hypergradient $\nabla\Phi(x) = \frac{\partial f(x, y^*(x))}{\partial x}$, we have

$$
\nabla\Phi(x) = \nabla_x f(x, y^*(x)) + \frac{\partial y^*(x)}{\partial x}\nabla_y f(x, y^*(x)).
\tag{9}
$$

By optimality condition of $y^*(x)$, we have $\nabla_y g(x, y^*(x)) = 0$. Then taking implicit differentiation with respect to $x$ yields

$$
\nabla_x\nabla_y g(x, y^*(x)) + \frac{\partial y^*(x)}{\partial x}\nabla_y^2 g(x, y^*(x)) = 0.
\tag{10}
$$

By Assumption 2, function $g(x, y)$ is $\mu$-strongly convex with respect to $y$, thus $[\nabla_y^2 g(x, y^*(x))]^{-1}$ is non-singular. Plugging (10) into (9) yields

$$
\nabla\Phi(x) = \nabla_x f(x, y^*(x)) - \nabla_x\nabla_y g(x, y^*(x))[\nabla_y^2 g(x, y^*(x))]^{-1}\nabla_y f(x, y^*(x)).
\tag{11}
$$

Also note that $z^*(x)$ takes the form

$$
z^*(x) = [\nabla_y^2 g(x, y^*(x))]^{-1}\nabla_y f(x, y^*(x)),
$$

hence hypergradient $\nabla\Phi(x)$ can also be represented as

$$
\nabla\Phi(x) = \nabla_x f(x, y^*(x)) - \nabla_x\nabla_y g(x, y^*(x))z^*(x).
$$

$\square$

**Lemma C.2.** *Suppose Assumption 1 and 2 hold. Then, we have*

(a) $y^*(x)$ *is $\frac{C_{g_{xy}}}{\mu}$-Lipschitz continous.*

(b) $z^*(x)$ *is $L_{z^*}$-Lipschitz continous, i.e.,*

$$
\|z^*(x) - z^*(x')\| \le L_{z^*}\|x - x'\|
$$

*if $\|x - x'\| \le \dfrac{1}{\sqrt{2\left(1 + \frac{C_{g_{xy}}^2}{\mu^2}\right)(L_{x,1}^2 + L_{y,1}^2)}}$, where $L_{z^*}$ is defined as*

$$
L_{z^*} := \sqrt{1 + \left(\frac{C_{g_{xy}}}{\mu}\right)^2\left(\frac{\rho M}{\mu^2} + \frac{1}{\mu}(L_{y,0} + L_{y,1}M)\right)}.
\tag{12}
$$

(c) $\nabla_x f(x, y^*(x))$ *and $\nabla\Phi(x)$ satisfy the following:*

$$
\|\nabla_x f(x, y^*(x))\| \le \|\nabla\Phi(x)\| + \frac{C_{g_{xy}}M}{\mu}.
\tag{13}
$$

***Proof of Lemma C.2.*** By (10) in Lemma C.1, we have

$$\frac{\partial \boldsymbol{y}^*(\boldsymbol{x})}{\partial \boldsymbol{x}} = -\nabla_{\boldsymbol{x}} \nabla_{\boldsymbol{y}} g(\boldsymbol{x}, \boldsymbol{y}^*(\boldsymbol{x})) [\nabla_{\boldsymbol{y}}^2 g(\boldsymbol{x}, \boldsymbol{y}^*(\boldsymbol{x}))]^{-1}.$$

Now we proceed to prove the lemma.

$(a)$. Note that by Assumption 2, for any $\boldsymbol{x}$ we have

$$\left\| \frac{\partial \boldsymbol{y}^*(\boldsymbol{x})}{\partial \boldsymbol{x}} \right\| = \left\| \nabla_{\boldsymbol{x}} \nabla_{\boldsymbol{y}} g(\boldsymbol{x}, \boldsymbol{y}^*(\boldsymbol{x})) [\nabla_{\boldsymbol{y}}^2 g(\boldsymbol{x}, \boldsymbol{y}^*(\boldsymbol{x}))]^{-1} \right\| \le \frac{C_{g_{xy}}}{\mu}.$$

Therefore, $\boldsymbol{y}^*(\boldsymbol{x})$ is $\frac{C_{g_{xy}}}{\mu}$-Lipschitz continuous.

$(b)$. Let $\boldsymbol{u} = (\boldsymbol{x}, \boldsymbol{y}^*(\boldsymbol{x}))$ and $\boldsymbol{u}' = (\boldsymbol{x}', \boldsymbol{y}^*(\boldsymbol{x}'))$, by condition (14) we have

$$\begin{aligned}
\|\boldsymbol{u} - \boldsymbol{u}'\| &= \sqrt{\|\boldsymbol{x} - \boldsymbol{x}'\|^2 + \|\boldsymbol{y}^*(\boldsymbol{x}) - \boldsymbol{y}^*(\boldsymbol{x}')\|^2} \\
&\overset{(i)}{\le} \sqrt{1 + \left(\frac{C_{g_{xy}}}{\mu}\right)^2} \|\boldsymbol{x} - \boldsymbol{x}'\| \le \frac{1}{\sqrt{2(L_{\boldsymbol{x},1}^2 + L_{\boldsymbol{y},1}^2)}},
\end{aligned}$$

where $(i)$ follows from $(a)$ in Lemma C.2 that $\boldsymbol{y}^*(\boldsymbol{x})$ is Lipschitz. Hence the condition for applying Assumption 1 is satisfied. By definition (8) of $\boldsymbol{z}^*(\boldsymbol{x})$, for any $\boldsymbol{x}, \boldsymbol{x}'$ we have

$$\begin{aligned}
& \|\boldsymbol{z}^*(\boldsymbol{x}) - \boldsymbol{z}^*(\boldsymbol{x}')\| \\
={}& \left\| [\nabla_{\boldsymbol{y}}^2 g(\boldsymbol{x}, \boldsymbol{y}^*(\boldsymbol{x}))]^{-1} \nabla_{\boldsymbol{y}} f(\boldsymbol{x}, \boldsymbol{y}^*(\boldsymbol{x})) - [\nabla_{\boldsymbol{y}}^2 g(\boldsymbol{x}', \boldsymbol{y}^*(\boldsymbol{x}'))]^{-1} \nabla_{\boldsymbol{y}} f(\boldsymbol{x}', \boldsymbol{y}^*(\boldsymbol{x}')) \right\| \\
\le{}& \left\| [\nabla_{\boldsymbol{y}}^2 g(\boldsymbol{x}, \boldsymbol{y}^*(\boldsymbol{x}))]^{-1} \nabla_{\boldsymbol{y}} f(\boldsymbol{x}, \boldsymbol{y}^*(\boldsymbol{x})) - [\nabla_{\boldsymbol{y}}^2 g(\boldsymbol{x}', \boldsymbol{y}^*(\boldsymbol{x}'))]^{-1} \nabla_{\boldsymbol{y}} f(\boldsymbol{x}, \boldsymbol{y}^*(\boldsymbol{x})) \right\| \\
& + \left\| [\nabla_{\boldsymbol{y}}^2 g(\boldsymbol{x}', \boldsymbol{y}^*(\boldsymbol{x}'))]^{-1} \nabla_{\boldsymbol{y}} f(\boldsymbol{x}, \boldsymbol{y}^*(\boldsymbol{x})) - [\nabla_{\boldsymbol{y}}^2 g(\boldsymbol{x}', \boldsymbol{y}^*(\boldsymbol{x}'))]^{-1} \nabla_{\boldsymbol{y}} f(\boldsymbol{x}', \boldsymbol{y}^*(\boldsymbol{x}')) \right\| \\
\le{}& M \left\| [\nabla_{\boldsymbol{y}}^2 g(\boldsymbol{x}, \boldsymbol{y}^*(\boldsymbol{x}))]^{-1} - [\nabla_{\boldsymbol{y}}^2 g(\boldsymbol{x}', \boldsymbol{y}^*(\boldsymbol{x}'))]^{-1} \right\| + \frac{1}{\mu} \left\| \nabla_{\boldsymbol{y}} f(\boldsymbol{x}, \boldsymbol{y}^*(\boldsymbol{x})) - \nabla_{\boldsymbol{y}} f(\boldsymbol{x}', \boldsymbol{y}^*(\boldsymbol{x}')) \right\| \\
\overset{(i)}{\le}{}& M \left\| [\nabla_{\boldsymbol{y}}^2 g(\boldsymbol{x}, \boldsymbol{y}^*(\boldsymbol{x}))]^{-1} \right\| \left\| [\nabla_{\boldsymbol{y}}^2 g(\boldsymbol{x}', \boldsymbol{y}^*(\boldsymbol{x}'))]^{-1} \right\| \left\| \nabla_{\boldsymbol{y}}^2 g(\boldsymbol{x}, \boldsymbol{y}^*(\boldsymbol{x})) - \nabla_{\boldsymbol{y}}^2 g(\boldsymbol{x}', \boldsymbol{y}^*(\boldsymbol{x}')) \right\| \\
& + \frac{1}{\mu} \left( L_{\boldsymbol{y},0} + L_{\boldsymbol{y},1} \|\nabla_{\boldsymbol{y}} f(\boldsymbol{x}, \boldsymbol{y}^*(\boldsymbol{x}))\| \right) \sqrt{\|\boldsymbol{x} - \boldsymbol{x}'\|^2 + \|\boldsymbol{y}^*(\boldsymbol{x}) - \boldsymbol{y}^*(\boldsymbol{x}')\|^2} \\
\overset{(ii)}{\le}{}& \frac{\rho M}{\mu^2} \sqrt{\|\boldsymbol{x} - \boldsymbol{x}'\|^2 + \|\boldsymbol{y}^*(\boldsymbol{x}) - \boldsymbol{y}^*(\boldsymbol{x}')\|^2} + \frac{1}{\mu} \left( L_{\boldsymbol{y},0} + L_{\boldsymbol{y},1} M \right) \sqrt{\|\boldsymbol{x} - \boldsymbol{x}'\|^2 + \|\boldsymbol{y}^*(\boldsymbol{x}) - \boldsymbol{y}^*(\boldsymbol{x}')\|^2} \\
={}& \left( \frac{\rho M}{\mu^2} + \frac{1}{\mu} \left( L_{\boldsymbol{y},0} + L_{\boldsymbol{y},1} M \right) \right) \sqrt{\|\boldsymbol{x} - \boldsymbol{x}'\|^2 + \|\boldsymbol{y}^*(\boldsymbol{x}) - \boldsymbol{y}^*(\boldsymbol{x}')\|^2} \\
\overset{(iii)}{\le}{}& \left( \frac{\rho M}{\mu^2} + \frac{1}{\mu} \left( L_{\boldsymbol{y},0} + L_{\boldsymbol{y},1} M \right) \right) \sqrt{\|\boldsymbol{x} - \boldsymbol{x}'\|^2 + \left(\frac{C_{g_{xy}}}{\mu}\right)^2 \|\boldsymbol{x} - \boldsymbol{x}'\|^2} \\
={}& \sqrt{1 + \left(\frac{C_{g_{xy}}}{\mu}\right)^2} \left( \frac{\rho M}{\mu^2} + \frac{1}{\mu} \left( L_{\boldsymbol{y},0} + L_{\boldsymbol{y},1} M \right) \right) \|\boldsymbol{x} - \boldsymbol{x}'\|
\end{aligned}$$

where $(i)$ follows from Assumption 1 and the fact that

$$\|H_1^{-1} - H_2^{-1}\| = \|H_1^{-1}(H_1 - H_2)H_2^{-1}\| \le \|H_1^{-1}\| \|H_2^{-1}\| \|H_1 - H_2\|,$$

$(ii)$ follows from Assumption 2 and $(iii)$ uses the fact that $\boldsymbol{y}^*(\boldsymbol{x})$ is $\frac{C_{g_{xy}}}{\mu}$-Lipschitz continuous. For notation convenience, we define

$$L_{\boldsymbol{z}^*} := \sqrt{1 + \left(\frac{C_{g_{xy}}}{\mu}\right)^2} \left( \frac{\rho M}{\mu^2} + \frac{1}{\mu} \left( L_{\boldsymbol{y},0} + L_{\boldsymbol{y},1} M \right) \right)$$

to be the Lipschitz constant. Therefore, $\boldsymbol{z}^*(\boldsymbol{x})$ is $L_{\boldsymbol{z}^*}$-Lipschitz continuous.

$(c)$. By Lemma C.1, we have

$$
\begin{aligned}
\|\nabla_{\boldsymbol{x}} f(\boldsymbol{x}, \boldsymbol{y}^*(\boldsymbol{x})) - \nabla \Phi(\boldsymbol{x})\| &= \left\| \frac{\partial \boldsymbol{y}^*(\boldsymbol{x})}{\partial \boldsymbol{x}} \nabla_{\boldsymbol{y}} f(\boldsymbol{x}, \boldsymbol{y}^*(\boldsymbol{x})) \right\| \\
&= \left\| \nabla_{\boldsymbol{x}} \nabla_{\boldsymbol{y}} g(\boldsymbol{x}, \boldsymbol{y}^*(\boldsymbol{x})) [\nabla_{\boldsymbol{y}}^2 g(\boldsymbol{x}, \boldsymbol{y}^*(\boldsymbol{x}))]^{-1} \nabla_{\boldsymbol{y}} f(\boldsymbol{x}, \boldsymbol{y}^*(\boldsymbol{x})) \right\| \\
&\overset{(i)}{\leq} \frac{C_{g_{xy}} M}{\mu},
\end{aligned}
$$

where $(i)$ follows from Assumption 2. Therefore, we have

$$
\|\nabla_{\boldsymbol{x}} f(\boldsymbol{x}_k, \boldsymbol{y}_k^*)\| \leq \|\nabla \Phi(\boldsymbol{x}_k)\| + \frac{C_{g_{xy}} M}{\mu}.
$$

$\square$

Under the Assumption 1 and 2, we can show in the following lemma that the function $\Phi(\boldsymbol{x})$ satisfies standard relaxed smoothness condition: $\|\nabla \Phi(\boldsymbol{x}) - \nabla \Phi(\boldsymbol{x}')\| \leq (K_0 + K_1 \|\nabla \Phi(\boldsymbol{x}')\|) \|\boldsymbol{x} - \boldsymbol{x}'\|$ with some $K_0$ and $K_1$ if $\boldsymbol{x}$ and $\boldsymbol{x}'$ are not far away from each other. This is very important because it allows us to apply the descent lemma in the standard relaxed smoothness setting for analyzing the dynamics of our algorithm.

**Lemma C.3.** *Suppose Assumption 1 and 2 hold. Then for any $\boldsymbol{x}, \boldsymbol{x}'$ such that*

$$
\|\boldsymbol{x} - \boldsymbol{x}'\| \leq \frac{1}{\sqrt{2 \left( 1 + \frac{C_{g_{xy}}^2}{\mu^2} \right) (L_{\boldsymbol{x},1}^2 + L_{\boldsymbol{y},1}^2)}}, \tag{14}
$$

*we have*

$$
\|\nabla \Phi(\boldsymbol{x}) - \nabla \Phi(\boldsymbol{x}')\| \leq (K_0 + K_1 \|\nabla \Phi(\boldsymbol{x}')\|) \|\boldsymbol{x} - \boldsymbol{x}'\|, \tag{15}
$$

*where $K_0, K_1$ are defined as*

$$
\begin{aligned}
K_0 &= \sqrt{1 + \left( \frac{C_{g_{xy}}}{\mu} \right)^2} \left( L_{\boldsymbol{x},0} + L_{\boldsymbol{x},1} \frac{C_{g_{xy}} M}{\mu} + \frac{C_{g_{xy}}}{\mu} (L_{\boldsymbol{y},0} + L_{\boldsymbol{y},1} M) + M \frac{C_{g_{xy}} \rho + \tau \mu}{\mu^2} \right), \\
K_1 &= \sqrt{1 + \left( \frac{C_{g_{xy}}}{\mu} \right)^2} L_{\boldsymbol{x},1}.
\end{aligned}
\tag{16}
$$

***Proof of Lemma C.3.*** Let $\boldsymbol{u} = (\boldsymbol{x}, \boldsymbol{y}^*(\boldsymbol{x}))$ and $\boldsymbol{u}' = (\boldsymbol{x}', \boldsymbol{y}^*(\boldsymbol{x}'))$, by condition (14) we have

$$
\begin{aligned}
\|\boldsymbol{u} - \boldsymbol{u}'\| &= \sqrt{\|\boldsymbol{x} - \boldsymbol{x}'\|^2 + \|\boldsymbol{y}^*(\boldsymbol{x}) - \boldsymbol{y}^*(\boldsymbol{x}')\|^2} \\
&\overset{(i)}{\leq} \sqrt{1 + \left( \frac{C_{g_{xy}}}{\mu} \right)^2} \|\boldsymbol{x} - \boldsymbol{x}'\| \leq \frac{1}{\sqrt{2(L_{\boldsymbol{x},1}^2 + L_{\boldsymbol{y},1}^2)}},
\end{aligned}
$$

where $(i)$ follows from $(a)$ in Lemma C.2 that $\boldsymbol{y}^*(\boldsymbol{x})$ is Lipschitz. Hence the condition for applying Assumption 1 is satisfied. Then we have

$$
\begin{aligned}
&\|\nabla\Phi(\boldsymbol{x}) - \nabla\Phi(\boldsymbol{x}')\| \\
&= \left\|\nabla_{\boldsymbol{x}}f(\boldsymbol{x},\boldsymbol{y}^*(\boldsymbol{x})) + \frac{\partial \boldsymbol{y}^*(\boldsymbol{x})}{\partial \boldsymbol{x}}\nabla_{\boldsymbol{y}}f(\boldsymbol{x},\boldsymbol{y}^*(\boldsymbol{x})) - \nabla_{\boldsymbol{x}}f(\boldsymbol{x}',\boldsymbol{y}^*(\boldsymbol{x}')) - \frac{\partial \boldsymbol{y}^*(\boldsymbol{x}')}{\partial \boldsymbol{x}'}\nabla_{\boldsymbol{y}}f(\boldsymbol{x}',\boldsymbol{y}^*(\boldsymbol{x}'))\right\| \\
&\leq \|\nabla_{\boldsymbol{x}}f(\boldsymbol{x},\boldsymbol{y}^*(\boldsymbol{x})) - \nabla_{\boldsymbol{x}}f(\boldsymbol{x}',\boldsymbol{y}^*(\boldsymbol{x}'))\| + \left\|\frac{\partial \boldsymbol{y}^*(\boldsymbol{x})}{\partial \boldsymbol{x}}\nabla_{\boldsymbol{y}}f(\boldsymbol{x},\boldsymbol{y}^*(\boldsymbol{x})) - \frac{\partial \boldsymbol{y}^*(\boldsymbol{x}')}{\partial \boldsymbol{x}'}\nabla_{\boldsymbol{y}}f(\boldsymbol{x}',\boldsymbol{y}^*(\boldsymbol{x}'))\right\| \\
&\leq \|\nabla_{\boldsymbol{x}}f(\boldsymbol{x},\boldsymbol{y}^*(\boldsymbol{x})) - \nabla_{\boldsymbol{x}}f(\boldsymbol{x}',\boldsymbol{y}^*(\boldsymbol{x}'))\| + \left\|\frac{\partial \boldsymbol{y}^*(\boldsymbol{x})}{\partial \boldsymbol{x}}(\nabla_{\boldsymbol{y}}f(\boldsymbol{x},\boldsymbol{y}^*(\boldsymbol{x})) - \nabla_{\boldsymbol{y}}f(\boldsymbol{x}',\boldsymbol{y}^*(\boldsymbol{x}')))\right\| \\
&\quad + \left\|\left(\frac{\partial \boldsymbol{y}^*(\boldsymbol{x})}{\partial \boldsymbol{x}} - \frac{\partial \boldsymbol{y}^*(\boldsymbol{x}')}{\partial \boldsymbol{x}'}\right)\nabla_{\boldsymbol{y}}f(\boldsymbol{x}',\boldsymbol{y}^*(\boldsymbol{x}'))\right\| \\
&\overset{(i)}{\leq} \|\nabla_{\boldsymbol{x}}f(\boldsymbol{x},\boldsymbol{y}^*(\boldsymbol{x})) - \nabla_{\boldsymbol{x}}f(\boldsymbol{x}',\boldsymbol{y}^*(\boldsymbol{x}'))\| + \frac{C_{g_{xy}}}{\mu}\|\nabla_{\boldsymbol{y}}f(\boldsymbol{x},\boldsymbol{y}^*(\boldsymbol{x})) - \nabla_{\boldsymbol{y}}f(\boldsymbol{x}',\boldsymbol{y}^*(\boldsymbol{x}'))\| \\
&\quad + M\left\|\nabla_{\boldsymbol{x}}\nabla_{\boldsymbol{y}}g(\boldsymbol{x},\boldsymbol{y}^*(\boldsymbol{x}))[\nabla_{\boldsymbol{y}}^2 g(\boldsymbol{x},\boldsymbol{y}^*(\boldsymbol{x}))]^{-1} - \nabla_{\boldsymbol{x}}\nabla_{\boldsymbol{y}}g(\boldsymbol{x}',\boldsymbol{y}^*(\boldsymbol{x}'))[\nabla_{\boldsymbol{y}}^2 g(\boldsymbol{x}',\boldsymbol{y}^*(\boldsymbol{x}'))]^{-1}\right\| \\
&\leq \|\nabla_{\boldsymbol{x}}f(\boldsymbol{x},\boldsymbol{y}^*(\boldsymbol{x})) - \nabla_{\boldsymbol{x}}f(\boldsymbol{x}',\boldsymbol{y}^*(\boldsymbol{x}'))\| + \frac{C_{g_{xy}}}{\mu}\|\nabla_{\boldsymbol{y}}f(\boldsymbol{x},\boldsymbol{y}^*(\boldsymbol{x})) - \nabla_{\boldsymbol{y}}f(\boldsymbol{x}',\boldsymbol{y}^*(\boldsymbol{x}'))\| \\
&\quad + M\left\|\nabla_{\boldsymbol{x}}\nabla_{\boldsymbol{y}}g(\boldsymbol{x},\boldsymbol{y}^*(\boldsymbol{x}))[\nabla_{\boldsymbol{y}}^2 g(\boldsymbol{x},\boldsymbol{y}^*(\boldsymbol{x}))]^{-1} - \nabla_{\boldsymbol{x}}\nabla_{\boldsymbol{y}}g(\boldsymbol{x}',\boldsymbol{y}^*(\boldsymbol{x}'))[\nabla_{\boldsymbol{y}}^2 g(\boldsymbol{x},\boldsymbol{y}^*(\boldsymbol{x}))]^{-1}\right\| \\
&\quad + M\left\|\nabla_{\boldsymbol{x}}\nabla_{\boldsymbol{y}}g(\boldsymbol{x}',\boldsymbol{y}^*(\boldsymbol{x}'))[\nabla_{\boldsymbol{y}}^2 g(\boldsymbol{x},\boldsymbol{y}^*(\boldsymbol{x}))]^{-1} - \nabla_{\boldsymbol{x}}\nabla_{\boldsymbol{y}}g(\boldsymbol{x}',\boldsymbol{y}^*(\boldsymbol{x}'))[\nabla_{\boldsymbol{y}}^2 g(\boldsymbol{x}',\boldsymbol{y}^*(\boldsymbol{x}'))]^{-1}\right\| \\
&\overset{(ii)}{\leq} (L_{\boldsymbol{x},0} + L_{\boldsymbol{x},1}\|\nabla_{\boldsymbol{x}}f(\boldsymbol{x}',\boldsymbol{y}^*(\boldsymbol{x}'))\|)\sqrt{\|\boldsymbol{x}-\boldsymbol{x}'\|^2 + \|\boldsymbol{y}^*(\boldsymbol{x})-\boldsymbol{y}^*(\boldsymbol{x}')\|^2} \\
&\quad + \frac{C_{g_{xy}}}{\mu}(L_{\boldsymbol{y},0} + L_{\boldsymbol{y},1}\|\nabla_{\boldsymbol{y}}f(\boldsymbol{x},\boldsymbol{y}^*(\boldsymbol{x}))\|)\sqrt{\|\boldsymbol{x}-\boldsymbol{x}'\|^2 + \|\boldsymbol{y}^*(\boldsymbol{x})-\boldsymbol{y}^*(\boldsymbol{x}')\|^2} \\
&\quad + \frac{\tau M}{\mu}\sqrt{\|\boldsymbol{x}-\boldsymbol{x}'\|^2 + \|\boldsymbol{y}^*(\boldsymbol{x})-\boldsymbol{y}^*(\boldsymbol{x}')\|^2} \\
&\quad + MC_{g_{xy}}\left\|[\nabla_{\boldsymbol{y}}^2 g(\boldsymbol{x},\boldsymbol{y}^*(\boldsymbol{x}))]^{-1}\right\|\left\|[\nabla_{\boldsymbol{y}}^2 g(\boldsymbol{x}',\boldsymbol{y}^*(\boldsymbol{x}'))]^{-1}\right\|\left\|\nabla_{\boldsymbol{y}}^2 g(\boldsymbol{x},\boldsymbol{y}^*(\boldsymbol{x})) - \nabla_{\boldsymbol{y}}^2 g(\boldsymbol{x}',\boldsymbol{y}^*(\boldsymbol{x}'))\right\| \\
&\leq (L_{\boldsymbol{x},0} + L_{\boldsymbol{x},1}\|\nabla_{\boldsymbol{x}}f(\boldsymbol{x}',\boldsymbol{y}^*(\boldsymbol{x}'))\|)\sqrt{\|\boldsymbol{x}-\boldsymbol{x}'\|^2 + \|\boldsymbol{y}^*(\boldsymbol{x})-\boldsymbol{y}^*(\boldsymbol{x}')\|^2} \\
&\quad + \left(\frac{C_{g_{xy}}}{\mu}(L_{\boldsymbol{y},0} + L_{\boldsymbol{y},1}M) + M\frac{C_{g_{xy}}\rho + \tau\mu}{\mu^2}\right)\sqrt{\|\boldsymbol{x}-\boldsymbol{x}'\|^2 + \|\boldsymbol{y}^*(\boldsymbol{x})-\boldsymbol{y}^*(\boldsymbol{x}')\|^2} \\
&\overset{(iii)}{\leq} \left(\frac{C_{g_{xy}}}{\mu}(L_{\boldsymbol{y},0} + L_{\boldsymbol{y},1}M) + M\frac{C_{g_{xy}}\rho + \tau\mu}{\mu^2} + L_{\boldsymbol{x},0} + L_{\boldsymbol{x},1}\left(\frac{C_{g_{xy}}M}{\mu} + \|\nabla\Phi(\boldsymbol{x}')\|\right)\right)\sqrt{1 + \left(\frac{C_{g_{xy}}}{\mu}\right)^2}\|\boldsymbol{x}-\boldsymbol{x}'\| \\
&= \sqrt{1 + \left(\frac{C_{g_{xy}}}{\mu}\right)^2}\left(L_{\boldsymbol{x},0} + L_{\boldsymbol{x},1}\frac{C_{g_{xy}}M}{\mu} + \frac{C_{g_{xy}}}{\mu}(L_{\boldsymbol{y},0} + L_{\boldsymbol{y},1}M) + M\frac{C_{g_{xy}}\rho + \tau\mu}{\mu^2} + L_{\boldsymbol{x},1}\|\nabla\Phi(\boldsymbol{x}')\|\right)\|\boldsymbol{x}-\boldsymbol{x}'\|.
\end{aligned}
$$

where $(i)$ follows from (10), $(ii)$ follows from Assumption 1, 2 and the fact that

$$
\|H_1^{-1} - H_2^{-1}\| = \|H_1^{-1}(H_1 - H_2)H_2^{-1}\| \leq \|H_1^{-1}\|\|H_2^{-1}\|\|H_1 - H_2\|,
$$

and $(iii)$ follows from $(a)$ in Lemma C.2 that $\boldsymbol{y}^*(\boldsymbol{x})$ is Lipschitz. Recall the definition of $K_0, K_1$ in (16), therefore, the objective function $\Phi(\boldsymbol{x})$ is $(K_0, K_1)$-smooth, i.e.,

$$
\|\nabla\Phi(\boldsymbol{x}) - \nabla\Phi(\boldsymbol{x}')\| \leq (K_0 + K_1\|\nabla\Phi(\boldsymbol{x}')\|)\|\boldsymbol{x}-\boldsymbol{x}'\|.
$$

$\square$

We now present a descent inequality for $(K_0, K_1)$-smooth functions which will be used in our subsequent analysis.

**Lemma C.4** (Descent Inequality). *Let $\Phi$ be $(K_0, K_1)$-smooth. Then for any $\boldsymbol{x}, \boldsymbol{x}' \in \mathbb{R}^d$ such that*

$$
\|\boldsymbol{x}-\boldsymbol{x}'\| \leq \frac{1}{\sqrt{2\left(1 + \frac{C_{g_{xy}}^2}{\mu^2}\right)(L_{\boldsymbol{x},1}^2 + L_{\boldsymbol{y},1}^2)}},
$$

*we have*

$$\Phi(\boldsymbol{x}) \le \Phi(\boldsymbol{x}') + \langle \nabla\Phi(\boldsymbol{x}'), \boldsymbol{x} - \boldsymbol{x}' \rangle + \frac{K_0 + K_1 \|\nabla\Phi(\boldsymbol{x}')\|}{2} \|\boldsymbol{x} - \boldsymbol{x}'\|^2,$$

*where constants $K_0$ and $K_1$ are defined in (16).*

**Proof of Lemma C.4.** By Lemma C.3, for $\boldsymbol{x}, \boldsymbol{x}'$ such that

$$\|\boldsymbol{x} - \boldsymbol{x}'\| \le \frac{1}{\sqrt{2\left(1 + \frac{C_{g_{xy}}^2}{\mu^2}\right)\left(L_{\boldsymbol{x},1}^2 + L_{\boldsymbol{y},1}^2\right)}},$$

we have

$$\|\nabla\Phi(\boldsymbol{x}) - \nabla\Phi(\boldsymbol{x}')\| \le \left(K_0 + K_1\|\nabla\Phi(\boldsymbol{x}')\|\right)\|\boldsymbol{x} - \boldsymbol{x}'\|.$$

By definition above we have

$$
\begin{aligned}
\Phi(\boldsymbol{x}) - \Phi(\boldsymbol{x}') - \langle \nabla\Phi(\boldsymbol{x}'), \boldsymbol{x} - \boldsymbol{x}' \rangle &\le \int_0^1 \langle \nabla\Phi(\theta\boldsymbol{x} + (1-\theta)\boldsymbol{x}') - \nabla\Phi(\boldsymbol{x}'), \boldsymbol{x} - \boldsymbol{x}' \rangle \, \mathrm{d}\theta \\
&\le \int_0^1 \|\nabla\Phi(\theta\boldsymbol{x} + (1-\theta)\boldsymbol{x}') - \nabla\Phi(\boldsymbol{x}')\| \|\boldsymbol{x} - \boldsymbol{x}'\| \mathrm{d}\theta \\
&\le \int_0^1 \left(K_0 + K_1\|\nabla\Phi(\boldsymbol{x}')\|\right)\|\theta(\boldsymbol{x} - \boldsymbol{x}')\|\|\boldsymbol{x} - \boldsymbol{x}'\| \mathrm{d}\theta \\
&\le \frac{K_0 + K_1\|\nabla\Phi(\boldsymbol{x}')\|}{2}\|\boldsymbol{x} - \boldsymbol{x}'\|^2.
\end{aligned}
$$

Thus we conclude our proof by rearranging. $\qquad\square$

Next, we introduce a simple algebraic lemma.

**Lemma C.5** ((Zhang et al., 2020a, Lemma B.1)). *Let $\omega > 0$, and $\boldsymbol{u}, \boldsymbol{v} \in \mathbb{R}^d$, then*

$$-\frac{\langle \boldsymbol{u}, \boldsymbol{v} \rangle}{\|\boldsymbol{v}\|} \le -\omega\|\boldsymbol{u}\| - (1-\omega)\|\boldsymbol{v}\| + (1+\omega)\|\boldsymbol{v} - \boldsymbol{u}\|. \tag{17}$$

# D  PROOF OF LEMMAS IN SECTION 3.3

## D.1  FILTRATIONS AND NOTATIONS

For convenience, we will restate a few concepts here. Let $\mathcal{F}_k$ denote the filtration of the random variables for updating $\boldsymbol{z}_k$, $\boldsymbol{m}_k$ and $\boldsymbol{x}_k$ before iteration $k$, i.e.,

$$\mathcal{F}_k \coloneqq \sigma\left\{\xi_0, \dots, \xi_{k-1}, \zeta_0, \dots, \zeta_{k-1}\right\}$$

for $k \ge 1$, where $\sigma\{\cdot\}$ denotes the $\sigma$-algebra generated by the random variables. Let $\widetilde{\mathcal{F}}_0^{s,t}$ denote the filtration of the random variables for updating lower-level variable $\boldsymbol{y}_0$ starting at the $s$-th epoch before iteration $t$, i.e.,

$$\widetilde{\mathcal{F}}_0^{s,t} \coloneqq \sigma\left\{\tilde{\xi}_0^{s,0}, \dots, \tilde{\xi}_0^{s,t-1}\right\}$$

for $1 \le t \le T_s$ and $0 \le s \le k^\dagger - 1$. Let $\widetilde{\mathcal{F}}_k^t$ denote the filtration of the random variables for updating lower-level variable $\boldsymbol{y}_k$ $(k \ge 1)$ before iteration $t$, i.e.,

$$\widetilde{\mathcal{F}}_k^t \coloneqq \sigma\left\{\tilde{\xi}_k^0, \dots, \tilde{\xi}_k^{t-1}\right\}$$

for $1 \le t \le N$ and $k = jI$, where $1 \le j \le \lfloor \frac{K}{I} \rfloor$ and $I$ denotes the update period for $\boldsymbol{y}_k$. Let $\widetilde{\mathcal{F}}_K$ denote the filtration of all random variables for updating lower-level variable $\boldsymbol{y}_k$ $(k \ge 0)$, i.e.,

$$\widetilde{\mathcal{F}}_K \coloneqq \sigma\left\{\left(\bigcup_{s=0}^{k^\dagger-1} \widetilde{\mathcal{F}}_0^{s,T_s}\right) \bigcup \left(\bigcup_{k=1}^{K-1} \widetilde{\mathcal{F}}_k^N\right)\right\}.$$

## D.2 PROOF OF LEMMA 3.2

In this section we first present one technical lemma which provides high probability bound for SGD. We follow the same technique and procedure as Theorem 1 in Lan (2012), just simplify the modified mirror descent SA algorithm to SGD. For completeness of proof and consistency of notations in our paper, we paraphrase Theorem 1 in Lan (2012) as the following lemma.

**Lemma D.1.** *Consider the $s$-th epoch in Algorithm 3, let $D^2$ be an upper bound for $\frac{1}{2}\left\|\boldsymbol{y}_0^{s,0} - \boldsymbol{y}_0^*\right\|^2$, and assume that the fixed step-size $\alpha_s$ satisfies $0 < \alpha_s \leq \frac{1}{2L}$. Apply $T_s$ iterations of the update starting from $\boldsymbol{y}_0^{s,0}$,*

$$\boldsymbol{y}_0^{s,t+1} = \underset{\boldsymbol{v} \in \mathcal{B}\left(\boldsymbol{y}_0^{s,0}, \sqrt{2}D\right)}{\arg\min} \frac{1}{2}\left\|\boldsymbol{v} - \left(\boldsymbol{y}_0^{s,t} - \alpha_s \nabla_{\boldsymbol{y}} G(\boldsymbol{x}_0, \boldsymbol{y}_0^{s,t}; \tilde{\xi}_0^{s,t})\right)\right\|^2, \tag{18}$$

*where the stochastic gradient estimators $\nabla_{\boldsymbol{y}} G(\boldsymbol{x}_0, \boldsymbol{y}_0^{s,t}; \tilde{\xi}_0^{s,t})$ satisfy Assumption 3. Then for any $\lambda > 0$ and $T_s > 0$, we have*

$$Pr\left[g(\boldsymbol{x}_0, \boldsymbol{y}_0^{s+1,0}) - g(\boldsymbol{x}_0, \boldsymbol{y}_0^*) > \mathcal{P}_0(T_s) + \lambda \mathcal{P}_1(T_s)\right] \leq \exp(-\lambda^2/3) + \exp(-\lambda), \tag{19}$$

*where*

$$\begin{aligned}
\mathcal{P}_0(T_s) &:= \frac{1}{\alpha_s T_s}\left[D^2 + 2\sigma_{g,1}^2 \alpha_s^2 T_s\right], \\
\mathcal{P}_1(T_s) &:= \frac{1}{\alpha_s T_s}\left[2\sqrt{2}D\sigma_{g,1}\alpha_s\sqrt{T_s} + 2\sigma_{g,1}^2\alpha_s^2 T_s\right].
\end{aligned} \tag{20}$$

*Proof of Lemma D.1.* First, we have

$$\begin{aligned}
&\alpha_s g(\boldsymbol{x}_0, \boldsymbol{y}_0^{s,t+1}) \\
&\leq \alpha_s\left[g(\boldsymbol{x}_0, \boldsymbol{y}_0^{s,t}) + \left\langle \nabla_{\boldsymbol{y}} g(\boldsymbol{x}_0, \boldsymbol{y}_0^{s,t}), \boldsymbol{y}_0^{s,t+1} - \boldsymbol{y}_0^{s,t}\right\rangle + \frac{L}{2}\left\|\boldsymbol{y}_0^{s,t+1} - \boldsymbol{y}_0^{s,t}\right\|^2\right] \\
&\leq \alpha_s\left[g(\boldsymbol{x}_0, \boldsymbol{y}_0^{s,t}) + \left\langle \nabla_{\boldsymbol{y}} g(\boldsymbol{x}_0, \boldsymbol{y}_0^{s,t}), \boldsymbol{y}_0^{s,t+1} - \boldsymbol{y}_0^{s,t}\right\rangle\right] + \frac{1}{2}\left\|\boldsymbol{y}_0^{s,t+1} - \boldsymbol{y}_0^{s,t}\right\|^2 - \frac{1 - L\alpha_s}{2}\left\|\boldsymbol{y}_0^{s,t+1} - \boldsymbol{y}_0^{s,t}\right\|^2 \\
&\leq \alpha_s\left[g(\boldsymbol{x}_0, \boldsymbol{y}_0^{s,t}) + \left\langle \nabla_{\boldsymbol{y}} G(\boldsymbol{x}_0, \boldsymbol{y}_0^{s,t}; \tilde{\xi}_0^{s,t}), \boldsymbol{y}_0^{s,t+1} - \boldsymbol{y}_0^{s,t}\right\rangle\right] + \frac{1}{2}\left\|\boldsymbol{y}_0^{s,t+1} - \boldsymbol{y}_0^{s,t}\right\|^2 - \frac{1 - L\alpha_s}{2}\left\|\boldsymbol{y}_0^{s,t+1} - \boldsymbol{y}_0^{s,t}\right\|^2 \\
&\quad - \alpha_s\left\langle \nabla_{\boldsymbol{y}} G(\boldsymbol{x}_0, \boldsymbol{y}_0^{s,t}; \tilde{\xi}_0^{s,t}) - \nabla_{\boldsymbol{y}} g(\boldsymbol{x}_0, \boldsymbol{y}_0^{s,t}), \boldsymbol{y}_0^{s,t+1} - \boldsymbol{y}_0^{s,t}\right\rangle \\
&\leq \alpha_s\left[g(\boldsymbol{x}_0, \boldsymbol{y}_0^{s,t}) + \left\langle \nabla_{\boldsymbol{y}} G(\boldsymbol{x}_0, \boldsymbol{y}_0^{s,t}; \tilde{\xi}_0^{s,t}), \boldsymbol{y}_0^{s,t+1} - \boldsymbol{y}_0^{s,t}\right\rangle\right] + \frac{1}{2}\left\|\boldsymbol{y}_0^{s,t+1} - \boldsymbol{y}_0^{s,t}\right\|^2 \\
&\quad + \alpha_s\left\|\nabla_{\boldsymbol{y}} G(\boldsymbol{x}_0, \boldsymbol{y}_0^{s,t}; \tilde{\xi}_0^{s,t}) - \nabla_{\boldsymbol{y}} g(\boldsymbol{x}_0, \boldsymbol{y}_0^{s,t})\right\| \left\|\boldsymbol{y}_0^{s,t+1} - \boldsymbol{y}_0^{s,t}\right\| - \frac{1 - L\alpha_s}{2}\left\|\boldsymbol{y}_0^{s,t+1} - \boldsymbol{y}_0^{s,t}\right\|^2 \\
&\stackrel{(i)}{\leq} \alpha_s\left[g(\boldsymbol{x}_0, \boldsymbol{y}_0^{s,t}) + \left\langle \nabla_{\boldsymbol{y}} G(\boldsymbol{x}_0, \boldsymbol{y}_0^{s,t}; \tilde{\xi}_0^{s,t}), \boldsymbol{y}_0^{s,t+1} - \boldsymbol{y}_0^{s,t}\right\rangle\right] + \frac{1}{2}\left\|\boldsymbol{y}_0^{s,t+1} - \boldsymbol{y}_0^{s,t}\right\|^2 \\
&\quad + \frac{\alpha_s^2\left\|\nabla_{\boldsymbol{y}} G(\boldsymbol{x}_0, \boldsymbol{y}_0^{s,t}; \tilde{\xi}_0^{s,t}) - \nabla_{\boldsymbol{y}} g(\boldsymbol{x}_0, \boldsymbol{y}_0^{s,t})\right\|^2}{2(1 - L\alpha_s)} \\
&\stackrel{(ii)}{\leq} \alpha_s\left[g(\boldsymbol{x}_0, \boldsymbol{y}_0^{s,t}) + \left\langle \nabla_{\boldsymbol{y}} G(\boldsymbol{x}_0, \boldsymbol{y}_0^{s,t}; \tilde{\xi}_0^{s,t}), \boldsymbol{y}_0^{s,t+1} - \boldsymbol{y}_0^{s,t}\right\rangle\right] + \frac{1}{2}\left\|\boldsymbol{y}_0^{s,t+1} - \boldsymbol{y}_0^{s,t}\right\|^2 \\
&\quad + \alpha_s^2\left\|\nabla_{\boldsymbol{y}} G(\boldsymbol{x}_0, \boldsymbol{y}_0^{s,t}; \tilde{\xi}_0^{s,t}) - \nabla_{\boldsymbol{y}} g(\boldsymbol{x}_0, \boldsymbol{y}_0^{s,t})\right\|^2
\end{aligned} \tag{21}$$

where $(i)$ follows from the inequality $bu - \frac{au^2}{2} \leq \frac{b^2}{2a}$ for any $a > 0$, where we set

$$u = \left\|\boldsymbol{y}_0^{s,t+1} - \boldsymbol{y}_0^{s,t}\right\|, \quad b = \alpha_s\left\|\nabla_{\boldsymbol{y}} G(\boldsymbol{x}_0, \boldsymbol{y}_0^{s,t}; \tilde{\xi}_0^{s,t}) - \nabla_{\boldsymbol{y}} g(\boldsymbol{x}_0, \boldsymbol{y}_0^{s,t})\right\|, \quad a = 1 - L\alpha_s,$$

and $(ii)$ follows from $\alpha_s \leq \frac{1}{2L}$.

Also, we have

$$\alpha_s \left[ g(\boldsymbol{x}_0, \boldsymbol{y}_0^{s,t}) + \left\langle \nabla_{\boldsymbol{y}} G(\boldsymbol{x}_0, \boldsymbol{y}_0^{s,t}; \tilde{\xi}_0^{s,t}), \boldsymbol{y}_0^{s,t+1} - \boldsymbol{y}_0^{s,t} \right\rangle \right] + \frac{1}{2} \left\| \boldsymbol{y}_0^{s,t+1} - \boldsymbol{y}_0^{s,t} \right\|^2$$

$$\overset{(i)}{\leq} \alpha_s g(\boldsymbol{x}_0, \boldsymbol{y}_0^{s,t}) + \left[ \alpha_s \left\langle \nabla_{\boldsymbol{y}} G(\boldsymbol{x}_0, \boldsymbol{y}_0^{s,t}; \tilde{\xi}_0^{s,t}), \boldsymbol{y}_0^{s,t+1} - \boldsymbol{y}_0^{s,t} \right\rangle + \frac{1}{2} \left\| \boldsymbol{y}_0^{s,t+1} - \boldsymbol{y}_0^{s,t} \right\|^2 \right]$$

$$+ \left[ \alpha_s \left\langle \nabla_{\boldsymbol{y}} G(\boldsymbol{x}_0, \boldsymbol{y}_0^{s,t}; \tilde{\xi}_0^{s,t}), \boldsymbol{y}_0^* - \boldsymbol{y}_0^{s,t+1} \right\rangle - \frac{1}{2} \left\| \boldsymbol{y}_0^{s,t+1} - \boldsymbol{y}_0^{s,t} \right\|^2 + \frac{1}{2} \left\| \boldsymbol{y}_0^{s,t} - \boldsymbol{y}_0^* \right\|^2 - \frac{1}{2} \left\| \boldsymbol{y}_0^{s,t+1} - \boldsymbol{y}_0^* \right\|^2 \right]$$

$$\leq \alpha_s \left[ g(\boldsymbol{x}_0, \boldsymbol{y}_0^{s,t}) + \left\langle \nabla_{\boldsymbol{y}} g(\boldsymbol{x}_0, \boldsymbol{y}_0^{s,t}), \boldsymbol{y}_0^* - \boldsymbol{y}_0^{s,t} \right\rangle \right] + \alpha_s \left\langle \nabla_{\boldsymbol{y}} G(\boldsymbol{x}_0, \boldsymbol{y}_0^{s,t}; \tilde{\xi}_0^{s,t}) - \nabla_{\boldsymbol{y}} g(\boldsymbol{x}_0, \boldsymbol{y}_0^{s,t}), \boldsymbol{y}_0^* - \boldsymbol{y}_0^{s,t} \right\rangle$$

$$+ \frac{1}{2} \left\| \boldsymbol{y}_0^{s,t} - \boldsymbol{y}_0^* \right\|^2 - \frac{1}{2} \left\| \boldsymbol{y}_0^{s,t+1} - \boldsymbol{y}_0^* \right\|^2$$

$$\overset{(ii)}{\leq} \alpha_s g(\boldsymbol{x}_0, \boldsymbol{y}_0^*) + \alpha_s \left\langle \nabla_{\boldsymbol{y}} G(\boldsymbol{x}_0, \boldsymbol{y}_0^{s,t}; \tilde{\xi}_0^{s,t}) - \nabla_{\boldsymbol{y}} g(\boldsymbol{x}_0, \boldsymbol{y}_0^{s,t}), \boldsymbol{y}_0^* - \boldsymbol{y}_0^{s,t} \right\rangle + \frac{1}{2} \left\| \boldsymbol{y}_0^{s,t} - \boldsymbol{y}_0^* \right\|^2 - \frac{1}{2} \left\| \boldsymbol{y}_0^{s,t+1} - \boldsymbol{y}_0^* \right\|^2$$

$$\tag{22}$$

where $(i)$ follows from

$$\alpha_s \left\langle \nabla_{\boldsymbol{y}} G(\boldsymbol{x}_0, \boldsymbol{y}_0^{s,t}; \tilde{\xi}_0^{s,t}), \boldsymbol{y}_0^* - \boldsymbol{y}_0^{s,t+1} \right\rangle - \frac{1}{2} \left\| \boldsymbol{y}_0^{s,t+1} - \boldsymbol{y}_0^{s,t} \right\|^2 + \frac{1}{2} \left\| \boldsymbol{y}_0^{s,t} - \boldsymbol{y}_0^* \right\|^2 - \frac{1}{2} \left\| \boldsymbol{y}_0^{s,t+1} - \boldsymbol{y}_0^* \right\|^2$$

$$= \alpha_s \left\langle \nabla_{\boldsymbol{y}} G(\boldsymbol{x}_0, \boldsymbol{y}_0^{s,t}; \tilde{\xi}_0^{s,t}), \boldsymbol{y}_0^* - \boldsymbol{y}_0^{s,t+1} \right\rangle - \frac{1}{2} \left\| \boldsymbol{y}_0^{s,t+1} - \boldsymbol{y}_0^* + \boldsymbol{y}_0^* - \boldsymbol{y}_0^{s,t} \right\|^2 + \frac{1}{2} \left\| \boldsymbol{y}_0^{s,t} - \boldsymbol{y}_0^* \right\|^2 - \frac{1}{2} \left\| \boldsymbol{y}_0^{s,t+1} - \boldsymbol{y}_0^* \right\|^2$$

$$= \alpha_s \left\langle \nabla_{\boldsymbol{y}} G(\boldsymbol{x}_0, \boldsymbol{y}_0^{s,t}; \tilde{\xi}_0^{s,t}), \boldsymbol{y}_0^* - \boldsymbol{y}_0^{s,t+1} \right\rangle - \frac{1}{2} \left\| \boldsymbol{y}_0^{s,t+1} - \boldsymbol{y}_0^* \right\|^2 - \frac{1}{2} \left\| \boldsymbol{y}_0^* - \boldsymbol{y}_0^{s,t} \right\|^2 - \left\langle \boldsymbol{y}_0^{s,t+1} - \boldsymbol{y}_0^*, \boldsymbol{y}_0^* - \boldsymbol{y}_0^{s,t} \right\rangle$$

$$+ \frac{1}{2} \left\| \boldsymbol{y}_0^{s,t} - \boldsymbol{y}_0^* \right\|^2 - \frac{1}{2} \left\| \boldsymbol{y}_0^{s,t+1} - \boldsymbol{y}_0^* \right\|^2$$

$$= \alpha_s \left\langle \nabla_{\boldsymbol{y}} G(\boldsymbol{x}_0, \boldsymbol{y}_0^{s,t}; \tilde{\xi}_0^{s,t}), \boldsymbol{y}_0^* - \boldsymbol{y}_0^{s,t+1} \right\rangle - \left\| \boldsymbol{y}_0^{s,t+1} - \boldsymbol{y}_0^* \right\|^2 - \left\langle \boldsymbol{y}_0^{s,t+1} - \boldsymbol{y}_0^*, \boldsymbol{y}_0^* - \boldsymbol{y}_0^{s,t} \right\rangle$$

$$= \alpha_s \left\langle \nabla_{\boldsymbol{y}} G(\boldsymbol{x}_0, \boldsymbol{y}_0^{s,t}; \tilde{\xi}_0^{s,t}), \boldsymbol{y}_0^* - \boldsymbol{y}_0^{s,t+1} \right\rangle - \left\langle \boldsymbol{y}_0^{s,t+1} - \boldsymbol{y}_0^*, \boldsymbol{y}_0^{s,t+1} - \boldsymbol{y}_0^* \right\rangle - \left\langle \boldsymbol{y}_0^{s,t+1} - \boldsymbol{y}_0^*, \boldsymbol{y}_0^* - \boldsymbol{y}_0^{s,t} \right\rangle$$

$$= \alpha_s \left\langle \nabla_{\boldsymbol{y}} G(\boldsymbol{x}_0, \boldsymbol{y}_0^{s,t}; \tilde{\xi}_0^{s,t}), \boldsymbol{y}_0^* - \boldsymbol{y}_0^{s,t+1} \right\rangle - \left\langle \boldsymbol{y}_0^{s,t+1} - \boldsymbol{y}_0^*, \boldsymbol{y}_0^{s,t+1} - \boldsymbol{y}_0^{s,t} \right\rangle$$

$$= \left\langle \boldsymbol{y}_0^{s,t+1} - \boldsymbol{y}_0^*, \boldsymbol{y}_0^{s,t} - \alpha_s \nabla_{\boldsymbol{y}} G(\boldsymbol{x}_0, \boldsymbol{y}_0^{s,t}; \tilde{\xi}_0^{s,t}) - \boldsymbol{y}_0^{s,t+1} \right\rangle \geq 0,$$

and $(ii)$ follows from (strong) convexity of $g(\boldsymbol{x}, \boldsymbol{y})$ with respect to $\boldsymbol{y}$.

Combing (21) and (22) yields

$$\alpha_s g(\boldsymbol{x}_0, \boldsymbol{y}_0^{s,t+1}) - \alpha_s g(\boldsymbol{x}_0, \boldsymbol{y}_0^*) \leq \frac{1}{2} \left\| \boldsymbol{y}_0^{s,t} - \boldsymbol{y}_0^* \right\|^2 - \frac{1}{2} \left\| \boldsymbol{y}_0^{s,t+1} - \boldsymbol{y}_0^* \right\|^2 + \pi_t(\boldsymbol{y}_0^*), \tag{23}$$

where we define

$$\pi_t(\boldsymbol{y}_0^*) := 2\alpha_s^2 \left\| \nabla_{\boldsymbol{y}} G(\boldsymbol{x}_0, \boldsymbol{y}_0^{s,t}; \tilde{\xi}_0^{s,t}) - \nabla_{\boldsymbol{y}} g(\boldsymbol{x}_0, \boldsymbol{y}_0^{s,t}) \right\|^2 + \alpha_s \left\langle \nabla_{\boldsymbol{y}} G(\boldsymbol{x}_0, \boldsymbol{y}_0^{s,t}; \tilde{\xi}_0^{s,t}) - \nabla_{\boldsymbol{y}} g(\boldsymbol{x}_0, \boldsymbol{y}_0^{s,t}), \boldsymbol{y}_0^* - \boldsymbol{y}_0^{s,t} \right\rangle.$$

Summing up (23) from $t = 0$ to $T_s - 1$, we have

$$\sum_{t=0}^{T_s-1} \alpha_s \left[ g(\boldsymbol{x}_0, \boldsymbol{y}_0^{s,t+1}) - g(\boldsymbol{x}_0, \boldsymbol{y}_0^*) \right] \leq \frac{1}{2} \left\| \boldsymbol{y}_0^{s,0} - \boldsymbol{y}_0^* \right\|^2 - \frac{1}{2} \left\| \boldsymbol{y}_0^{s,T_s} - \boldsymbol{y}_0^* \right\|^2 + \sum_{t=0}^{T_s-1} \pi_t(\boldsymbol{y}_0^*)$$

$$\leq \frac{1}{2} \left\| \boldsymbol{y}_0^{s,0} - \boldsymbol{y}_0^* \right\|^2 + \sum_{t=0}^{T_s-1} \pi_t(\boldsymbol{y}_0^*) \leq D^2 + \sum_{t=0}^{T_s-1} \pi_t(\boldsymbol{y}_0^*)$$

By (strong) convexity of $g(\boldsymbol{x}, \boldsymbol{y})$ with respect to $\boldsymbol{y}$, we have

$$g(\boldsymbol{x}_0, \boldsymbol{y}_0^{s+1,0}) = g\left( \boldsymbol{x}_0, \frac{1}{T_s} \sum_{t=1}^{T_s} \boldsymbol{y}_0^{s,t} \right) \leq \frac{1}{T_s} \sum_{t=1}^{T_s} g(\boldsymbol{x}_0, \boldsymbol{y}_0^{s,t}),$$

which implies that

$$\sum_{t=0}^{T_s-1} \alpha_s \left[ g(\boldsymbol{x}_0, \boldsymbol{y}_0^{s+1,0}) - g(\boldsymbol{x}_0, \boldsymbol{y}_0^*) \right] \leq D^2 + \sum_{t=0}^{T_s-1} \pi_t(\boldsymbol{y}_0^*).$$

Denote $\psi_t := \alpha_s \left\langle \nabla_{\boldsymbol{y}} G(\boldsymbol{x}_0, \boldsymbol{y}_0^{s,t}; \tilde{\xi}_0^{s,t}) - \nabla_{\boldsymbol{y}} g(\boldsymbol{x}_0, \boldsymbol{y}_0^{s,t}), \boldsymbol{y}_0^* - \boldsymbol{y}_0^{s,t} \right\rangle$ and observing that

$$\pi_t(\boldsymbol{y}_0^*) = \psi_t + 2\alpha_s^2 \left\| \nabla_{\boldsymbol{y}} G(\boldsymbol{x}_0, \boldsymbol{y}_0^{s,t}; \tilde{\xi}_0^{s,t}) - \nabla_{\boldsymbol{y}} g(\boldsymbol{x}_0, \boldsymbol{y}_0^{s,t}) \right\|^2,$$

we then conclude that

$$\left( \sum_{t=0}^{T_s-1} \alpha_s \right) \left[ g(\boldsymbol{x}_0, \boldsymbol{y}_0^{s+1,0}) - g(\boldsymbol{x}_0, \boldsymbol{y}_0^*) \right] \leq D^2 + \sum_{t=0}^{T_s-1} \left[ \psi_t + 2\alpha_s^2 \left\| \nabla_{\boldsymbol{y}} G(\boldsymbol{x}_0, \boldsymbol{y}_0^{s,t}; \tilde{\xi}_0^{s,t}) - \nabla_{\boldsymbol{y}} g(\boldsymbol{x}_0, \boldsymbol{y}_0^{s,t}) \right\|^2 \right]. \tag{24}$$

Note that by Assumption 3 we have $\mathbb{E}[\psi_t \mid \widetilde{\mathcal{F}}_0^{s,t}] = 0$, thus $\{\psi_t\}_{t\geq 0}$ is a martingale-difference sequence. Moreover, $\left\| \boldsymbol{y}_0^* - \boldsymbol{y}_0^{s,t} \right\| \leq \left\| \boldsymbol{y}_0^* - \boldsymbol{y}_0^{s,0} \right\| + \left\| \boldsymbol{y}_0^{s,0} - \boldsymbol{y}_0^{s,t} \right\| \leq 2\sqrt{2}D$, then we obtain

$$\mathbb{E}\left[ \exp\left( \frac{\psi_t^2}{\left(2\sqrt{2}D\alpha_s\sigma_{g,1}\right)^2} \right) \mid \widetilde{\mathcal{F}}_0^{s,t} \right] \leq \mathbb{E}\left[ \exp\left( \frac{\alpha_s^2 \left\| \nabla_{\boldsymbol{y}} G(\boldsymbol{x}_0, \boldsymbol{y}_0^{s,t}; \tilde{\xi}_0^{s,t}) - \nabla_{\boldsymbol{y}} g(\boldsymbol{x}_0, \boldsymbol{y}_0^{s,t}) \right\|^2 \left\| \boldsymbol{y}_0^* - \boldsymbol{y}_0^{s,t} \right\|^2}{\alpha_s^2 \sigma_{g,1}^2 \left(2\sqrt{2}D\right)^2} \right) \mid \widetilde{\mathcal{F}}_0^{s,t} \right]$$

$$\leq \mathbb{E}\left[ \exp\left( \frac{\left\| \nabla_{\boldsymbol{y}} G(\boldsymbol{x}_0, \boldsymbol{y}_0^{s,t}; \tilde{\xi}_0^{s,t}) - \nabla_{\boldsymbol{y}} g(\boldsymbol{x}_0, \boldsymbol{y}_0^{s,t}) \right\|^2}{\sigma_{g,1}^2} \right) \mid \widetilde{\mathcal{F}}_0^{s,t} \right]$$

$$\overset{(i)}{\leq} \exp(1)$$

where $(i)$ follows from Assumption 3. By Lemma 2 in Lan et al. (2012), for any $\lambda \geq 0$ we have

$$\Pr\left[ \sum_{t=0}^{T_s-1} \psi_t > \lambda \left( 2\sqrt{2}D\sigma_{g,1}\alpha_s\sqrt{T_s} \right) \right] = \Pr\left[ \sum_{t=0}^{T_s-1} \psi_t > \lambda \left( 2\sqrt{2}D\sigma_{g,1}\sqrt{\sum_{t=0}^{T_s-1} \alpha_s^2} \right) \right] \tag{25}$$

$$\leq \exp(-\lambda^2/3),$$

where the probability is taken over randomness in $\widetilde{\mathcal{F}}_0^{s,T_s}$. Also, we have

$$\exp\left( \frac{1}{T_s\alpha_s^2} \sum_{t=0}^{T_s-1} \frac{\alpha_s^2 \left\| \nabla_{\boldsymbol{y}} G(\boldsymbol{x}_0, \boldsymbol{y}_0^{s,t}; \tilde{\xi}_0^{s,t}) - \nabla_{\boldsymbol{y}} g(\boldsymbol{x}_0, \boldsymbol{y}_0^{s,t}) \right\|^2}{\sigma_{g,1}^2} \right)$$

$$\leq \frac{1}{T_s\alpha_s^2} \sum_{t=0}^{T_s-1} \alpha_s^2 \exp\left( \frac{\left\| \nabla_{\boldsymbol{y}} G(\boldsymbol{x}_0, \boldsymbol{y}_0^{s,t}; \tilde{\xi}_0^{s,t}) - \nabla_{\boldsymbol{y}} g(\boldsymbol{x}_0, \boldsymbol{y}_0^{s,t}) \right\|^2}{\sigma_{g,1}^2} \right).$$

Then take expectations (with respect to $\widetilde{\mathcal{F}}_0^{s,T_s}$) on both sides and we get

$$\mathbb{E}\left[ \exp\left( \frac{\sum_{t=0}^{T_s-1} \alpha_s^2 \left\| \nabla_{\boldsymbol{y}} G(\boldsymbol{x}_0, \boldsymbol{y}_0^{s,t}; \tilde{\xi}_0^{s,t}) - \nabla_{\boldsymbol{y}} g(\boldsymbol{x}_0, \boldsymbol{y}_0^{s,t}) \right\|^2}{T_s\alpha_s^2\sigma_{g,1}^2} \right) \right] \overset{(i)}{\leq} \frac{1}{T_s\alpha_s^2} \sum_{t=0}^{T_s-1} \alpha_s^2 \exp(1) \leq \exp(1),$$

where $(i)$ follows from Assumption 3.

Using Markov's inequality, for any $\lambda \geq 0$ we have

$$\Pr\left[ \sum_{t=0}^{T_s-1} \alpha_s^2 \left\| \nabla_{\boldsymbol{y}} G(\boldsymbol{x}_0, \boldsymbol{y}_0^{s,t}; \tilde{\xi}_0^{s,t}) - \nabla_{\boldsymbol{y}} g(\boldsymbol{x}_0, \boldsymbol{y}_0^{s,t}) \right\|^2 > (1+\lambda)\sigma_{g,1}^2\alpha_s^2 T_s \right] \leq \exp(-\lambda). \tag{26}$$

Combining (24), (25) and (26), and rearranging the terms, we obtain

$$\Pr\left[\left(\sum_{t=0}^{T_s-1}\alpha_s\right)\left[g(\boldsymbol{x}_0,\boldsymbol{y}_0^{s+1,0})-g(\boldsymbol{x}_0,\boldsymbol{y}_0^*)\right]>D^2+\lambda\left(2\sqrt{2}D\sigma_{g,1}\alpha_s\sqrt{T_s}\right)+2(1+\lambda)\sigma_{g,1}^2\alpha_s^2T_s\right]$$

$$\leq\Pr\left[D^2+\sum_{t=0}^{T_s-1}\psi_t+\sum_{t=0}^{T_s-1}2\alpha_s^2\left\|\nabla_{\boldsymbol{y}}G(\boldsymbol{x}_0,\boldsymbol{y}_0^{s,t};\tilde{\xi}_0^{s,t})-\nabla_{\boldsymbol{y}}g(\boldsymbol{x}_0,\boldsymbol{y}_0^{s,t})\right\|^2\right.$$
$$\left.>D^2+\lambda\left(2\sqrt{2}D\sigma_{g,1}\alpha_s\sqrt{T_s}\right)+2(1+\lambda)\sigma_{g,1}^2\alpha_s^2T_s\right]$$

$$\leq\Pr\left[\sum_{t=0}^{T_s-1}\psi_t>\lambda\left(2\sqrt{2}D\sigma_{g,1}\alpha_s\sqrt{T_s}\right)\right]+\Pr\left[\sum_{t=0}^{T_s-1}\alpha_s^2\left\|\nabla_{\boldsymbol{y}}G(\boldsymbol{x}_0,\boldsymbol{y}_0^{s,t};\tilde{\xi}_0^{s,t})-\nabla_{\boldsymbol{y}}g(\boldsymbol{x}_0,\boldsymbol{y}_0^{s,t})\right\|^2>(1+\lambda)\sigma_{g,1}^2\alpha_s^2T_s\right]$$

$$\leq\exp(-\lambda^2/3)+\exp(-\lambda).$$

Note that $\sum_{t=0}^{T_s-1}\alpha_s=\alpha_sT_s$, and recall the definition (20) of $\mathcal{P}_0(T_s)$ and $\mathcal{P}_1(T_s)$, by rearranging we finally conclude that for any $\lambda\geq0$, we have

$$\Pr\left[g(\boldsymbol{x}_0,\boldsymbol{y}_0^{s+1,0})-g(\boldsymbol{x}_0,\boldsymbol{y}_0^*)>\mathcal{P}_0(T_s)+\lambda\mathcal{P}_1(T_s)\right]\leq\exp(-\lambda^2/3)+\exp(-\lambda).$$

$\square$

In Lemma D.1, we got the high probability convergence results for SGD for one epoch. We now adopt a shrinking ball technique with multiple epochs to improve the high probability bound compared with the one epoch result (Hazan & Kale, 2014; Ghadimi & Lan, 2013a). The main difference in our setting is that we directly utilize SGD in each epoch and consider smooth and strongly convex functions (i.e., our lower-level problem). In contrast, Hazan & Kale (2014) considered a nonsmooth strongly convex function while Ghadimi & Lan (2013a) considered an accelerated algorithm AC-SA for each epoch. This result is stated in Lemma 3.2.

For notation convenience, we define $\mathcal{V}_0$ as the upper bound for $g(\boldsymbol{x}_0,\boldsymbol{y}_0^{0,0})-g(\boldsymbol{x}_0,\boldsymbol{y}_0^*)$, i.e.,

$$g(\boldsymbol{x}_0,\boldsymbol{y}_0^{0,0})-g(\boldsymbol{x}_0,\boldsymbol{y}_0^*)\leq\mathcal{V}_0. \tag{27}$$

We also define the projection ball $\mathcal{B}_s$ and set the number of iterations $T_s$ and the fixed step-size $\alpha_s$ at $s$-th epoch in Alogorithm 3 as following:

$$\mathcal{B}_s:=\left\{\boldsymbol{y}\in\mathbb{R}^{d_y}:\frac{1}{2}\left\|\boldsymbol{y}-\boldsymbol{y}_0^{s,0}\right\|^2\leq\frac{\mathcal{V}_0}{\mu2^s}\right\}, \tag{28}$$

$$T_s=\left\lceil\max\left\{\frac{16L}{\mu},\frac{32\max\{\sigma_{g,1}^2,4\lambda^2\sigma_{g,1}^2\}}{\mu\mathcal{V}_02^{-(s+2)}}\right\}\right\rceil, \tag{29}$$

$$\alpha_s=\min\left\{\frac{1}{2L},\frac{1}{\sigma_{g,1}}\sqrt{\frac{\mathcal{V}_02^{-s}}{2\mu T_s}}\right\}. \tag{30}$$

**Lemma D.2** (Initialization Refinement, Lemma 3.2 restated). *Given $\delta\in(0,1)$ and $\epsilon'>0$, set parameter $k^\dagger=\lceil\log_2(\mathcal{V}_0/\epsilon')\rceil$, where $\mathcal{V}_0$ is defined in (27). If we run Algorithm 3 for $k^\dagger$ epochs, with projection ball $\mathcal{B}_s$, the number of iterations $T_s$ and the fixed step-size $\alpha_s$ at each epoch defined as (28), (29) and (30), where we set $\lambda$ to be*

$$\lambda=\max\left(\sqrt{3\ln\left(\frac{2k^\dagger}{\delta}\right)},\ln\left(\frac{2k^\dagger}{\delta}\right)\right),$$

*then with probability at least $1-\delta/2$ over randomness in $\sigma\left\{\cup_{s=0}^{k^\dagger-1}\widetilde{\mathcal{F}}_0^{s,T_s}\right\}$, we have*

$$Pr\left[g(\boldsymbol{x}_0,\boldsymbol{y}_0)-g(\boldsymbol{x}_0,\boldsymbol{y}^*(\boldsymbol{x}_0))>\epsilon'\right]\leq\frac{\delta}{2}. \tag{31}$$

*Moreover, the total number of iterations performed by Algorithm 3 to find such a solution is bounded by $O(\mathcal{T}(\epsilon',\delta))$, where*

$$\mathcal{T}(\epsilon',\delta):=\frac{L}{\mu}\max\left(1,\log_2\left(\frac{\mathcal{V}_0}{\epsilon'}\right)\right)+\frac{\sigma_{g,1}^2}{\mu\epsilon'}+\left[\ln\left(\frac{2\log_2(\mathcal{V}_0/\epsilon')}{\delta}\right)\right]^2\frac{\sigma_{g,1}^2}{\mu\epsilon'}. \tag{32}$$

In particular, if we set $\epsilon' = \frac{\mu}{2}\left(\frac{\epsilon}{8K_0}\right)^2$, then with probability at least $1 - \delta/2$ over randomness in $\sigma\left\{\cup_{s=0}^{k^\dagger-1}\widetilde{\mathcal{F}}_0^{s,T_s}\right\}$ (this event is denoted as $\mathcal{E}_0$), we have $\|\boldsymbol{y}_0 - \boldsymbol{y}^*(\boldsymbol{x}_0)\| \leq \epsilon/8K_0$ in $\widetilde{O}\left(\sigma_{g,1}^2 K_0^2/\mu^2\epsilon^2\right)$ iterations.

***Proof of Lemma D.2***. For any $s \geq 0$, let $\mathcal{V}_s = \mathcal{V}_0 2^{-s}$ and denote the event $\mathcal{A}_s := \left\{g(\boldsymbol{x}_0, \boldsymbol{y}_0^{s,0}) - g(\boldsymbol{x}_0, \boldsymbol{y}_0^*) \leq \mathcal{V}_s\right\}$. We first show that

$$\Pr\left[g(\boldsymbol{x}_0, \boldsymbol{y}_0^{s+1,0}) - g(\boldsymbol{x}_0, \boldsymbol{y}_0^*) \geq \mathcal{V}_{s+1} \mid \mathcal{A}_s\right] \leq \frac{\delta}{2k^\dagger}. \tag{33}$$

By strong convexity of $g(\boldsymbol{x}, \boldsymbol{y})$ with respect to $\boldsymbol{y}$, we have

$$\frac{1}{2}\left\|\boldsymbol{y}_0^{s,0} - \boldsymbol{y}_0^*\right\|^2 \leq \frac{g(\boldsymbol{x}_0, \boldsymbol{y}_0^{s,0}) - g(\boldsymbol{x}_0, \boldsymbol{y}_0^*)}{\mu} \leq \frac{\mathcal{V}_s}{\mu}.$$

With this upper bound for $\frac{1}{2}\left\|\boldsymbol{y}_0^{s,0} - \boldsymbol{y}_0^*\right\|^2$ under the event $\mathcal{A}_s$, we can substitute $\mathcal{V}_s/\mu$ for $D^2$ in (20) of Lemma D.1. Then we redefine $\mathcal{P}_0(T_s)$ and $\mathcal{P}_1(T_s)$ as

$$\begin{aligned}
\mathcal{P}_0(T_s) &:= \frac{1}{\alpha_s T_s}\left[\frac{\mathcal{V}_s}{\mu} + 2\sigma_{g,1}^2\alpha_s^2 T_s\right], \\
\mathcal{P}_1(T_s) &:= \frac{1}{\alpha_s T_s}\left[2\sqrt{2}\sqrt{\frac{\mathcal{V}_s}{\mu}}\sigma_{g,1}\alpha_s\sqrt{T_s} + 2\sigma_{g,1}^2\alpha_s^2 T_s\right].
\end{aligned} \tag{34}$$

By Lemma D.1 and definition (34) we have

$$\Pr\left[g(\boldsymbol{x}_0, \boldsymbol{y}_0^{s+1,0}) - g(\boldsymbol{x}_0, \boldsymbol{y}_0^*) > \mathcal{P}_0(T_s) + \lambda\mathcal{P}_1(T_s) \mid \mathcal{A}_s\right] \leq \exp(-\lambda^2/3) + \exp(-\lambda). \tag{35}$$

Define

$$\begin{aligned}
\mathcal{R}_1(T_s) &:= \frac{2L\mathcal{V}_s}{\mu}\frac{1}{T_s} \overset{(i)}{\leq} \frac{2L\mathcal{V}_s}{\mu}\frac{\mu}{16L} = \frac{\mathcal{V}_s}{8} = \frac{\mathcal{V}_{s+1}}{4}, \\
\mathcal{R}_2(T_s) &:= \frac{2\sigma_{g,1}^2\mathcal{V}_s}{\mu}\frac{1}{T_s} \overset{(ii)}{\leq} \frac{2\sigma_{g,1}^2\mathcal{V}_s}{\mu}\frac{\mu\mathcal{V}_0 2^{-(s+2)}}{32\sigma_{g,1}^2} = \frac{\mathcal{V}_{s+2}\mathcal{V}_s}{16} = \frac{\mathcal{V}_{s+1}^2}{16},
\end{aligned} \tag{36}$$

where both $(i)$ and $(ii)$ follow from (29).

Then we conclude that

$$\begin{aligned}
\mathcal{P}_0(T_s) &= \frac{\mathcal{V}_s}{\mu T_s}\frac{1}{\alpha_s} + 2\sigma_{g,1}^2\alpha_s \\
&\overset{(i)}{\leq} \frac{\mathcal{V}_s}{\mu T_s}\max\left\{2L, \sigma_{g,1}\sqrt{\frac{2\mu T_s}{\mathcal{V}_0 2^{-s}}}\right\} + 2\sigma_{g,1}\sqrt{\frac{\mathcal{V}_0 2^{-s}}{2\mu T_s}} \\
&\leq \max\left\{\frac{2L\mathcal{V}_s}{\mu T_s}, \sigma_{g,1}\sqrt{\frac{2\mathcal{V}_s}{\mu T_s}}\right\} + \sigma_{g,1}\sqrt{\frac{2\mathcal{V}_s}{\mu T_s}} \\
&\overset{(ii)}{\leq} \max\left\{\mathcal{R}_1(T_s), \sqrt{\mathcal{R}_2(T_s)}\right\} + \sqrt{\mathcal{R}_2(T_s)} \\
&\leq \frac{\mathcal{V}_{s+1}}{4} + \frac{\mathcal{V}_{s+1}}{4} \leq \frac{\mathcal{V}_{s+1}}{2}.
\end{aligned}$$

where $(i)$ follows from (30) and $(ii)$ follows from (36).

Also, we have

$$\mathcal{P}_1(T_s) = 2\sqrt{2}\sigma_{g,1}\sqrt{\frac{\mathcal{V}_s}{\mu T_s}} + 2\sigma_{g,1}^2\alpha_s$$

$$\overset{(i)}{\leq} 2\sqrt{2}\sigma_{g,1}\sqrt{\frac{\mathcal{V}_s}{\mu T_s}} + 2\sigma_{g,1}\sqrt{\frac{\mathcal{V}_0 2^{-s}}{2\mu T_s}} \leq 2\sqrt{2}\sigma_{g,1}\sqrt{\frac{\mathcal{V}_s}{\mu T_s}} + \sqrt{2}\sigma_{g,1}\sqrt{\frac{\mathcal{V}_s}{\mu T_s}}$$

$$= 3\sqrt{2}\sigma_{g,1}\sqrt{\frac{\mathcal{V}_s}{\mu T_s}} \overset{(ii)}{\leq} 3\sqrt{2}\sigma_{g,1}\sqrt{\frac{\mathcal{V}_s}{\mu}}\sqrt{\frac{\mu \mathcal{V}_0 2^{-(s+2)}}{128\lambda^2\sigma_{g,1}^2}}$$

$$\leq 3\sqrt{2}\sigma_{g,1}\sqrt{\frac{\mathcal{V}_s \mathcal{V}_{s+2}}{128\lambda^2\sigma_{g,1}^2}}$$

$$= \frac{3\mathcal{V}_{s+1}}{8\lambda} \leq \frac{\mathcal{V}_{s+1}}{2\lambda}.$$

where $(i)$ follows from (30) and $(ii)$ follows from (29). Therefore, we have

$$\mathcal{P}_0(T_s) + \lambda\mathcal{P}_1(T_s) \leq \mathcal{V}_{s+1},$$

which implies that

$$\Pr\left[g(\boldsymbol{x}_0, \boldsymbol{y}_0^{s+1,0}) - g(\boldsymbol{x}_0, \boldsymbol{y}_0^*) > \mathcal{V}_{s+1} \mid \mathcal{A}_s\right] \leq \Pr\left[g(\boldsymbol{x}_0, \boldsymbol{y}_0^{s+1,0}) - g(\boldsymbol{x}_0, \boldsymbol{y}_0^*) > \mathcal{P}_0(T_s) + \lambda\mathcal{P}_1(T_s) \mid \mathcal{A}_s\right]$$

$$\overset{(i)}{\leq} \exp(-\lambda^2/3) + \exp(-\lambda)$$

$$\overset{(ii)}{\leq} \frac{\delta}{2k^\dagger}, \tag{37}$$

where $(i)$ follows from (35) and in $(ii)$ we set

$$\lambda = \max\left(\sqrt{3\ln\left(\frac{2k^\dagger}{\delta}\right)}, \ln\left(\frac{2k^\dagger}{\delta}\right)\right) \tag{38}$$

so that $\exp(-\lambda^2/3) + \exp(-\lambda) \leq \delta/2k^\dagger$.

Next, we proceed to show that for any given $\delta \in (0,1)$ and $\epsilon' > 0$, we have

$$\Pr\left[g(\boldsymbol{x}_0, \boldsymbol{y}_0^{k^\dagger,0}) - g(\boldsymbol{x}_0, \boldsymbol{y}_0^*) > \epsilon'\right] \leq \frac{\delta}{2}. \tag{39}$$

Let event $\bar{\mathcal{A}}_s$ be the complement of event $\mathcal{A}_s$. Obviously we have $\Pr[\mathcal{A}_0] = 1$, and thus $\Pr[\bar{\mathcal{A}}_0] = 0$. It can also be easily seen that

$$\Pr\left[g(\boldsymbol{x}_0, \boldsymbol{y}_0^{s+1,0}) - g(\boldsymbol{x}_0, \boldsymbol{y}_0^*) > \mathcal{V}_{s+1}\right]$$

$$= \Pr\left[g(\boldsymbol{x}_0, \boldsymbol{y}_0^{s+1,0}) - g(\boldsymbol{x}_0, \boldsymbol{y}_0^*) > \mathcal{V}_{s+1} \mid \mathcal{A}_s\right]\Pr[\mathcal{A}_s] + \Pr\left[g(\boldsymbol{x}_0, \boldsymbol{y}_0^{s+1,0}) - g(\boldsymbol{x}_0, \boldsymbol{y}_0^*) > \mathcal{V}_{s+1} \mid \bar{\mathcal{A}}_s\right]\Pr[\bar{\mathcal{A}}_s]$$

$$\leq \Pr\left[g(\boldsymbol{x}_0, \boldsymbol{y}_0^{s+1,0}) - g(\boldsymbol{x}_0, \boldsymbol{y}_0^*) > \mathcal{V}_{s+1} \mid \mathcal{A}_s\right] + \Pr[\bar{\mathcal{A}}_s]$$

$$\overset{(i)}{\leq} \frac{\delta}{2k^\dagger} + \Pr\left[g(\boldsymbol{x}_0, \boldsymbol{y}_0^{s,0}) - g(\boldsymbol{x}_0, \boldsymbol{y}_0^*) > \mathcal{V}_s\right],$$

where $(i)$ follows from (37).

Summing up both sides of the above inequality from $s = 0$ to $k^\dagger - 1$, we obtain

$$\Pr\left[g(\boldsymbol{x}_0, \boldsymbol{y}_0^{k^\dagger,0}) - g(\boldsymbol{x}_0, \boldsymbol{y}_0^*) > \epsilon'\right] = \Pr\left[g(\boldsymbol{x}_0, \boldsymbol{y}_0^{k^\dagger,0}) - g(\boldsymbol{x}_0, \boldsymbol{y}_0^*) > \epsilon'\right] - \Pr[\bar{\mathcal{A}}_0]$$

$$= \sum_{s=0}^{k^\dagger-1}\left\{\Pr\left[g(\boldsymbol{x}_0, \boldsymbol{y}_0^{s+1,0}) - g(\boldsymbol{x}_0, \boldsymbol{y}_0^*) > \mathcal{V}_{s+1}\right] - \Pr\left[g(\boldsymbol{x}_0, \boldsymbol{y}_0^{s,0}) - g(\boldsymbol{x}_0, \boldsymbol{y}_0^*) > \mathcal{V}_s\right]\right\}$$

$$\leq \frac{\delta}{2k^\dagger}k^\dagger = \frac{\delta}{2}.$$

Therefore, we conclude

$$\Pr\left[g(\boldsymbol{x}_0, \boldsymbol{y}_0) - g(\boldsymbol{x}_0, \boldsymbol{y}_0^*) > \epsilon'\right] \stackrel{(i)}{=} \Pr\left[g(\boldsymbol{x}_0, \boldsymbol{y}_0^{k^\dagger,0}) - g(\boldsymbol{x}_0, \boldsymbol{y}_0^*) > \epsilon'\right] \leq \frac{\delta}{2}.$$

where $(i)$ follows by recalling line 1 in Algorithm 1 and line 11 in Algorithm 3, i.e., $\boldsymbol{y}_0 = \boldsymbol{y}_0^{k^\dagger,0}$ is the output of Algorithm 3.

Moreover, the total number of iterations can be bounded by

$$
\begin{aligned}
\sum_{s=0}^{k^\dagger-1} T_s &\leq \sum_{s=0}^{k^\dagger-1}\left(\frac{16L}{\mu} + \frac{32\max\left\{\sigma_{g,1}^2, 4\lambda^2\sigma_{g,1}^2\right\}}{\mu\mathcal{V}_0 2^{-(s+2)}} + 1\right) \\
&\leq k^\dagger\left(\frac{16L}{\mu}+1\right) + \frac{32\max\left\{\sigma_{g,1}^2, 4\lambda^2\sigma_{g,1}^2\right\}}{\mu\mathcal{V}_0}\sum_{s=0}^{k^\dagger-1}2^{k+2} \\
&= k^\dagger\left(\frac{16L}{\mu}+1\right) + \frac{32\max\left\{\sigma_{g,1}^2, 4\lambda^2\sigma_{g,1}^2\right\}}{\mu\mathcal{V}_0}2^{k^\dagger+2} \\
&\leq \left(\log_2\left(\frac{\mathcal{V}_0}{\epsilon'}\right)+1\right)\left(\frac{16L}{\mu}+1\right) + \frac{256(4\lambda^2+1)\sigma_{g,1}^2}{\mu\epsilon'}
\end{aligned}
\tag{40}
$$

Using the above conclusion, the fact that $k^\dagger = \lceil\log_2(\mathcal{V}_0/\epsilon')\rceil$, the observation that $\lambda = O(\ln(k^\dagger/\delta))$, we conclude that the total number of iterations for Algorithm 3 is bounded by $O(\mathcal{T}(\epsilon',\delta))$, where

$$\mathcal{T}(\epsilon',\delta) := \frac{L}{\mu}\max\left(1, \log_2\left(\frac{\mathcal{V}_0}{\epsilon'}\right)\right) + \frac{\sigma_{g,1}^2}{\mu\epsilon'} + \left[\ln\left(\frac{2\log_2(\mathcal{V}_0/\epsilon')}{\delta}\right)\right]^2\frac{\sigma_{g,1}^2}{\mu\epsilon'}. \tag{41}$$

Specifically, for any given $\epsilon > 0$, if we need $\|\boldsymbol{y}_0 - \boldsymbol{y}^*(\boldsymbol{x}_0)\| \leq \frac{\epsilon}{8K_0}$ holds with probability at least $1 - \delta/2$, then by strong convexity of $g(\boldsymbol{x}, \boldsymbol{y})$ with respect to $\boldsymbol{y}$, we have to set $\epsilon' = \frac{\mu}{2}\left(\frac{\epsilon}{8K_0}\right)^2$ and thus by (40) we need to run Algorithm 3 for at most

$$\left(\log_2\left(\frac{128K_0^2\mathcal{V}_0}{\mu\epsilon^2}\right)+1\right)\left(\frac{16L}{\mu}+1\right) + \frac{256\times128K_0^2(4\lambda^2+1)\sigma_{g,1}^2}{\mu^2\epsilon^2} \tag{42}$$

iterations in total, where

$$
\begin{aligned}
\lambda &= \max\left(\sqrt{3\ln\left(\frac{2k^\dagger}{\delta}\right)}, \ln\left(\frac{2k^\dagger}{\delta}\right)\right) \\
&= \max\left\{\sqrt{3\ln\left[\frac{2\left\lceil\log_2\left(\frac{128\mathcal{V}_0K_0^2}{\mu\epsilon^2}\right)\right\rceil}{\delta}\right]}, \ln\left[\frac{2\left\lceil\log_2\left(\frac{128\mathcal{V}_0K_0^2}{\mu\epsilon^2}\right)\right\rceil}{\delta}\right]\right\}.
\end{aligned}
\tag{43}
$$

Therefore, the total number of iterations performed by Algorithm 3 is at most $\widetilde{O}\left(\sigma_{g,1}^2 K_0^2/\mu^2\epsilon^2\right)$ iterations in total. $\qquad\square$

### D.3 Proof of Lemma 3.3

The following lemma follows the same technique as in Lemma D.1, which can be regarded as high probability guarantee for one epoch SGD.

**Lemma D.3** (Periodic Updates, Lemma 3.3 restated). *Given $\delta \in (0,1)$ and $\epsilon > 0$, choose $R = \frac{\epsilon}{4K_0}$. Under $\mathcal{E}_0$, for any $\widetilde{\boldsymbol{x}}_k$ such that $\widetilde{\boldsymbol{x}}_0 = \boldsymbol{x}_0$ and $\|\widetilde{\boldsymbol{x}}_{k+1} - \widetilde{\boldsymbol{x}}_k\| = \eta$, where $\eta \leq \frac{\mu\epsilon}{8K_0 IC_{g_{xy}}}$, if we run Algorithm 2 with input $\{\widetilde{\boldsymbol{x}}_k\}_{k=1}^K$ and generate outputs $\{\widetilde{\boldsymbol{y}}_k\}_{k=1}^K$, and the fixed step-size $\gamma$ satisfies*

$$\gamma = \frac{\mu\epsilon^2}{512K_0^2\sigma_{g,1}^2\sqrt{\lambda+1}\left(\lambda+\sqrt{\lambda+1}\right)}, \tag{44}$$

*and the number of iterations $N$ for update in each period satisfies*

$$N = \frac{64^2 \sigma_{g,1}^2 K_0^2 \left(\lambda + \sqrt{\lambda+1}\right)^2}{\mu^2 \epsilon^2},\tag{45}$$

*where we set $\lambda$ to be*

$$\lambda = \max\left(\sqrt{3\ln\left(\frac{2K}{\delta I}\right)}, \ln\left(\frac{2K}{\delta I}\right)\right),\tag{46}$$

*then with probability at least $1 - \delta/2$ over randomness in $\sigma\left\{\bigcup_{k=1}^{K-1} \widetilde{\mathcal{F}}_k^N\right\}$ (this event is denoted as $\mathcal{E}_1$), we have $\|\boldsymbol{y}^*(\widetilde{\boldsymbol{x}}_k) - \widetilde{\boldsymbol{y}}_k\| \leq \epsilon/4K_0$ for any $k \geq 1$ in $O\left(K\sigma_{g,1}^2 K_0^2 \left(\lambda + \sqrt{\lambda+1}\right)^2 / I\mu^2\epsilon^2\right)$ iterations.*

***Proof of Lemma D.3.*** We denote $\widetilde{\boldsymbol{y}}_k^* = \boldsymbol{y}^*(\widetilde{\boldsymbol{x}}_k)$ for simplicity. By Lemma 3.2, under event $\mathcal{E}_0$, we have $\|\widetilde{\boldsymbol{y}}_0 - \widetilde{\boldsymbol{y}}_0^*\| \leq \frac{\epsilon}{8K_0}$. Suppose $1 \leq k \leq I$, then we have

$$\begin{aligned}
\|\widetilde{\boldsymbol{y}}_k - \widetilde{\boldsymbol{y}}_k^*\| = \|\widetilde{\boldsymbol{y}}_0 - \widetilde{\boldsymbol{y}}_k^*\| &\leq \|\widetilde{\boldsymbol{y}}_0 - \widetilde{\boldsymbol{y}}_0^*\| + \|\widetilde{\boldsymbol{y}}_0^* - \widetilde{\boldsymbol{y}}_k^*\| \\
&\leq \frac{\epsilon}{8K_0} + \sum_{i=0}^{k-1} \|\widetilde{\boldsymbol{y}}_i^* - \widetilde{\boldsymbol{y}}_{i+1}^*\| \overset{(i)}{\leq} \frac{\epsilon}{8K_0} + \sum_{i=0}^{k-1} \frac{C_{g_{xy}}}{\mu} \|\widetilde{\boldsymbol{x}}_i - \widetilde{\boldsymbol{x}}_{i+1}\| \\
&= \frac{\epsilon}{8K_0} + \frac{IC_{g_{xy}}}{\mu}\eta \overset{(ii)}{\leq} \frac{\epsilon}{4K_0},
\end{aligned}\tag{47}$$

where $(i)$ follows from $(a)$ in Lemma C.2 and $(ii)$ follows from $\eta \leq \frac{\mu\epsilon}{8K_0 IC_{g_{xy}}}$.

Now we proceed to update $\widetilde{\boldsymbol{y}}_k$ (with $\widetilde{\boldsymbol{y}}_k^0 = \widetilde{\boldsymbol{y}}_k$) at $k$-th iteration, where $k = I$. Following the same technique and proof as Lemma D.1, and note that $\frac{1}{2}\left(\frac{\epsilon}{4K_0}\right)^2$ is an upper bound for $\frac{1}{2}\|\widetilde{\boldsymbol{y}}_I - \widetilde{\boldsymbol{y}}_I^*\|^2$, then for $k = I$ and any $\lambda > 0$ we have

$$\Pr\left[g(\widetilde{\boldsymbol{x}}_k, \widetilde{\boldsymbol{y}}_{k+1}) - g(\widetilde{\boldsymbol{x}}_k, \widetilde{\boldsymbol{y}}_k^*) > \mathcal{P}_0(N) + \lambda\mathcal{P}_1(N)\right] \leq \exp(-\lambda^2/3) + \exp(-\lambda),$$

where $\widetilde{\boldsymbol{y}}_{k+1} = \frac{1}{N}\sum_{t=1}^{N} \widetilde{\boldsymbol{y}}_k^t$ (see line 8 in Algorithm 2). Also, by substituting $\frac{1}{2}\left(\frac{\epsilon}{4K_0}\right)^2$ for $D^2$ in definition (20) of Lemma D.1, we redefine $\mathcal{P}_0(N)$ and $\mathcal{P}_1(N)$ as

$$\begin{aligned}
\mathcal{P}_0(N) &:= \frac{1}{\gamma N}\left[\frac{1}{2}\left(\frac{\epsilon}{4K_0}\right)^2 + 2\sigma_{g,1}^2\gamma^2 N\right], \\
\mathcal{P}_1(N) &:= \frac{1}{\gamma N}\left[2\sqrt{2}\sqrt{\frac{1}{2}\left(\frac{\epsilon}{4K_0}\right)^2}\sigma_{g,1}\gamma\sqrt{N} + 2\sigma_{g,1}^2\gamma^2 N\right].
\end{aligned}\tag{48}$$

Set $\lambda$ such that $\exp(-\lambda^2/3) + \exp(-\lambda) \leq \frac{\delta I}{2K}$, then

$$\lambda = \max\left(\sqrt{3\ln\left(\frac{2K}{\delta I}\right)}, \ln\left(\frac{2K}{\delta I}\right)\right).$$

Thus under event $\mathcal{E}_0$, with probability at least $1 - \frac{\delta I}{2K}$ over randomness in $\widetilde{\mathcal{F}}_I^N$, we have

$$\begin{aligned}
g(\widetilde{\boldsymbol{x}}_k, &\widetilde{\boldsymbol{y}}_{k+1}) - g(\widetilde{\boldsymbol{x}}_k, \widetilde{\boldsymbol{y}}_k^*) \leq \mathcal{P}_0(N) + \lambda\mathcal{P}_1(N) \\
&= \frac{1}{\gamma N}\left[\frac{1}{2}\left(\frac{\epsilon}{4K_0}\right)^2 + 2\sigma_{g,1}^2\gamma^2 N\right] + \lambda\frac{1}{\gamma N}\left[2\sqrt{2}\sqrt{\frac{1}{2}\left(\frac{\epsilon}{4K_0}\right)^2}\sigma_{g,1}\gamma\sqrt{N} + 2\sigma_{g,1}^2\gamma^2 N\right] \\
&\leq \frac{\epsilon^2}{32\gamma K_0^2 N} + 2\sigma_{g,1}^2\gamma + \lambda\left(\frac{\epsilon\sigma_{g,1}}{2K_0\sqrt{N}} + 2\sigma_{g,1}^2\gamma\right) \\
&\leq \frac{\epsilon^2}{32\gamma K_0^2 N} + 2\sigma_{g,1}^2(\lambda+1)\gamma + \frac{\lambda\epsilon\sigma_{g,1}}{2K_0\sqrt{N}}
\end{aligned}\tag{49}$$

If we choose

$$\gamma = \frac{\epsilon}{8\sigma_{g,1}K_0\sqrt{\lambda+1}\sqrt{N}}, \quad N = \frac{64^2\sigma_{g,1}^2K_0^2\left(\lambda+\sqrt{\lambda+1}\right)^2}{\mu^2\epsilon^2}, \tag{50}$$

Then $N = \widetilde{O}(\sigma_{g,1}^2K_0^2/\mu^2\epsilon^2)$ and we have

$$\begin{aligned}
g(\widetilde{\boldsymbol{x}}_k, \widetilde{\boldsymbol{y}}_{k+1}) - g(\widetilde{\boldsymbol{x}}_k, \widetilde{\boldsymbol{y}}_k^*) &\leq \frac{\epsilon\sigma_{g,1}\sqrt{\lambda+1}}{2K_0\sqrt{N}} + \frac{\epsilon\sigma_{g,1}\lambda}{2K_0\sqrt{N}} \\
&\leq \frac{\epsilon\sigma_{g,1}\left(\lambda+\sqrt{\lambda+1}\right)}{2K_0\sqrt{N}} \\
&\leq \frac{\mu}{2}\left(\frac{\epsilon}{8K_0}\right)^2
\end{aligned}$$

By strong convexity of $g(\boldsymbol{x}, \boldsymbol{y})$ with respect to $\boldsymbol{y}$, we obtain

$$\frac{\mu}{2}\left\|\widetilde{\boldsymbol{y}}_{k+1} - \widetilde{\boldsymbol{y}}_k^*\right\|^2 \leq g(\widetilde{\boldsymbol{x}}_k, \widetilde{\boldsymbol{y}}_{k+1}) - g(\widetilde{\boldsymbol{x}}_k, \widetilde{\boldsymbol{y}}_k^*) \leq \frac{\mu}{2}\left(\frac{\epsilon}{8K_0}\right)^2, \tag{51}$$

which implies $\|\widetilde{\boldsymbol{y}}_{k+1} - \widetilde{\boldsymbol{y}}_k^*\| \leq \frac{\epsilon}{8K_0}$ for $k = I$. Thus for any $k$ such that $I+1 \leq k \leq 2I$, we have

$$\begin{aligned}
\left\|\widetilde{\boldsymbol{y}}_k - \widetilde{\boldsymbol{y}}_k^*\right\| = \left\|\widetilde{\boldsymbol{y}}_{I+1} - \widetilde{\boldsymbol{y}}_k^*\right\| &\leq \left\|\widetilde{\boldsymbol{y}}_{I+1} - \widetilde{\boldsymbol{y}}_I^*\right\| + \left\|\widetilde{\boldsymbol{y}}_I^* - \widetilde{\boldsymbol{y}}_k^*\right\| \\
&\leq \frac{\epsilon}{8K_0} + \sum_{i=I}^{k-1}\left\|\widetilde{\boldsymbol{y}}_i^* - \widetilde{\boldsymbol{y}}_{i+1}^*\right\| \overset{(i)}{\leq} \frac{\epsilon}{8K_0} + \sum_{i=I}^{k-1}\frac{C_{g_{xy}}}{\mu}\left\|\widetilde{\boldsymbol{x}}_i - \widetilde{\boldsymbol{x}}_{i+1}\right\| \\
&\leq \frac{\epsilon}{8K_0} + \frac{IC_{g_{xy}}}{\mu}\eta \\
&\overset{(ii)}{\leq} \frac{\epsilon}{4K_0}.
\end{aligned} \tag{52}$$

where $(i)$ follows from $(a)$ in Lemma C.2 and $(ii)$ follows from $\eta \leq \frac{\mu\epsilon}{8K_0IC_{g_{xy}}}$.

In general, for $k = jI$ where $1 \leq j \leq \left\lfloor\frac{K}{I}\right\rfloor$, we update $\widetilde{\boldsymbol{y}}_k$ by Algorithm 2. Under event $\mathcal{E}_0$, with probability at least $1 - \frac{j\delta I}{2K}$ we have $\|\widetilde{\boldsymbol{y}}_{k+1} - \widetilde{\boldsymbol{y}}_k^*\| \leq \frac{\epsilon}{8K_0}$ in at most

$$N = \frac{64^2\sigma_{g,1}^2K_0^2\left(\lambda+\sqrt{\lambda+1}\right)^2}{\mu^2\epsilon^2}$$

iterations, i.e., $O\left(\sigma_{g,1}^2K_0^2\left(\lambda+\sqrt{\lambda+1}\right)^2/\mu^2\epsilon^2\right)$ iterations. Also, by repeatedly applying (49) + (50) + (51) or (52) for all $k \geq 1$ and using union bound, under event $\mathcal{E}_0$ we have $\|\widetilde{\boldsymbol{y}}_k - \widetilde{\boldsymbol{y}}_k^*\| \leq \frac{\epsilon}{4K_0}$ with probability at least $1 - \delta/2$ for any $k \geq 1$. Moreover, we need to run Algorithm 2 for at most

$$\left\lfloor\frac{K}{I}\right\rfloor N \leq \frac{KN}{I} = \frac{64^2K\sigma_{g,1}^2K_0^2\left(\lambda+\sqrt{\lambda+1}\right)^2}{I\mu^2\epsilon^2} \tag{53}$$

iterations in total, where

$$\lambda = \max\left(\sqrt{3\ln\left(\frac{2K}{\delta I}\right)}, \ln\left(\frac{2K}{\delta I}\right)\right). \tag{54}$$

Therefore, the total number of iterations performed by Algorithm 2 is at most $O\left(K\sigma_{g,1}^2K_0^2\left(\lambda+\sqrt{\lambda+1}\right)^2/I\mu^2\epsilon^2\right)$. $\qquad\square$

## D.4   PROOF OF LEMMA 3.4

**Lemma D.4** (Error Control for the Lower-level Problem, Lemma 3.4 restated). *Under event $\mathcal{E} = \mathcal{E}_0 \cap \mathcal{E}_1$, we have $\|\boldsymbol{y}^*(\widetilde{\boldsymbol{x}}_k) - \widetilde{\boldsymbol{y}}_k\| \leq \epsilon/4K_0$ for any $k \geq 0$ and $Pr(\mathcal{E}) \geq 1 - \delta$ and the probability is taken over randomness in $\widetilde{\mathcal{F}}_K$.*

**Proof of Lemma D.4.** By Lemma 3.2 and 3.3, under event $\mathcal{E} = \mathcal{E}_0 \cap \mathcal{E}_1$, we have $\|\boldsymbol{y}^*(\widetilde{\boldsymbol{x}}_k) - \widetilde{\boldsymbol{y}}_k\| \leq \epsilon/8K_0$ for $k = 0$ and $\|\boldsymbol{y}^*(\widetilde{\boldsymbol{x}}_k) - \widetilde{\boldsymbol{y}}_k\| \leq \epsilon/4K_0$ for any $k \geq 1$. Therefore, under event $\mathcal{E}$ we have $\|\boldsymbol{y}^*(\widetilde{\boldsymbol{x}}_k) - \widetilde{\boldsymbol{y}}_k\| \leq \epsilon/4K_0$ for any $k \geq 0$. Moreover, we have

$$\Pr(\mathcal{E}) = \Pr(\mathcal{E}_0 \cap \mathcal{E}_1) = 1 - \Pr(\mathcal{E}_0 \cup \mathcal{E}_1) \geq 1 - [\Pr(\mathcal{E}_0) + \Pr(\mathcal{E}_1)] \geq 1 - \frac{\delta}{2} - \frac{\delta}{2} = 1 - \delta,$$

where the probability is taken over randomness in $\widetilde{\mathcal{F}}_K$. Thus we conclude our proof. $\square$

### D.5 AUXILIARY LEMMAS

In the next lemma, the goal is to bound accumulated expected error of $\|\boldsymbol{z}_k - \boldsymbol{z}_k^*\|$, which is similar to Lemma B.6 in Chen et al. (2023b).

**Lemma D.5.** *Suppose Assumptions 1, 2 and 3 hold. If we choose $\nu \leq \min\left(\frac{1}{4\mu}, \frac{\mu}{16\sigma_{g,2}^2}\right)$, then under event $\mathcal{E}$, we have*

$$
\begin{aligned}
\sum_{k=0}^{K-1} \mathbb{E}[\|\boldsymbol{z}_k - \boldsymbol{z}_k^*\|^2] &\leq \frac{\Delta_{\boldsymbol{z},0}}{\nu\mu} + \frac{5K}{\mu^2}\left(\frac{\rho^2 M^2}{\mu^2} + (L_{\boldsymbol{y},0} + L_{\boldsymbol{y},1}M)^2\right)\left(\frac{\epsilon}{4K_0}\right)^2 \\
&\quad + K\left[\frac{2}{\mu}\left(\frac{2M^2}{\mu^2}\sigma_{g,2}^2 + \sigma_{f,1}^2\right)\nu + \frac{4L_{\boldsymbol{z}^*}^2}{\mu^2}\frac{\eta^2}{\nu^2}\right],
\end{aligned}
\tag{55}
$$

*where we define $\Delta_{\boldsymbol{z},0} := \|\boldsymbol{z}_0 - \boldsymbol{z}_0^*\|^2$, and the expectation is taken over all randomness in $\mathcal{F}_K$.*

**Proof of Lemma D.5.** First, we have

$$
\begin{aligned}
\left\|\boldsymbol{z}_{k+1} - \boldsymbol{z}_{k+1}^*\right\|^2 &\overset{(i)}{\leq} \left(1 + \frac{\nu\mu}{3}\right)\|\boldsymbol{z}_{k+1} - \boldsymbol{z}_k^*\|^2 + \left(1 + \frac{3}{\nu\mu}\right)\left\|\boldsymbol{z}_{k+1}^* - \boldsymbol{z}_k^*\right\|^2 \\
&\overset{(ii)}{\leq} \left(1 + \frac{\nu\mu}{3}\right)\|\boldsymbol{z}_{k+1} - \boldsymbol{z}_k^*\|^2 + \left(1 + \frac{3}{\nu\mu}\right)L_{\boldsymbol{z}^*}^2\eta^2,
\end{aligned}
\tag{56}
$$

where $(i)$ follows from Young's inequality and $(ii)$ follows from $(b)$ in Lemma C.2 that $\boldsymbol{z}^*(\boldsymbol{x})$ is $L_{\boldsymbol{z}^*}$-Lipschitz. Then we decompose $\boldsymbol{z}_{k+1} - \boldsymbol{z}_k^*$ as follows,

$$
\begin{aligned}
\boldsymbol{z}_{k+1} - \boldsymbol{z}_k^* &= \boldsymbol{z}_k - \nu\left(\nabla_{\boldsymbol{y}}^2 G(\boldsymbol{x}_k, \boldsymbol{y}_k; \xi_k)\boldsymbol{z}_k - \nabla_{\boldsymbol{y}}F(\boldsymbol{x}_k, \boldsymbol{y}_k; \zeta_k)\right) - \boldsymbol{z}_k^* \\
&= \boldsymbol{z}_k - \nu\left(\nabla_{\boldsymbol{y}}^2 g(\boldsymbol{x}_k, \boldsymbol{y}_k)\boldsymbol{z}_k - \nabla_{\boldsymbol{y}}f(\boldsymbol{x}_k, \boldsymbol{y}_k)\right) - \boldsymbol{z}_k^* - \nu\left(\nabla_{\boldsymbol{y}}^2 G(\boldsymbol{x}_k, \boldsymbol{y}_k; \xi_k) - \nabla_{\boldsymbol{y}}^2 g(\boldsymbol{x}_k, \boldsymbol{y}_k)\right)\boldsymbol{z}_k \\
&\quad + \nu\left(\nabla_{\boldsymbol{y}}F(\boldsymbol{x}_k, \boldsymbol{y}_k; \zeta_k) - \nabla_{\boldsymbol{y}}f(\boldsymbol{x}_k, \boldsymbol{y}_k)\right).
\end{aligned}
$$

Take conditional expectation on $\mathcal{F}_k$ and we have

$$
\mathbb{E}_k[\|\boldsymbol{z}_{k+1} - \boldsymbol{z}_k^*\|^2]
$$

$$
\overset{(i)}{=} \|\boldsymbol{z}_k - \nu(\nabla_{\boldsymbol{y}}^2 g(\boldsymbol{x}_k, \boldsymbol{y}_k)\boldsymbol{z}_k - \nabla_{\boldsymbol{y}} f(\boldsymbol{x}_k, \boldsymbol{y}_k)) - \boldsymbol{z}_k^*\|^2 + \nu^2 \sigma_{g,2}^2 \|\boldsymbol{z}_k\|^2 + \nu^2 \sigma_{f,1}^2
$$

$$
\leq \|(I - \nu\nabla_{\boldsymbol{y}}^2 g(\boldsymbol{x}_k, \boldsymbol{y}_k))(\boldsymbol{z}_k - \boldsymbol{z}_k^*) - \nu(\nabla_{\boldsymbol{y}}^2 g(\boldsymbol{x}_k, \boldsymbol{y}_k)\boldsymbol{z}_k^* - \nabla_{\boldsymbol{y}} f(\boldsymbol{x}_k, \boldsymbol{y}_k))\|^2
$$

$$
\quad + 2\nu^2 \sigma_{g,2}^2 (\|\boldsymbol{z}_k - \boldsymbol{z}_k^*\|^2 + \|\boldsymbol{z}_k^*\|^2) + \nu^2 \sigma_{f,1}^2
$$

$$
\overset{(ii)}{\leq} (1 + \frac{\nu\mu}{2})\|(I - \nu\nabla_{\boldsymbol{y}}^2 g(\boldsymbol{x}_k, \boldsymbol{y}_k))(\boldsymbol{z}_k - \boldsymbol{z}_k^*)\|^2 + 2\nu^2 \sigma_{g,2}^2 (\|\boldsymbol{z}_k - \boldsymbol{z}_k^*\|^2 + \|\boldsymbol{z}_k^*\|^2) + \nu^2 \sigma_{f,1}^2
$$

$$
\quad + (1 + \frac{2}{\nu\mu})\|\nu(\nabla_{\boldsymbol{y}}^2 g(\boldsymbol{x}_k, \boldsymbol{y}_k)\boldsymbol{z}_k^* - \nabla_{\boldsymbol{y}}^2 g(\boldsymbol{x}_k, \boldsymbol{y}_k^*)\boldsymbol{z}_k^* + \nabla_{\boldsymbol{y}} f(\boldsymbol{x}_k, \boldsymbol{y}_k^*) - \nabla_{\boldsymbol{y}} f(\boldsymbol{x}_k, \boldsymbol{y}_k))\|^2
$$

$$
\leq ((1 + \frac{\nu\mu}{2})(1 - \nu\mu)^2 + 2\nu^2 \sigma_{g,2}^2)\|\boldsymbol{z}_k - \boldsymbol{z}_k^*\|^2 + 2\nu^2 \sigma_{g,2}^2 \|\boldsymbol{z}_k^*\|^2 + \nu^2 \sigma_{f,1}^2
$$

$$
\quad + 2\nu^2(1 + \frac{2}{\nu\mu})(\|\nabla_{\boldsymbol{y}}^2 g(\boldsymbol{x}_k, \boldsymbol{y}_k) - \nabla_{\boldsymbol{y}}^2 g(\boldsymbol{x}_k, \boldsymbol{y}_k^*)\|^2 \|\boldsymbol{z}_k^*\|^2 + \|\nabla_{\boldsymbol{y}} f(\boldsymbol{x}_k, \boldsymbol{y}_k^*) - \nabla_{\boldsymbol{y}} f(\boldsymbol{x}_k, \boldsymbol{y}_k)\|^2)
$$

$$
\leq ((1 + \frac{\nu\mu}{2})(1 - \nu\mu)^2 + 2\nu^2 \sigma_{g,2}^2)\|\boldsymbol{z}_k - \boldsymbol{z}_k^*\|^2 + 2\nu^2 \sigma_{g,2}^2 \|\boldsymbol{z}_k^*\|^2 + \nu^2 \sigma_{f,1}^2
$$

$$
\quad + (2\nu^2 + \frac{4\nu}{\mu})(\rho^2\|\boldsymbol{z}_k^*\|^2 + (L_{\boldsymbol{y},0} + L_{\boldsymbol{y},1}M)^2)\|\boldsymbol{y}_k - \boldsymbol{y}_k^*\|^2
$$

$$
\leq ((1 + \frac{\nu\mu}{2})(1 - \nu\mu)^2 + 2\nu^2 \sigma_{g,2}^2)\|\boldsymbol{z}_k - \boldsymbol{z}_k^*\|^2 + (\frac{2M^2}{\mu^2}\sigma_{g,2}^2 + \sigma_{f,1}^2)\nu^2
$$

$$
\quad + (2\nu^2 + \frac{4\nu}{\mu})(\frac{\rho^2 M^2}{\mu^2} + (L_{\boldsymbol{y},0} + L_{\boldsymbol{y},1}M)^2)\|\boldsymbol{y}_k - \boldsymbol{y}_k^*\|^2
$$

$$
\overset{(iii)}{\leq} (1 - \frac{4\nu\mu}{3})\|\boldsymbol{z}_k - \boldsymbol{z}_k^*\|^2 + (\frac{2M^2}{\mu^2}\sigma_{g,2}^2 + \sigma_{f,1}^2)\nu^2 + (2\nu^2 + \frac{4\nu}{\mu})(\frac{\rho^2 M^2}{\mu^2} + (L_{\boldsymbol{y},0} + L_{\boldsymbol{y},1}M)^2)\|\boldsymbol{y}_k - \boldsymbol{y}_k^*\|,
$$
(57)

where $(i)$ follows from Assumption 3 and the fact that stochastic estimators are unbiased, $(ii)$ follows from Young's inequality and the definition of linear system solution $\boldsymbol{z}_k^*$, i.e., $\nabla_{\boldsymbol{y}}^2 g(\boldsymbol{x}_k, \boldsymbol{y}_k^*)\boldsymbol{z}_k^* = \nabla_{\boldsymbol{y}} f(\boldsymbol{x}_k, \boldsymbol{y}_k^*)$. Inequality $(iii)$ holds since we choose $\nu \leq \min(\frac{1}{4\mu}, \frac{\mu}{16\sigma_{g,2}^2})$ and thus we have

$$
\left(1 + \frac{\nu\mu}{2}\right)(1 - \nu\mu)^2 + 2\nu^2 \sigma_{g,2}^2 = 1 - \frac{3}{2}\nu\mu + \frac{1}{2}\nu^3\mu^3 + 2\nu^2 \sigma_{g,2}^2 \overset{(i)}{\leq} 1 - \frac{3}{2}\nu\mu + \frac{1}{32}\nu\mu + \frac{1}{8}\nu\mu
$$

$$
= 1 - \frac{43}{32}\nu\mu \leq 1 - \frac{4}{3}\nu\mu,
$$

where $(i)$ follows from $\nu^2\mu^2 \leq 1/16$ and $\nu\sigma_{g,2}^2 \leq \mu/16$. Plug the inequality (57) into (56), then take conditional expectation on $\mathcal{F}_k$ for both sides of (56) and we obtain

$$
\mathbb{E}_k[\|\boldsymbol{z}_{k+1} - \boldsymbol{z}_{k+1}^*\|^2]
$$

$$
\leq (1 + \frac{\nu\mu}{3})\mathbb{E}_k[\|\boldsymbol{z}_{k+1} - \boldsymbol{z}_k^*\|^2] + (1 + \frac{3}{\nu\mu})L_{\boldsymbol{z}^*}^2 \eta^2
$$

$$
\leq (1 + \frac{\nu\mu}{3})[(1 - \frac{4\nu\mu}{3})\|\boldsymbol{z}_k - \boldsymbol{z}_k^*\|^2 + (2\nu^2 + \frac{4\nu}{\mu})(\frac{\rho^2 M^2}{\mu^2} + (L_{\boldsymbol{y},0} + L_{\boldsymbol{y},1}M)^2)\|\boldsymbol{y}_k - \boldsymbol{y}_k^*\|^2]
$$

$$
\quad + (1 + \frac{\nu\mu}{3})(\frac{2M^2}{\mu^2}\sigma_{g,2}^2 + \sigma_{f,1}^2)\nu^2 + (1 + \frac{3}{\nu\mu})L_{\boldsymbol{z}^*}^2 \eta^2
$$

$$
\leq (1 - \nu\mu)\|\boldsymbol{z}_k - \boldsymbol{z}_k^*\|^2 + (\frac{2\nu^3\mu}{3} + \frac{4\nu}{\mu} + \frac{10\nu^2}{3})(\frac{\rho^2 M^2}{\mu^2} + (L_{\boldsymbol{y},0} + L_{\boldsymbol{y},1}M)^2)\|\boldsymbol{y}_k - \boldsymbol{y}_k^*\|^2
$$

$$
\quad + (1 + \frac{\nu\mu}{3})(\frac{2M^2}{\mu^2}\sigma_{g,2}^2 + \sigma_{f,1}^2)\nu^2 + (1 + \frac{3}{\nu\mu})L_{\boldsymbol{z}^*}^2 \eta^2
$$

$$
\overset{(i)}{\leq} (1 - \nu\mu)\|\boldsymbol{z}_k - \boldsymbol{z}_k^*\|^2 + \frac{5\nu}{\mu}(\frac{\rho^2 M^2}{\mu^2} + (L_{\boldsymbol{y},0} + L_{\boldsymbol{y},1}M)^2)\|\boldsymbol{y}_k - \boldsymbol{y}_k^*\|^2 + 2(\frac{2M^2}{\mu^2}\sigma_{g,2}^2 + \sigma_{f,1}^2)\nu^2 + \frac{4}{\nu\mu}L_{\boldsymbol{z}^*}^2 \eta^2
$$

$$
\overset{(ii)}{\leq} (1 - \nu\mu)\|\boldsymbol{z}_k - \boldsymbol{z}_k^*\|^2 + \frac{5\nu}{\mu}(\frac{\rho^2 M^2}{\mu^2} + (L_{\boldsymbol{y},0} + L_{\boldsymbol{y},1}M)^2)(\frac{\epsilon}{4K_0})^2 + 2(\frac{2M^2}{\mu^2}\sigma_{g,2}^2 + \sigma_{f,1}^2)\nu^2 + \frac{4}{\nu\mu}L_{\boldsymbol{z}^*}^2 \eta^2
$$

where $(i)$ follows by $\nu \leq \frac{1}{4\mu}$ and $(ii)$ follows from Lemma 3.4. Take expectation with respect to $\mathcal{F}_k$ and we have

$$
\mathbb{E}[\|\boldsymbol{z}_k - \boldsymbol{z}_k^*\|^2] = \mathbb{E}_{\mathcal{F}_k}[\mathbb{E}_k[\|\boldsymbol{z}_k - \boldsymbol{z}_k^*\|^2]]
$$

$$
\leq (1-\nu\mu)^k \|\boldsymbol{z}_0 - \boldsymbol{z}_0^*\|^2 + \sum_{i=0}^{k-1} (1-\nu\mu)^{k-i-1} \left[ \frac{5\nu}{\mu} \left( \frac{\rho^2 M^2}{\mu^2} + (L_{\boldsymbol{y},0} + L_{\boldsymbol{y},1} M)^2 \right) \left( \frac{\epsilon}{4K_0} \right)^2 \right.
$$

$$
\left. + 2 \left( \frac{2M^2}{\mu^2} \sigma_{g,2}^2 + \sigma_{f,1}^2 \right) \nu^2 + \frac{4}{\nu\mu} L_{\boldsymbol{z}^*}^2 \eta^2 \right].
$$

Take summation on both sides and we finally conclude

$$
\sum_{k=0}^{K-1} \mathbb{E}[\|\boldsymbol{z}_k - \boldsymbol{z}_k^*\|^2]
$$

$$
\leq \sum_{k=0}^{K-1} (1-\nu\mu)^k \|\boldsymbol{z}_0 - \boldsymbol{z}_0^*\|^2 + \sum_{k=0}^{K-1} \sum_{i=0}^{k-1} (1-\nu\mu)^{k-i-1} \left[ \frac{5\nu}{\mu} \left( \frac{\rho^2 M^2}{\mu^2} + (L_{\boldsymbol{y},0} + L_{\boldsymbol{y},1} M)^2 \right) \left( \frac{\epsilon}{4K_0} \right)^2 \right.
$$

$$
\left. + 2 \left( \frac{2M^2}{\mu^2} \sigma_{g,2}^2 + \sigma_{f,1}^2 \right) \nu^2 + \frac{4}{\nu\mu} L_{\boldsymbol{z}^*}^2 \eta^2 \right]
$$

$$
\leq \frac{1}{\nu\mu} \|\boldsymbol{z}_0 - \boldsymbol{z}_0^*\|^2 + \frac{1}{\nu\mu} \sum_{k=0}^{K-1} \left[ \frac{5\nu}{\mu} \left( \frac{\rho^2 M^2}{\mu^2} + (L_{\boldsymbol{y},0} + L_{\boldsymbol{y},1} M)^2 \right) \left( \frac{\epsilon}{4K_0} \right)^2 \right.
$$

$$
\left. + 2 \left( \frac{2M^2}{\mu^2} \sigma_{g,2}^2 + \sigma_{f,1}^2 \right) \nu^2 + \frac{4}{\nu\mu} L_{\boldsymbol{z}^*}^2 \eta^2 \right]
$$

$$
\overset{(i)}{\leq} \frac{\Delta_{\boldsymbol{z},0}}{\nu\mu} + K \left[ \frac{5}{\mu^2} \left( \frac{\rho^2 M^2}{\mu^2} + (L_{\boldsymbol{y},0} + L_{\boldsymbol{y},1} M)^2 \right) \left( \frac{\epsilon}{4K_0} \right)^2 + \frac{2}{\mu} \left( \frac{2M^2}{\mu^2} \sigma_{g,2}^2 + \sigma_{f,1}^2 \right) \nu + \frac{4L_{\boldsymbol{z}^*}^2}{\mu^2} \frac{\eta^2}{\nu^2} \right],
$$

where $(i)$ follows from the definition of $\|\boldsymbol{z}_0 - \boldsymbol{z}_0^*\|^2$. $\qquad\square$

For simplicity, we denote hypergradient estimator as the following:

$$
\widehat{\nabla}\Phi(\boldsymbol{x}_k, \boldsymbol{y}_k, \boldsymbol{z}_k; \zeta_k, \xi_k) \coloneqq \nabla_{\boldsymbol{x}} F(\boldsymbol{x}_k, \boldsymbol{y}_k; \zeta_k) - \nabla_{\boldsymbol{x}} \nabla_{\boldsymbol{y}} G(\boldsymbol{x}_k, \boldsymbol{y}_k; \xi_k) \boldsymbol{z}_k. \tag{58}
$$

In the following lemma, we bound the variance of the hypergradient estimator.

**Lemma D.6.** *Suppose Assumptions 1, 2 and 3 hold. Then under event $\mathcal{E}$, we have*

$$
\mathbb{E}[\|\widehat{\nabla}\Phi(\boldsymbol{x}_k, \boldsymbol{y}_k, \boldsymbol{z}_k; \zeta_k, \xi_k) - \mathbb{E}_k[\widehat{\nabla}\Phi(\boldsymbol{x}_k, \boldsymbol{y}_k, \boldsymbol{z}_k; \zeta_k, \xi_k)]\|^2] \leq \sigma_{f,1}^2 + \frac{2M^2}{\mu^2} \sigma_{g,2}^2 + 2\sigma_{g,2}^2 \mathbb{E}[\|\boldsymbol{z}_k - \boldsymbol{z}_k^*\|^2],
$$

$$\tag{59}$$

*where the expectation is taken over all randomness in $\mathcal{F}_K$.*

***Proof of Lemma D.6.*** We first decompose $\widehat{\nabla}\Phi(\boldsymbol{x}_k, \boldsymbol{y}_k, \boldsymbol{z}_k; \zeta_k, \xi_k) - \mathbb{E}_k[\widehat{\nabla}\Phi(\boldsymbol{x}_k, \boldsymbol{y}_k, \boldsymbol{z}_k; \zeta_k, \xi_k)]$ as follows,

$$
\widehat{\nabla}\Phi(\boldsymbol{x}_k, \boldsymbol{y}_k, \boldsymbol{z}_k; \zeta_k, \xi_k) - \mathbb{E}_k[\widehat{\nabla}\Phi(\boldsymbol{x}_k, \boldsymbol{y}_k, \boldsymbol{z}_k; \zeta_k, \xi_k)]
$$

$$
= [\nabla_{\boldsymbol{x}} F(\boldsymbol{x}_k, \boldsymbol{y}_k; \zeta_k) - \nabla_{\boldsymbol{x}} \nabla_{\boldsymbol{y}} G(\boldsymbol{x}_k, \boldsymbol{y}_k; \xi_k) \boldsymbol{z}_k] - [\nabla_{\boldsymbol{x}} f(\boldsymbol{x}_k, \boldsymbol{y}_k) - \nabla_{\boldsymbol{x}} \nabla_{\boldsymbol{y}} g(\boldsymbol{x}_k, \boldsymbol{y}_k) \boldsymbol{z}_k]
$$

$$
= [\nabla_{\boldsymbol{x}} F(\boldsymbol{x}_k, \boldsymbol{y}_k; \zeta_k) - \nabla_{\boldsymbol{x}} f(\boldsymbol{x}_k, \boldsymbol{y}_k)] - [\nabla_{\boldsymbol{x}} \nabla_{\boldsymbol{y}} G(\boldsymbol{x}_k, \boldsymbol{y}_k; \xi_k) - \nabla_{\boldsymbol{x}} \nabla_{\boldsymbol{y}} g(\boldsymbol{x}_k, \boldsymbol{y}_k)] \boldsymbol{z}_k.
$$

Take conditional expectation on $\mathcal{F}_k$ and we have

$$
\mathbb{E}_k[\|\widehat{\nabla}\Phi(\boldsymbol{x}_k, \boldsymbol{y}_k, \boldsymbol{z}_k; \zeta_k, \xi_k) - \mathbb{E}_k[\widehat{\nabla}\Phi(\boldsymbol{x}_k, \boldsymbol{y}_k, \boldsymbol{z}_k; \zeta_k, \xi_k)]\|^2]
$$

$$
\overset{(i)}{=} \mathbb{E}_k[\|\nabla_{\boldsymbol{x}} F(\boldsymbol{x}_k, \boldsymbol{y}_k; \zeta_k) - \nabla_{\boldsymbol{x}} f(\boldsymbol{x}_k, \boldsymbol{y}_k)\|^2] + \mathbb{E}_k[\|\nabla_{\boldsymbol{x}} \nabla_{\boldsymbol{y}} G(\boldsymbol{x}_k, \boldsymbol{y}_k; \xi_k) - \nabla_{\boldsymbol{x}} \nabla_{\boldsymbol{y}} g(\boldsymbol{x}_k, \boldsymbol{y}_k)\|^2] \|\boldsymbol{z}_k\|^2
$$

$$
\leq \sigma_{f,1}^2 + \sigma_{g,2}^2 \|\boldsymbol{z}_k\|^2
$$

$$
\leq \sigma_{f,1}^2 + 2\sigma_{g,2}^2 \|\boldsymbol{z}_k - \boldsymbol{z}_k^*\|^2 + 2\sigma_{g,2}^2 \|\boldsymbol{z}_k^*\|^2
$$

$$
\leq \sigma_{f,1}^2 + \frac{2M^2}{\mu^2} \sigma_{g,2}^2 + 2\sigma_{g,2}^2 \|\boldsymbol{z}_k - \boldsymbol{z}_k^*\|^2,
$$

$$\tag{60}$$

where $(i)$ follows from Assumption 3 that stochastic estimators are unbiased and $\xi_k$, $\zeta_k$ are mutually independent. Take expectation with respect to $\mathcal{F}_k$ on both sides of (60), and then we have

$$
\begin{aligned}
&\mathbb{E}[\|\widehat{\nabla}\Phi(\boldsymbol{x}_k, \boldsymbol{y}_k, \boldsymbol{z}_k; \zeta_k, \xi_k) - \mathbb{E}_k[\widehat{\nabla}\Phi(\boldsymbol{x}_k, \boldsymbol{y}_k, \boldsymbol{z}_k; \zeta_k, \xi_k)]\|^2] \\
&\quad = \mathbb{E}_{\mathcal{F}_k}[\mathbb{E}_k[\|\widehat{\nabla}\Phi(\boldsymbol{x}_k, \boldsymbol{y}_k, \boldsymbol{z}_k; \zeta_k, \xi_k) - \mathbb{E}_k[\widehat{\nabla}\Phi(\boldsymbol{x}_k, \boldsymbol{y}_k, \boldsymbol{z}_k; \zeta_k, \xi_k)]\|^2]] \\
&\quad \leq \sigma_{f,1}^2 + \frac{2M^2}{\mu^2}\sigma_{g,2}^2 + 2\sigma_{g,2}^2\mathbb{E}[\|\boldsymbol{z}_k - \boldsymbol{z}_k^*\|^2].
\end{aligned}
\tag{61}
$$

$\square$

## D.6 PROOF OF LEMMA 3.5

**Lemma D.7** (Bias of the Hypergradient Estimator, Lemma 3.5 restated). *Suppose Assumptions 1, 2 and 3 hold. Then under event $\mathcal{E}$, we have*

$$
\begin{aligned}
&\left\|\mathbb{E}_k[\widehat{\nabla}\Phi(\boldsymbol{x}_k, \boldsymbol{y}_k, \boldsymbol{z}_k; \zeta_k, \xi_k)] - \nabla\Phi(\boldsymbol{x}_k)\right\| \\
&\quad \leq \frac{L_{\boldsymbol{x},1}\epsilon}{4K_0}\|\nabla\Phi(\boldsymbol{x}_k)\| + \left(L_{\boldsymbol{x},0} + L_{\boldsymbol{x},1}\frac{C_{g_{xy}}M}{\mu} + \frac{\tau M}{\mu}\right)\frac{\epsilon}{4K_0} + C_{g_{xy}}\|\boldsymbol{z}_k - \boldsymbol{z}_k^*\|.
\end{aligned}
\tag{62}
$$

***Proof of Lemma D.7.*** We first decompose $\mathbb{E}_k[\widehat{\nabla}\Phi(\boldsymbol{x}_k, \boldsymbol{y}_k, \boldsymbol{z}_k; \zeta_k, \xi_k)] - \nabla\Phi(\boldsymbol{x}_k)$ as follows,

$$
\begin{aligned}
&\mathbb{E}_k[\widehat{\nabla}\Phi(\boldsymbol{x}_k, \boldsymbol{y}_k, \boldsymbol{z}_k; \zeta_k, \xi_k)] - \nabla\Phi(\boldsymbol{x}_k) \\
&= \mathbb{E}_k\left[\nabla_{\boldsymbol{x}}F(\boldsymbol{x}_k, \boldsymbol{y}_k; \zeta_k) - \nabla_{\boldsymbol{x}}\nabla_{\boldsymbol{y}}G(\boldsymbol{x}_k, \boldsymbol{y}_k; \xi_k)\boldsymbol{z}_k\right] - \nabla\Phi(\boldsymbol{x}_k) \\
&= (\nabla_{\boldsymbol{x}}f(\boldsymbol{x}_k, \boldsymbol{y}_k) - \nabla_{\boldsymbol{x}}f(\boldsymbol{x}_k, \boldsymbol{y}_k^*)) - \nabla_{\boldsymbol{x}}\nabla_{\boldsymbol{y}}g(\boldsymbol{x}_k, \boldsymbol{y}_k)(\boldsymbol{z}_k - \boldsymbol{z}_k^*) - (\nabla_{\boldsymbol{x}}\nabla_{\boldsymbol{y}}g(\boldsymbol{x}_k, \boldsymbol{y}_k) - \nabla_{\boldsymbol{x}}\nabla_{\boldsymbol{y}}g(\boldsymbol{x}_k, \boldsymbol{y}_k^*))\boldsymbol{z}_k^*.
\end{aligned}
$$

Then we obtain

$$
\begin{aligned}
&\left\|\mathbb{E}_k[\widehat{\nabla}\Phi(\boldsymbol{x}_k, \boldsymbol{y}_k, \boldsymbol{z}_k; \zeta_k, \xi_k)] - \nabla\Phi(\boldsymbol{x}_k)\right\| \\
&= \|(\nabla_{\boldsymbol{x}}f(\boldsymbol{x}_k, \boldsymbol{y}_k) - \nabla_{\boldsymbol{x}}f(\boldsymbol{x}_k, \boldsymbol{y}_k^*)) - \nabla_{\boldsymbol{x}}\nabla_{\boldsymbol{y}}g(\boldsymbol{x}_k, \boldsymbol{y}_k)(\boldsymbol{z}_k - \boldsymbol{z}_k^*) - (\nabla_{\boldsymbol{x}}\nabla_{\boldsymbol{y}}g(\boldsymbol{x}_k, \boldsymbol{y}_k) - \nabla_{\boldsymbol{x}}\nabla_{\boldsymbol{y}}g(\boldsymbol{x}_k, \boldsymbol{y}_k^*))\boldsymbol{z}_k^*\| \\
&\overset{(i)}{\leq} (L_{\boldsymbol{x},0} + L_{\boldsymbol{x},1}\|\nabla_{\boldsymbol{x}}f(\boldsymbol{x}_k, \boldsymbol{y}_k^*)\|)\|\boldsymbol{y}_k - \boldsymbol{y}_k^*\| + C_{g_{xy}}\|\boldsymbol{z}_k - \boldsymbol{z}_k^*\| + \tau\|\boldsymbol{y}_k - \boldsymbol{y}_k^*\|\|\boldsymbol{z}_k^*\| \\
&\overset{(ii)}{\leq} \left(L_{\boldsymbol{x},0} + L_{\boldsymbol{x},1}\left(\frac{C_{g_{xy}}M}{\mu} + \|\nabla\Phi(\boldsymbol{x}_k)\|\right)\right)\|\boldsymbol{y}_k - \boldsymbol{y}_k^*\| + C_{g_{xy}}\|\boldsymbol{z}_k - \boldsymbol{z}_k^*\| + \frac{\tau M}{\mu}\|\boldsymbol{y}_k - \boldsymbol{y}_k^*\| \\
&\leq L_{\boldsymbol{x},1}\|\boldsymbol{y}_k - \boldsymbol{y}_k^*\|\|\nabla\Phi(\boldsymbol{x}_k)\| + \left(L_{\boldsymbol{x},0} + L_{\boldsymbol{x},1}\frac{C_{g_{xy}}M}{\mu} + \frac{\tau M}{\mu}\right)\|\boldsymbol{y}_k - \boldsymbol{y}_k^*\| + C_{g_{xy}}\|\boldsymbol{z}_k - \boldsymbol{z}_k^*\| \\
&\overset{(iii)}{\leq} \frac{L_{\boldsymbol{x},1}\epsilon}{4K_0}\|\nabla\Phi(\boldsymbol{x}_k)\| + \left(L_{\boldsymbol{x},0} + L_{\boldsymbol{x},1}\frac{C_{g_{xy}}M}{\mu} + \frac{\tau M}{\mu}\right)\frac{\epsilon}{4K_0} + C_{g_{xy}}\|\boldsymbol{z}_k - \boldsymbol{z}_k^*\|,
\end{aligned}
$$

where $(i)$ follows from Assumption 1, $(ii)$ follows from $(c)$ in Lemma C.2 and $(iii)$ follows from Lemma 3.4. $\square$

## D.7 PROOF OF LEMMA 3.6

**Lemma D.8** (Expected Error of the Moving-Average Hypergradient Estimator, Lemma 3.6 restated). *Suppose Assumptions 1, 2 and 3 hold. Define $\boldsymbol{\delta}_k := \boldsymbol{m}_{k+1} - \nabla\Phi(\boldsymbol{x}_k)$ to be the moving-average estimation error. Then under event $\mathcal{E}$, we have*

$$
\mathbb{E}\left[\sum_{k=0}^{K-1}\|\boldsymbol{\delta}_k\|\right] \leq Err_1 + Err_2,
\tag{63}
$$

*where the expectation is taken over randomness in $\mathcal{F}_K$, and $Err_1$, $Err_2$ are defined as*

$$
Err_1 := \frac{L_{\boldsymbol{x},1}\epsilon}{4K_0} \sum_{k=0}^{K-1} \|\nabla\Phi(\boldsymbol{x}_k)\| + K\left(L_{\boldsymbol{x},0} + L_{\boldsymbol{x},1}\frac{C_{g_{xy}}M}{\mu} + \frac{\tau M}{\mu}\right)\frac{\epsilon}{4K_0} + C_{g_{xy}}\sqrt{K}\sqrt{\sum_{k=0}^{K-1}\mathbb{E}[\|\boldsymbol{z}_k - \boldsymbol{z}_k^*\|^2]},
$$

$$
Err_2 := K\sqrt{1-\beta}\sqrt{\sigma_{f,1}^2 + \frac{2M^2}{\mu^2}\sigma_{g,2}^2} + \sqrt{2}\sigma_{g,2}\sqrt{1-\beta}\sqrt{K}\sqrt{\sum_{k=0}^{K-1}\mathbb{E}[\|\boldsymbol{z}_k - \boldsymbol{z}_k^*\|^2]}
$$
$$
+ \frac{K_1\eta\beta}{1-\beta}\sum_{k=0}^{K-1}\|\nabla\Phi(\boldsymbol{x}_k)\| + \frac{K_0K\eta\beta}{1-\beta} + \frac{\beta}{1-\beta}\|\boldsymbol{m}_0 - \nabla\Phi(\boldsymbol{x}_0)\|.
$$

$$(64)$$

***Proof of Lemma D.8.*** First we denote

$$
\boldsymbol{\delta}_k := \boldsymbol{m}_{k+1} - \nabla\Phi(\boldsymbol{x}_k), \quad \widehat{\boldsymbol{\delta}}_k := \widehat{\nabla}\Phi(\boldsymbol{x}_k, \boldsymbol{y}_k, \boldsymbol{z}_k; \zeta_k, \xi_k) - \nabla\Phi(\boldsymbol{x}_k), \quad S(\boldsymbol{a}, \boldsymbol{b}) := \nabla\Phi(\boldsymbol{a}) - \nabla\Phi(\boldsymbol{b}).
$$

We can upper bound $S(\boldsymbol{a}, \boldsymbol{b})$ using the definition of $(K_0, K_1)$-smoothness,

$$
S(\boldsymbol{a}, \boldsymbol{b}) \le (K_0 + K_1\|\nabla\Phi(\boldsymbol{a})\|)\|\boldsymbol{a} - \boldsymbol{b}\|. \tag{65}
$$

By definition of $\boldsymbol{m}_k$ and $S(\boldsymbol{a}, \boldsymbol{b})$, we can get a recursive formula on $\delta_k$,

$$
\begin{aligned}
\boldsymbol{\delta}_{k+1} &= \boldsymbol{m}_{k+2} - \nabla\Phi(\boldsymbol{x}_{k+1}) \\
&= \beta\boldsymbol{m}_{k+1} + (1-\beta)\widehat{\nabla}\Phi(\boldsymbol{x}_{k+1}, \boldsymbol{y}_{k+1}, \boldsymbol{z}_{k+1}; \zeta_{k+1}, \xi_{k+1}) - \nabla\Phi(\boldsymbol{x}_{k+1}) \\
&= \beta\left(\boldsymbol{m}_{k+1} - \nabla\Phi(\boldsymbol{x}_k)\right) + \beta\left(\nabla\Phi(\boldsymbol{x}_k) - \nabla\Phi(\boldsymbol{x}_{k+1})\right) + (1-\beta)(\widehat{\nabla}\Phi(\boldsymbol{x}_{k+1}, \boldsymbol{y}_{k+1}, \boldsymbol{z}_{k+1}; \zeta_{k+1}, \xi_{k+1}) - \nabla\Phi(\boldsymbol{x}_{k+1})) \\
&= \beta\boldsymbol{\delta}_k + \beta S(\boldsymbol{x}_k, \boldsymbol{x}_{k+1}) + (1-\beta)\widehat{\boldsymbol{\delta}}_{k+1}.
\end{aligned}
$$

$$(66)$$

Apply (66) recursively and we obtain

$$
\begin{aligned}
\boldsymbol{\delta}_k &= \beta^k\boldsymbol{\delta}_0 + \beta\sum_{i=0}^{k-1}\beta^{k-1-i}S(\boldsymbol{x}_i, \boldsymbol{x}_{i+1}) + (1-\beta)\sum_{i=0}^{k-1}\beta^{k-1-i}\widehat{\boldsymbol{\delta}}_{i+1} \\
&= \beta^k\left(\boldsymbol{m}_1 - \nabla\Phi(\boldsymbol{x}_0)\right) + \beta\sum_{i=0}^{k-1}\beta^{k-1-i}S(\boldsymbol{x}_i, \boldsymbol{x}_{i+1}) + (1-\beta)\sum_{i=0}^{k-1}\beta^{k-1-i}\widehat{\boldsymbol{\delta}}_{i+1} \\
&= \beta^k\left(\beta\boldsymbol{m}_0 + (1-\beta)\widehat{\nabla}\Phi(\boldsymbol{x}_0, \boldsymbol{y}_0, \boldsymbol{z}_0; \zeta_0, \xi_0) - \nabla\Phi(\boldsymbol{x}_0)\right) + \beta\sum_{i=0}^{k-1}\beta^{k-1-i}S(\boldsymbol{x}_i, \boldsymbol{x}_{i+1}) + (1-\beta)\sum_{i=0}^{k-1}\beta^{k-1-i}\widehat{\boldsymbol{\delta}}_{i+1} \\
&= \beta^{k+1}\left(\boldsymbol{m}_0 - \nabla\Phi(\boldsymbol{x}_0)\right) + (1-\beta)\beta^k\widehat{\boldsymbol{\delta}}_0 + \beta\sum_{i=0}^{k-1}\beta^{k-1-i}S(\boldsymbol{x}_i, \boldsymbol{x}_{i+1}) + (1-\beta)\sum_{i=0}^{k-1}\beta^{k-1-i}\widehat{\boldsymbol{\delta}}_{i+1} \\
&= \beta^{k+1}\left(\boldsymbol{m}_0 - \nabla\Phi(\boldsymbol{x}_0)\right) + \beta\sum_{i=0}^{k-1}\beta^{k-1-i}S(\boldsymbol{x}_i, \boldsymbol{x}_{i+1}) + (1-\beta)\sum_{i=0}^{k}\beta^{k-i}\widehat{\boldsymbol{\delta}}_i.
\end{aligned}
$$

Using triangle inequality and plugging (65) into the above inequality, we have

$$
\|\boldsymbol{\delta}_k\| \le (1-\beta)\left\|\sum_{i=0}^{k}\beta^{k-i}\widehat{\boldsymbol{\delta}}_i\right\| + \beta\eta\sum_{i=0}^{k-1}\beta^{k-1-i}(K_0 + K_1\|\nabla\Phi(\boldsymbol{x}_i)\|) + \beta^{k+1}\|\boldsymbol{m}_0 - \nabla\Phi(\boldsymbol{x}_0)\|.
$$

Take summation and we obtain

$$
\sum_{k=0}^{K-1} \|\boldsymbol{\delta}_k\| \le (1-\beta) \sum_{k=0}^{K-1} \left\| \sum_{i=0}^{k} \beta^{k-i} \widehat{\boldsymbol{\delta}}_i \right\| + \frac{K_0 K \eta \beta}{1-\beta} + \frac{K_1 \eta \beta}{1-\beta} \sum_{k=0}^{K-1} \|\nabla \Phi(\boldsymbol{x}_k)\| + \frac{\beta}{1-\beta} \|\boldsymbol{m}_0 - \nabla \Phi(\boldsymbol{x}_0)\|
$$

$$
\le \underbrace{(1-\beta) \sum_{k=0}^{K-1} \left\| \sum_{i=0}^{k} \beta^{k-i} \left( \widehat{\nabla} \Phi(\boldsymbol{x}_i, \boldsymbol{y}_i, \boldsymbol{z}_i; \zeta_i, \xi_i) - \mathbb{E}_i[\widehat{\nabla} \Phi(\boldsymbol{x}_i, \boldsymbol{y}_i, \boldsymbol{z}_i; \zeta_i, \xi_i)] \right) \right\|}_{(a)} + \frac{K_0 K \eta \beta}{1-\beta} + \frac{K_1 \eta \beta}{1-\beta} \sum_{k=0}^{K-1} \|\nabla \Phi(\boldsymbol{x}_k)\|
$$

$$
+ \underbrace{(1-\beta) \sum_{k=0}^{K-1} \left\| \sum_{i=0}^{k} \beta^{k-i} \left( \mathbb{E}_i[\widehat{\nabla} \Phi(\boldsymbol{x}_i, \boldsymbol{y}_i, \boldsymbol{z}_i; \zeta_i, \xi_i)] - \nabla \Phi(\boldsymbol{x}_i) \right) \right\|}_{(b)} + \frac{\beta}{1-\beta} \|\boldsymbol{m}_0 - \nabla \Phi(\boldsymbol{x}_0)\|.
$$

$$(67)$$

Taking expectation (with respect to $\mathcal{F}_K$) on both sides of part $(a)$, we have

$$
(1-\beta) \sum_{k=0}^{K-1} \mathbb{E} \left\| \sum_{i=0}^{k} \beta^{k-i} \left( \widehat{\nabla} \Phi(\boldsymbol{x}_i, \boldsymbol{y}_i, \boldsymbol{z}_i; \zeta_i, \xi_i) - \mathbb{E}_i[\widehat{\nabla} \Phi(\boldsymbol{x}_i, \boldsymbol{y}_i, \boldsymbol{z}_i; \zeta_i, \xi_i)] \right) \right\|
$$

$$
\overset{(i)}{\le} (1-\beta) \sum_{k=0}^{K-1} \sqrt{ \mathbb{E} \left\| \sum_{i=0}^{k} \beta^{k-i} \left( \widehat{\nabla} \Phi(\boldsymbol{x}_i, \boldsymbol{y}_i, \boldsymbol{z}_i; \zeta_i, \xi_i) - \mathbb{E}_i[\widehat{\nabla} \Phi(\boldsymbol{x}_i, \boldsymbol{y}_i, \boldsymbol{z}_i; \zeta_i, \xi_i)] \right) \right\|^2 }
$$

$$
\overset{(ii)}{=} (1-\beta) \sum_{k=0}^{K-1} \sqrt{ \sum_{i=0}^{k} \beta^{2(k-i)} \mathbb{E} \left\| \widehat{\nabla} \Phi(\boldsymbol{x}_i, \boldsymbol{y}_i, \boldsymbol{z}_i; \zeta_i, \xi_i) - \mathbb{E}_i[\widehat{\nabla} \Phi(\boldsymbol{x}_i, \boldsymbol{y}_i, \boldsymbol{z}_i; \zeta_i, \xi_i)] \right\|^2 }
$$

$$
\overset{(iii)}{\le} (1-\beta) \sum_{k=0}^{K-1} \sqrt{ \sum_{i=0}^{k} \beta^{2(k-i)} \left( \sigma_{f,1}^2 + \frac{2M^2}{\mu^2} \sigma_{g,2}^2 + 2\sigma_{g,2}^2 \mathbb{E}[\|\boldsymbol{z}_i - \boldsymbol{z}_i^*\|^2] \right) }
$$

$$
\overset{(iv)}{\le} (1-\beta) \sum_{k=0}^{K-1} \sqrt{ \sum_{i=0}^{k} \beta^{2(k-i)} \left( \sigma_{f,1}^2 + \frac{2M^2}{\mu^2} \sigma_{g,2}^2 \right) } + (1-\beta) \sum_{k=0}^{K-1} \sqrt{ 2\sigma_{g,2}^2 \sum_{i=0}^{k} \beta^{2(k-i)} \mathbb{E}[\|\boldsymbol{z}_i - \boldsymbol{z}_i^*\|^2] }
$$

$$
\overset{(v)}{\le} K \frac{\sqrt{1-\beta}}{\sqrt{1+\beta}} \sqrt{ \sigma_{f,1}^2 + \frac{2M^2}{\mu^2} \sigma_{g,2}^2 } + \sqrt{2} \sigma_{g,2} (1-\beta) \sqrt{K} \sqrt{ \sum_{k=0}^{K-1} \sum_{i=0}^{k} \beta^{2(k-i)} \mathbb{E}[\|\boldsymbol{z}_i - \boldsymbol{z}_i^*\|^2] }
$$

$$
\le K \frac{\sqrt{1-\beta}}{\sqrt{1+\beta}} \sqrt{ \sigma_{f,1}^2 + \frac{2M^2}{\mu^2} \sigma_{g,2}^2 } + \sqrt{2} \sigma_{g,2} \frac{\sqrt{1-\beta}}{\sqrt{1+\beta}} \sqrt{K} \sqrt{ \sum_{k=0}^{K-1} \mathbb{E}[\|\boldsymbol{z}_k - \boldsymbol{z}_k^*\|^2] }
$$

$$
\le K \sqrt{1-\beta} \sqrt{ \sigma_{f,1}^2 + \frac{2M^2}{\mu^2} \sigma_{g,2}^2 } + \sqrt{2} \sigma_{g,2} \sqrt{1-\beta} \sqrt{K} \sqrt{ \sum_{k=0}^{K-1} \mathbb{E}[\|\boldsymbol{z}_k - \boldsymbol{z}_k^*\|^2] },
$$

$$(68)$$

where $(i)$ follows from Jensen's inequality; $(ii)$ follows from Lemma D.9; $(iii)$ follows from Lemma D.6; $(iv)$ follows from the fact that $\sqrt{a+b} \le \sqrt{a} + \sqrt{b}$ for all $a \ge 0, b \ge 0$; $(v)$ follows from the fact that $\sum_{i=1}^{n} \sqrt{a_i} \le \sqrt{n} \sqrt{\sum_{i=1}^{n} a_i}$ for all $a_i \ge 0$.

Taking expectation (with respect to $\mathcal{F}_K$) on both sides of part $(b)$, we have

$$
(1 - \beta) \sum_{k=0}^{K-1} \mathbb{E} \left\| \sum_{i=0}^{k} \beta^{k-i} \left( \mathbb{E}_i[\widehat{\nabla}\Phi(\boldsymbol{x}_i, \boldsymbol{y}_i, \boldsymbol{z}_i; \zeta_i, \xi_i)] - \nabla\Phi(\boldsymbol{x}_i) \right) \right\|
$$

$$
\overset{(i)}{\leq} (1 - \beta) \sum_{k=0}^{K-1} \sum_{i=0}^{k} \beta^{k-i} \frac{L_{\boldsymbol{x},1}\epsilon}{4K_0} \mathbb{E}\|\nabla\Phi(\boldsymbol{x}_k)\| + (1 - \beta) \sum_{k=0}^{K-1} \sum_{i=0}^{k} \beta^{k-i} \left( L_{\boldsymbol{x},0} + L_{\boldsymbol{x},1} \frac{C_{g_{xy}}M}{\mu} + \frac{\tau M}{\mu} \right) \frac{\epsilon}{4K_0}
$$

$$
+ (1 - \beta) \sum_{k=0}^{K-1} \sum_{i=0}^{k} \beta^{k-i} C_{g_{xy}} \mathbb{E}\left[\|\boldsymbol{z}_k - \boldsymbol{z}_k^*\|\right]
$$

$$
\leq \frac{L_{\boldsymbol{x},1}\epsilon}{4K_0} \sum_{k=0}^{K-1} \mathbb{E}\|\nabla\Phi(\boldsymbol{x}_k)\| + K \left( L_{\boldsymbol{x},0} + L_{\boldsymbol{x},1} \frac{C_{g_{xy}}M}{\mu} + \frac{\tau M}{\mu} \right) \frac{\epsilon}{4K_0}
$$

$$
+ \underbrace{(1 - \beta)C_{g_{xy}} \sum_{k=0}^{K-1} \sum_{i=0}^{k} \beta^{k-i} \mathbb{E}\left[\|\boldsymbol{z}_k - \boldsymbol{z}_k^*\|\right]}_{(c)}.
$$

$$\tag{69}$$

where $(i)$ follows from Lemma 3.5.

For part $(c)$, we have

$$
(1 - \beta)C_{g_{xy}} \sum_{k=0}^{K-1} \sum_{i=0}^{k} \beta^{k-i} \mathbb{E}\left[\|\boldsymbol{z}_k - \boldsymbol{z}_k^*\|\right] \overset{(i)}{\leq} (1 - \beta)C_{g_{xy}} \sum_{k=0}^{K-1} \sum_{i=0}^{k} \beta^{k-i} \sqrt{\mathbb{E}[\|\boldsymbol{z}_k - \boldsymbol{z}_k^*\|^2]}
$$

$$
\leq C_{g_{xy}} \sum_{k=0}^{K-1} \sqrt{\mathbb{E}[\|\boldsymbol{z}_k - \boldsymbol{z}_k^*\|^2]} \overset{(ii)}{\leq} C_{g_{xy}} \sqrt{K} \sqrt{\sum_{k=0}^{K-1} \mathbb{E}[\|\boldsymbol{z}_k - \boldsymbol{z}_k^*\|^2]}.
$$

$$\tag{70}$$

where $(i)$ follows from Jensen's inequality and $(ii)$ follows from the fact that $\sum_{i=1}^{n} \sqrt{a_i} \leq \sqrt{n}\sqrt{\sum_{i=1}^{n} a_i}$ for all $a_i \geq 0$.

Therefore, combining (67), (68), (69) and (70) yields

$$
\mathbb{E}\left[ \sum_{k=0}^{K-1} \|\boldsymbol{\delta}_k\| \right] \leq (1 - \beta) \sum_{k=0}^{K-1} \mathbb{E} \left\| \sum_{i=0}^{k} \beta^{k-i} \left( \widehat{\nabla}\Phi(\boldsymbol{x}_i, \boldsymbol{y}_i, \boldsymbol{z}_i; \zeta_i, \xi_i) - \mathbb{E}_i[\widehat{\nabla}\Phi(\boldsymbol{x}_i, \boldsymbol{y}_i, \boldsymbol{z}_i; \zeta_i, \xi_i)] \right) \right\|
$$

$$
+ \frac{K_0 K \eta \beta}{1 - \beta} + \frac{K_1 \eta \beta}{1 - \beta} \sum_{k=0}^{K-1} \|\nabla\Phi(\boldsymbol{x}_k)\| + \frac{\beta}{1 - \beta} \|\boldsymbol{m}_0 - \nabla\Phi(\boldsymbol{x}_0)\|
$$

$$
+ (1 - \beta) \sum_{k=0}^{K-1} \mathbb{E} \left\| \sum_{i=0}^{k} \beta^{k-i} \left( \mathbb{E}_i[\widehat{\nabla}\Phi(\boldsymbol{x}_i, \boldsymbol{y}_i, \boldsymbol{z}_i; \zeta_i, \xi_i)] - \nabla\Phi(\boldsymbol{x}_i) \right) \right\|
$$

$$
\leq K\sqrt{1 - \beta}\sqrt{\sigma_{f,1}^2 + \frac{2M^2}{\mu^2}\sigma_{g,2}^2} + \sqrt{2}\sigma_{g,2}\sqrt{1 - \beta}\sqrt{K}\sqrt{\sum_{k=0}^{K-1} \mathbb{E}[\|\boldsymbol{z}_k - \boldsymbol{z}_k^*\|^2]}
$$

$$
+ \frac{L_{\boldsymbol{x},1}\epsilon}{4K_0} \sum_{k=0}^{K-1} \|\nabla\Phi(\boldsymbol{x}_k)\| + K \left( L_{\boldsymbol{x},0} + L_{\boldsymbol{x},1} \frac{C_{g_{xy}}M}{\mu} + \frac{\tau M}{\mu} \right) \frac{\epsilon}{4K_0} + \frac{K_0 K \eta \beta}{1 - \beta}
$$

$$
+ C_{g_{xy}} \sqrt{K} \sqrt{\sum_{k=0}^{K-1} \mathbb{E}[\|\boldsymbol{z}_k - \boldsymbol{z}_k^*\|^2]} + \frac{K_1 \eta \beta}{1 - \beta} \sum_{k=0}^{K-1} \|\nabla\Phi(\boldsymbol{x}_k)\| + \frac{\beta}{1 - \beta} \|\boldsymbol{m}_0 - \nabla\Phi(\boldsymbol{x}_0)\|.
$$

$$\tag{71}$$

$\square$

**Lemma D.9.** *We have the following fact*

$$
\mathbb{E}\left\|\sum_{i=0}^{k}\beta^{k-i}\left(\widehat{\nabla}\Phi(\boldsymbol{x}_i,\boldsymbol{y}_i,\boldsymbol{z}_i;\zeta_i,\xi_i)-\mathbb{E}_i[\widehat{\nabla}\Phi(\boldsymbol{x}_i,\boldsymbol{y}_i,\boldsymbol{z}_i;\zeta_i,\xi_i)]\right)\right\|^2
$$

$$
=\sum_{i=0}^{k}\beta^{2(k-i)}\mathbb{E}\left\|\widehat{\nabla}\Phi(\boldsymbol{x}_i,\boldsymbol{y}_i,\boldsymbol{z}_i;\zeta_i,\xi_i)-\mathbb{E}_i[\widehat{\nabla}\Phi(\boldsymbol{x}_i,\boldsymbol{y}_i,\boldsymbol{z}_i;\zeta_i,\xi_i)]\right\|^2,
$$

(72)

*where the expectation is taken over the randomness in $\mathcal{F}_K$.*

***Proof of Lemma D.9.*** We will show (72) by using conditional expectation, the law of total expectation and recursion.

$$
\mathbb{E}\left\|\sum_{i=0}^{k}\beta^{k-i}\left(\widehat{\nabla}\Phi(\boldsymbol{x}_i,\boldsymbol{y}_i,\boldsymbol{z}_i;\zeta_i,\xi_i)-\mathbb{E}_i[\widehat{\nabla}\Phi(\boldsymbol{x}_i,\boldsymbol{y}_i,\boldsymbol{z}_i;\zeta_i,\xi_i)]\right)\right\|^2
$$

$$
\stackrel{(i)}{=}\mathbb{E}_{\mathcal{F}_k}\left[\mathbb{E}_k\left[\left\|\sum_{i=0}^{k}\beta^{k-i}\left(\widehat{\nabla}\Phi(\boldsymbol{x}_i,\boldsymbol{y}_i,\boldsymbol{z}_i;\zeta_i,\xi_i)-\mathbb{E}_i[\widehat{\nabla}\Phi(\boldsymbol{x}_i,\boldsymbol{y}_i,\boldsymbol{z}_i;\zeta_i,\xi_i)]\right)\right\|^2\right]\right]
$$

$$
\stackrel{(ii)}{=}\mathbb{E}_{\mathcal{F}_k}\left[\mathbb{E}_k\left[\left\|\underbrace{\beta^0\left(\widehat{\nabla}\Phi(\boldsymbol{x}_k,\boldsymbol{y}_k,\boldsymbol{z}_k;\zeta_k,\xi_k)-\mathbb{E}_k[\widehat{\nabla}\Phi(\boldsymbol{x}_k,\boldsymbol{y}_k,\boldsymbol{z}_k;\zeta_k,\xi_k)]\right)}_{(a)}\right.\right.
$$

$$
\left.\left.+\underbrace{\sum_{i=0}^{k-1}\beta^{k-i}\left(\widehat{\nabla}\Phi(\boldsymbol{x}_i,\boldsymbol{y}_i,\boldsymbol{z}_i;\zeta_i,\xi_i)-\mathbb{E}_i[\widehat{\nabla}\Phi(\boldsymbol{x}_i,\boldsymbol{y}_i,\boldsymbol{z}_i;\zeta_i,\xi_i)]\right)}_{(b)}\right\|^2\right]\right]
$$

$$
\stackrel{(iii)}{=}\beta^{2\times 0}\mathbb{E}\left\|\widehat{\nabla}\Phi(\boldsymbol{x}_k,\boldsymbol{y}_k,\boldsymbol{z}_k;\zeta_k,\xi_k)-\mathbb{E}_k[\widehat{\nabla}\Phi(\boldsymbol{x}_k,\boldsymbol{y}_k,\boldsymbol{z}_k;\zeta_k,\xi_k)]\right\|^2
$$

$$
+\mathbb{E}\left[\left\|\sum_{i=0}^{k-1}\beta^{k-i}\left(\widehat{\nabla}\Phi(\boldsymbol{x}_i,\boldsymbol{y}_i,\boldsymbol{z}_i;\zeta_i,\xi_i)-\mathbb{E}_i[\widehat{\nabla}\Phi(\boldsymbol{x}_i,\boldsymbol{y}_i,\boldsymbol{z}_i;\zeta_i,\xi_i)]\right)\right\|^2\right]
$$

$$
=\sum_{i=k}^{k}\beta^{2(k-i)}\mathbb{E}\left\|\widehat{\nabla}\Phi(\boldsymbol{x}_i,\boldsymbol{y}_i,\boldsymbol{z}_i;\zeta_i,\xi_i)-\mathbb{E}_i[\widehat{\nabla}\Phi(\boldsymbol{x}_i,\boldsymbol{y}_i,\boldsymbol{z}_i;\zeta_i,\xi_i)]\right\|^2
$$

$$
+\mathbb{E}_{\mathcal{F}_{k-1}}\left[\mathbb{E}_{k-1}\left[\left\|\sum_{i=0}^{k-1}\beta^{k-i}\left(\widehat{\nabla}\Phi(\boldsymbol{x}_i,\boldsymbol{y}_i,\boldsymbol{z}_i;\zeta_i,\xi_i)-\mathbb{E}_i[\widehat{\nabla}\Phi(\boldsymbol{x}_i,\boldsymbol{y}_i,\boldsymbol{z}_i;\zeta_i,\xi_i)]\right)\right\|^2\right]\right]
$$

$$
\stackrel{(iv)}{=}\sum_{i=k-1}^{k}\beta^{2(k-i)}\mathbb{E}\left\|\widehat{\nabla}\Phi(\boldsymbol{x}_i,\boldsymbol{y}_i,\boldsymbol{z}_i;\zeta_i,\xi_i)-\mathbb{E}_i[\widehat{\nabla}\Phi(\boldsymbol{x}_i,\boldsymbol{y}_i,\boldsymbol{z}_i;\zeta_i,\xi_i)]\right\|^2
$$

$$
+\mathbb{E}\left[\left\|\sum_{i=0}^{k-2}\beta^{k-i}\left(\widehat{\nabla}\Phi(\boldsymbol{x}_i,\boldsymbol{y}_i,\boldsymbol{z}_i;\zeta_i,\xi_i)-\mathbb{E}_i[\widehat{\nabla}\Phi(\boldsymbol{x}_i,\boldsymbol{y}_i,\boldsymbol{z}_i;\zeta_i,\xi_i)]\right)\right\|^2\right]
$$

$$
\stackrel{(v)}{=}\sum_{i=0}^{k}\beta^{2(k-i)}\mathbb{E}\left\|\widehat{\nabla}\Phi(\boldsymbol{x}_i,\boldsymbol{y}_i,\boldsymbol{z}_i;\zeta_i,\xi_i)-\mathbb{E}_i[\widehat{\nabla}\Phi(\boldsymbol{x}_i,\boldsymbol{y}_i,\boldsymbol{z}_i;\zeta_i,\xi_i)]\right\|^2,
$$

where $(i)$ follows from the law of total expectation; $(ii)$ follows from the fact that part $(b)$ is $\mathcal{F}_k$-measurable and uncorrelated with part $(a)$; $(iii)$ follows from the law of total expectation; $(iv)$ follows from the same procedures as $(ii)$ and $(iii)$; $(v)$ follows from recursion and then proof is completed. $\qquad\square$

# E PROOF OF THEOREM 3.1

Before proving Theorem 3.1, we require the following lemma to characterize the function value decrease from iteration $k$ to iteration $k + 1$, which is similar to Lemma C.6 in Jin et al. (2021).

**Lemma E.1.** *For Algorithm 1, define $\boldsymbol{\delta}_k := \boldsymbol{m}_{k+1} - \nabla\Phi(\boldsymbol{x}_k)$ to be the the moving-average estimation error. Then we have*

$$\Phi(\boldsymbol{x}_{k+1}) - \Phi(\boldsymbol{x}_k) \leq -\left(\eta - \frac{1}{2}K_1\eta^2\right)\|\nabla\Phi(\boldsymbol{x}_k)\| + \frac{1}{2}K_0\eta^2 + 2\eta\|\boldsymbol{\delta}_k\|. \tag{73}$$

*Further, by a telescope sum we have*

$$\left(1 - \frac{1}{2}K_1\eta\right)\sum_{k=0}^{K-1}\|\nabla\Phi(\boldsymbol{x}_k)\| \leq \frac{\Delta}{\eta} + \frac{1}{2}K_0 K\eta + 2\sum_{k=0}^{K-1}\|\boldsymbol{\delta}_k\|, \tag{74}$$

*where $\Delta := \Phi(\boldsymbol{x}_0) - \Phi^*$ and $\Phi^* = \inf_{\boldsymbol{x}\in\mathbb{R}^d}\Phi(\boldsymbol{x})$.*

***Proof of Lemma E.1.*** For Algorithm 1 we have $\|\boldsymbol{x}_{k+1} - \boldsymbol{x}_k\| = \eta$, and by Lemma C.4 we obtain

$$\begin{aligned}
\Phi(\boldsymbol{x}_{k+1}) - \Phi(\boldsymbol{x}_k) &\leq -\frac{\eta}{\|\boldsymbol{m}_{k+1}\|}\langle\nabla\Phi(\boldsymbol{x}_k), \boldsymbol{m}_{k+1}\rangle + \frac{1}{2}\eta^2\left(K_0 + K_1\|\nabla\Phi(\boldsymbol{x}_k)\|\right) \\
&\overset{(i)}{\leq} \eta\left(-\|\nabla\Phi(\boldsymbol{x}_k)\| + 2\|\boldsymbol{\delta}_k\|\right) + \frac{1}{2}\eta^2\left(K_0 + K_1\|\nabla\Phi(\boldsymbol{x}_k)\|\right) \\
&= -\left(\eta - \frac{1}{2}K_1\eta^2\right)\|\nabla\Phi(\boldsymbol{x}_k)\| + \frac{1}{2}K_0\eta^2 + 2\eta\|\boldsymbol{\delta}_k\|,
\end{aligned}$$

where $(i)$ follows from Lemma C.5 with $\omega = 1$. $\square$

With Lemma 3.6 and Lemma E.1, now we proceed to prove Theorem 3.1.

**Theorem E.2** (Theorem 3.1 restated). *Suppose Assumptions 1, 2 and 3 hold. Run Algorithm 1 for $K$ iterations and let $\{\boldsymbol{x}_k\}_{k\geq 0}$ be the sequence produced by Algorithm 1. For $\epsilon \leq$*

$$\min\left(\frac{K_0}{K_1}, \sqrt{\frac{\sigma_{f,1}^2 + \frac{2M^2}{\mu^2}\sigma_{g,2}^2}{\min\left(1, \frac{\mu^2}{32C_{g_{xy}}^2}\right)}}\right) \text{ and given } \delta \in (0, 1), \text{ if we choose } \alpha_s \text{ as (30), } \gamma \text{ as (44), } N \text{ as (45),}$$

$I = \frac{\sigma_{g,1}^2 K_0^2}{\mu^2\epsilon^2}$, *and*

$$1 - \beta = \min\left(\frac{\epsilon^2}{\sigma_{f,1}^2 + \frac{2M^2}{\mu^2}\sigma_{g,2}^2}\min\left(1, \frac{\mu^2}{32C_{g_{xy}}^2}\right), \frac{C_{g_{xy}}^2}{8\sigma_{g,2}^2}, \frac{\mu^2}{16\sigma_{g,2}^2}, \frac{1}{4}\right), \quad \nu = \frac{1}{\mu}(1 - \beta), \quad I = \frac{\sigma_{g,1}^2 K_0^2}{\mu^2\epsilon^2},$$

$$\eta = \min\left(\frac{1}{8}\min\left(\frac{1}{K_1}, \frac{\epsilon}{K_0}, \frac{\Delta}{\|\nabla\Phi(\boldsymbol{x}_0)\|}, \frac{\epsilon\Delta}{C_{g_{xy}}^2\Delta_{\boldsymbol{z},0}}\right)(1 - \beta), \frac{1}{\sqrt{2\left(1 + \frac{C_{g_{xy}}^2}{\mu^2}\right)(L_{\boldsymbol{x},1}^2 + L_{\boldsymbol{y},1}^2)}}, \frac{\mu\epsilon}{8K_0 I C_{g_{xy}}}\right),$$

$$\tag{75}$$

*where $\Delta := \Phi(\boldsymbol{x}_0) - \inf_{\boldsymbol{x}\in\mathbb{R}^{d_x}}\Phi(\boldsymbol{x})$ and $\Delta_{\boldsymbol{z},0} := \|\boldsymbol{z}_0 - \boldsymbol{z}_0^*\|^2$, then with probability at least $1 - \delta$ over the randomness in $\widetilde{\mathcal{F}}_K$, Algorithm 1 guarantees $\frac{1}{K}\sum_{k=0}^{K-1}\mathbb{E}\|\nabla\Phi(\boldsymbol{x}_k)\| \leq 30\epsilon$ as long as $K = \frac{4\Delta}{\eta\epsilon}$, where the expectation is taken over the randomness in $\mathcal{F}_K$. In addition, the number of oracle calls for updating lower-level variable $\boldsymbol{y}$ (in Algorithm 2 and Algorithm 3) is at most $\widetilde{O}\left(\frac{\Delta}{\eta\epsilon}\right)$.*

***Proof of Theorem E.2.*** Taking total expectations (with respect to $\mathcal{F}_K$) on both sides of (74) in Lemma E.1, we obtain

$$\left(1 - \frac{1}{2}K_1\eta\right)\sum_{k=0}^{K-1}\mathbb{E}\|\nabla\Phi(\boldsymbol{x}_k)\| \leq \frac{\Delta}{\eta} + \frac{1}{2}K_0 K\eta + 2\mathbb{E}\left[\sum_{k=0}^{K-1}\|\boldsymbol{\delta}_k\|\right],$$

Now we plug (63) and (64) of Lemma 3.6 into the above inequality, rearrange and we have

$$
\left(1 - \left(\frac{1}{2} + \frac{2\beta}{1-\beta}\right) K_1\eta - \frac{L_{\boldsymbol{x},1}\epsilon}{2K_0}\right) \frac{1}{K} \sum_{k=0}^{K-1} \mathbb{E}\|\nabla\Phi(\boldsymbol{x}_k)\|
$$

$$
\leq \underbrace{2\left[\sqrt{1-\beta}\sqrt{\sigma_{f,1}^2 + \frac{2M^2}{\mu^2}\sigma_{g,2}^2} + \frac{K_0\eta\beta}{1-\beta} + \frac{1}{4}K_0\eta + \left(L_{\boldsymbol{x},0} + L_{\boldsymbol{x},1}\frac{C_{g_{xy}}M}{\mu} + \frac{\tau M}{\mu}\right)\frac{\epsilon}{4K_0}\right]}_{\text{(I)}}
$$

$$
+ \underbrace{2\left(2\sqrt{2}\sigma_{g,2}\sqrt{1-\beta} + 2C_{g_{xy}}\right)\sqrt{\frac{1}{K}\sum_{k=0}^{K-1}\mathbb{E}[\|\boldsymbol{z}_k - \boldsymbol{z}_k^*\|^2]} + \frac{2\beta}{K(1-\beta)}\|\boldsymbol{m}_0 - \nabla\Phi(\boldsymbol{x}_0)\| + \frac{\Delta}{K\eta}}_{\text{(II)}}.
$$

(76)

If we choose

$$
\epsilon \leq \min\left(\frac{K_0}{K_1}, \sqrt{\frac{\sigma_{f,1}^2 + \frac{2M^2}{\mu^2}\sigma_{g,2}^2}{\min\left(1, \frac{\mu^2}{32C_{g_{xy}}^2}\right)}}\right), \quad 1-\beta = \min\left(\frac{\epsilon^2}{\sigma_{f,1}^2 + \frac{2M^2}{\mu^2}\sigma_{g,2}^2}\min\left(1, \frac{\mu^2}{32C_{g_{xy}}^2}\right), \frac{C_{g_{xy}}^2}{8\sigma_{g,2}^2}, \frac{\mu^2}{16\sigma_{g,2}^2}, \frac{1}{4}\right),
$$

$$
\eta = \min\left(\frac{1}{8}\min\left(\frac{1}{K_1}, \frac{\epsilon}{K_0}, \frac{\Delta}{\|\nabla\Phi(\boldsymbol{x}_0)\|}, \frac{\epsilon\Delta}{C_{g_{xy}}^2\Delta_{\boldsymbol{z},0}}\right)(1-\beta), \frac{1}{\sqrt{2\left(1 + \frac{C_{g_{xy}}^2}{\mu^2}\right)(L_{\boldsymbol{x},1}^2 + L_{\boldsymbol{y},1}^2)}}, \frac{\mu\epsilon}{8K_0 I C_{g_{xy}}}\right),
$$

$$
\nu = \frac{1}{\mu}(1-\beta), \quad K = \frac{4\Delta}{\eta\epsilon}, \quad \boldsymbol{m}_0 = \boldsymbol{0},
$$

where $\Delta = \Phi(\boldsymbol{x}_0) - \Phi^*$ and $\Delta_{\boldsymbol{z},0} = \|\boldsymbol{z}_0 - \boldsymbol{z}_0^*\|^2$, then for left-hand side of (76) we have

$$
\left(1 - \left(\frac{1}{2} + \frac{2\beta}{1-\beta}\right)K_1\eta - \frac{L_{\boldsymbol{x},1}\epsilon}{2K_0}\right) \overset{(i)}{\geq} 1 - \frac{1+3\beta}{2(1-\beta)}K_1\eta - \frac{K_1\epsilon}{2K_0} \geq 1 - \frac{2K_1\eta}{1-\beta} - \frac{1}{2} \geq \frac{1}{4}, \quad (77)
$$

where $(i)$ follows from $L_{\boldsymbol{x},1} \leq K_1$ by definition (16) of $K_1$.

For the first part (I) of right-hand side of (76) we have

$$
\text{(I)} \overset{(i)}{\leq} 2\left(\frac{\epsilon}{\sqrt{\sigma_{f,1}^2 + \frac{2M^2}{\mu^2}\sigma_{g,2}^2}}\sqrt{\sigma_{f,1}^2 + \frac{2M^2}{\mu^2}\sigma_{g,2}^2} + \frac{\epsilon\beta(1-\beta)}{8(1-\beta)} + \frac{\epsilon(1-\beta)}{32} + \frac{\epsilon K_0}{4K_0}\right)
$$

$$
\leq 2\left(\epsilon + \frac{1}{8}\epsilon + \frac{1}{32}\epsilon + \frac{1}{4}\epsilon\right) = \frac{45}{16}\epsilon,
$$

(78)

where $(i)$ follows from the fact that (recall definition (16) of $K_0$)

$$
1-\beta \leq \frac{\epsilon^2}{\sigma_{f,1}^2 + \frac{2M^2}{\mu^2}\sigma_{g,2}^2}, \quad \eta \leq \frac{\epsilon}{8K_0}(1-\beta), \quad \left(L_{\boldsymbol{x},0} + L_{\boldsymbol{x},1}\frac{C_{g_{xy}}M}{\mu} + \frac{\tau M}{\mu}\right) \leq K_0.
$$

Also, for the second part (II) of right-hand side of (76) we have

$$
\begin{aligned}
(\text{II}) &\stackrel{(i)}{\leq} 2\left(2\sqrt{2}\sigma_{g,2}\sqrt{\frac{C_{g_{xy}}^2}{8\sigma_{g,2}^2}} + 2C_{g_{xy}}\right)\sqrt{\frac{1}{K}\sum_{k=0}^{K-1}\mathbb{E}[\|\boldsymbol{z}_k - \boldsymbol{z}_k^*\|^2]} + \frac{2\beta}{K(1-\beta)}\|\nabla\Phi(\boldsymbol{x}_0)\| + \frac{\Delta}{K\eta} \\
&\stackrel{(ii)}{\leq} 6C_{g_{xy}}\sqrt{\frac{1}{K}\sum_{k=0}^{K-1}\mathbb{E}[\|\boldsymbol{z}_k - \boldsymbol{z}_k^*\|^2]} + \frac{1}{16}\epsilon + \frac{1}{4}\epsilon \\
&\leq 6C_{g_{xy}}\left[\frac{\Delta_{\boldsymbol{z},0}}{K\nu\mu} + \frac{5}{\mu^2}\left(\frac{\rho^2 M^2}{\mu^2} + (L_{\boldsymbol{y},0} + L_{\boldsymbol{y},1}M)^2\right)\left(\frac{\epsilon}{4K_0}\right)^2 \right. \\
&\qquad\left. + \frac{2}{\mu}\left(\frac{2M^2}{\mu^2}\sigma_{g,2}^2 + \sigma_{f,1}^2\right)\nu + \frac{4L_{\boldsymbol{z}^*}^2\eta^2}{\mu^2}\frac{\eta^2}{\nu^2}\right]^{\frac{1}{2}} + \frac{5}{16}\epsilon \\
&\stackrel{(iii)}{\leq} 6\left[\frac{C_{g_{xy}}^2\Delta_{\boldsymbol{z},0}}{K(1-\beta)} + \frac{5C_{g_{xy}}^2}{\mu^2}\left(\frac{\rho^2 M^2}{\mu^2} + (L_{\boldsymbol{y},0} + L_{\boldsymbol{y},1}M)^2\right)\frac{\epsilon^2}{16K_0^2} \right. \\
&\qquad\left. + \frac{2C_{g_{xy}}^2}{\mu^2}\left(\frac{2M^2}{\mu^2}\sigma_{g,2}^2 + \sigma_{f,1}^2\right)(1-\beta) + \frac{4C_{g_{xy}}^2 L_{\boldsymbol{z}^*}^2}{\mu^2}\frac{\mu^2\epsilon^2}{64K_0^2}\right]^{\frac{1}{2}} + \frac{5}{16}\epsilon \\
&\stackrel{(iv)}{\leq} 6\sqrt{\frac{C_{g_{xy}}^2\eta\epsilon\Delta_{\boldsymbol{z},0}}{4\Delta(1-\beta)} + \frac{5K_0^2\epsilon^2}{16K_0^2} + \frac{1}{16}\epsilon^2 + \frac{K_0^2\epsilon^2}{16K_0^2}} + \frac{5}{16}\epsilon \\
&\leq 6\sqrt{\frac{C_{g_{xy}}^2\eta\epsilon\Delta_{\boldsymbol{z},0}}{4\Delta(1-\beta)} + \frac{7}{16}\epsilon^2} + \frac{5}{16}\epsilon \\
&\stackrel{(v)}{\leq} 6\sqrt{\frac{1}{32}\epsilon^2 + \frac{7}{16}\epsilon^2} + \frac{5}{16}\epsilon \\
&\leq \left(3\sqrt{2} + \frac{5}{16}\right)\epsilon,
\end{aligned}
$$
(79)

where $(i)$ follows from $1 - \beta \leq \frac{C_{g_{xy}}^2}{8\sigma_{g,2}^2}$ and $\boldsymbol{m}_0 = \boldsymbol{0}$; $(ii)$ follows from $K = \frac{4\Delta}{\eta\epsilon}$ and $\eta \leq \frac{\Delta(1-\beta)}{8\|\nabla\Phi(\boldsymbol{x}_0)\|}$; $(iii)$ follows from $\eta \leq \frac{\epsilon(1-\beta)}{8K_0}$ and $\nu = \frac{1-\beta}{\mu}$; $(iv)$ follows from $K = \frac{4\Delta}{\eta\epsilon}$ and the fact that (recall definition (16) of $K_0$ and definition (12) of $L_{\boldsymbol{z}^*}$)

$$
\frac{5C_{g_{xy}}^2}{\mu^2}\left(\frac{\rho^2 M^2}{\mu^2} + (L_{\boldsymbol{y},0} + L_{\boldsymbol{y},1}M)^2\right) \leq 5K_0^2, \quad C_{g_{xy}}^2 L_{\boldsymbol{z}^*}^2 \leq K_0^2, \quad 1 - \beta \leq \frac{\epsilon^2}{\sigma_{f,1}^2 + \frac{2M^2}{\mu^2}\sigma_{g,2}^2}\frac{\mu^2}{32C_{g_{xy}}^2};
$$

and $(v)$ follows from $\eta \leq \frac{\epsilon\Delta}{8C_{g_{xy}}^2\Delta_{\boldsymbol{z},0}}(1-\beta)$. Therefore, combining (76), (77), (78) and (79) yields

$$
\frac{1}{K}\sum_{k=0}^{K-1}\mathbb{E}\|\nabla\Phi(\boldsymbol{x}_k)\| \leq 4\left(\frac{45}{16}\epsilon + \left(3\sqrt{2} + \frac{5}{16}\right)\epsilon\right) \leq 30\epsilon.
$$

Next, we give more details about update period $I$, step-size $\gamma$, step-size $\alpha_s$ for $0 \leq s \leq k^\dagger - 1$ based on the chosen parameters above, and then we compute the total number of oracle calls for Algorithm 2 and Algorithm 3.

**Step-size $\gamma$ and number of oracle calls for Algorithm 2.** If we choose update period $I = \frac{\sigma_{g,1}^2 K_0^2}{\mu^2\epsilon^2}$ in Algorithm 2, then by (44), we need to set step-size $\gamma$ to be

$$
\gamma = \frac{\mu\epsilon^2}{512K_0^2\sigma_{g,1}^2\sqrt{\lambda_1 + 1}\left(\lambda_1 + \sqrt{\lambda_1 + 1}\right)},
$$
(80)

and by (53), (54) and Lemma 3.3, together with $K = \frac{4\Delta}{\eta\epsilon}$, we need at most

$$
\frac{K64^2\sigma_{g,1}^2 K_0^2\left(\lambda_1 + \sqrt{\lambda_1 + 1}\right)^2}{I\mu^2\epsilon^2} = \frac{128^2\Delta\left(\lambda_1 + \sqrt{\lambda_1 + 1}\right)^2}{\eta\epsilon}
$$
(81)

iterations in total performed by Algorithm 2 to update $\boldsymbol{y}_k$ for $k \geq 1$, where in (80) and (81)

$$\lambda_1 = \max\left(\sqrt{3\ln\left(\frac{2K}{\delta I}\right)}, \ln\left(\frac{2K}{\delta I}\right)\right) = \max\left\{\sqrt{3\ln\left(\frac{8\Delta\mu^2\epsilon}{\delta\eta\sigma_{g,1}^2 K_0^2}\right)}, \ln\left(\frac{8\Delta\mu^2\epsilon}{\delta\eta\sigma_{g,1}^2 K_0^2}\right)\right\}. \tag{82}$$

Therefore, the order of step-size $\gamma$ in Algorithm 2 is $\gamma = O(\mu\epsilon^2/K_0^2\sigma_{g,1}^2)$.

**Step-size $\alpha_s$ and number of oracle calls for Algorithm 3.** The step-size $\alpha_s$ for Algorithm 3 is given in (30). By (42), (43) and Lemma 3.2, we need at most

$$\left(\log_2\left(\frac{128K_0^2\mathcal{V}_0}{\mu\epsilon^2}\right) + 1\right)\left(\frac{16L}{\mu} + 1\right) + \frac{256 \times 128K_0^2(4\lambda_2^2 + 1)\sigma_{g,1}^2}{\mu^2\epsilon^2} \tag{83}$$

iterations in total performed by Algorithm 3 to update $\boldsymbol{y}_0$, where

$$\lambda_2 = \max\left(\sqrt{3\ln\left(\frac{2k^\dagger}{\delta}\right)}, \ln\left(\frac{2k^\dagger}{\delta}\right)\right)$$

$$= \max\left\{\sqrt{3\ln\left[\frac{2\left\lceil\log_2\left(\frac{128\mathcal{V}_0 K_0^2}{\mu\epsilon^2}\right)\right\rceil}{\delta}\right]}, \ln\left[\frac{2\left\lceil\log_2\left(\frac{128\mathcal{V}_0 K_0^2}{\mu\epsilon^2}\right)\right\rceil}{\delta}\right]\right\}. \tag{84}$$

**Total number of oracle calls in Algorithm 2 and Algorithm 3.** Combining (81), (82), (83) and (84), the number of oracle calls needed in Algorithm 2 and 3 are at most

$$\underbrace{\frac{128^2\Delta\left(\lambda_1 + \sqrt{\lambda_1 + 1}\right)^2}{\eta\epsilon}}_{\text{Algorithm 2}} + \underbrace{\left(\log_2\left(\frac{128K_0^2\mathcal{V}_0}{\mu\epsilon^2}\right) + 1\right)\left(\frac{16L}{\mu} + 1\right) + \frac{256 \times 128K_0^2(4\lambda_2^2 + 1)\sigma_{g,1}^2}{\mu^2\epsilon^2}}_{\text{Algorithm 3}},$$

which is at most $\widetilde{O}\left(\Delta/\eta\epsilon\right)$ number of oracle calls in total (recall how we choose $\eta$ and the order of $\eta$). Also, recall that we need at most $K = 4\Delta/\eta\epsilon$ number of iterations for Algorithm 1. Therefore, the total complexity is at most $\widetilde{O}(\Delta/\eta\epsilon)$. $\qquad\square$

# F  IMPLEMENTATION DETAILS OF EXPERIMENTS

## F.1  EXPERIMENTAL DETAILS OF HYPER-REPRESENTATION

The meta-learning experiments are performed on Amazon Reviews Dataset (Blitzer et al., 2006) for text classification. The data contains positive and negative reviews, coming from 25 different types (domains) of products, where three domains (i.e. "office_products", "automotive" and "computer_video_games") are selected as a testing set, which contains fewer samples. For each task $\mathcal{T}_i$, we randomly draw samples from random 3 domains, where 20 samples form support set $\mathcal{S}_i$ and 20 samples form query set $\mathcal{Q}_i$. Every 20 tasks form a task batch, and a meta update (3) for upper-level $\boldsymbol{w}$ is performed over a task batch (the size of task batch $m = 20$). The lower-level update for variable $\boldsymbol{\theta}_i(i = 1, .., m)$ of base learner $i$ is updated by SGD. The total number of iterations (i.e., the number of outer loops) in one epoch for updating $\boldsymbol{w}$ is set as $K = 400$. The total number of epochs (i.e., the number of passes over the data) for this experiment is set as 20.

For all the baseline methods, the meta model is a 2-layer RNN with input dimension=300, hidden-layer dimension=4096, and output dimension=512. The base model is a linear layer with input dimension=512 and output dimension=2. All the parameters are initialized to the range of $(-1.0, 1.0)$ uniformly.

**Parameter selection for the experiments in Figure 1(a) and Figure 2(a):** We use grid search to tune the lower-level and upper-level step sizes from $\{0.001, 0.005, 0.01, 0.05, 0.1, 0.5\}$ for all methods. The best combinations of lower-level and upper-level learning rates are $(0.05, 0.1)$ for MAML and ANIL, $(0.05, 0.05)$ for StocBio, $(0.1, 0.01)$ for TTSA, $(0.05, 0.05)$ for SOBA and SABA, $(0.05, 0.1)$ for MA-SOBA, and $(0.001, 0.01)$ for BO-REP. For double-loop methods (i.e.,

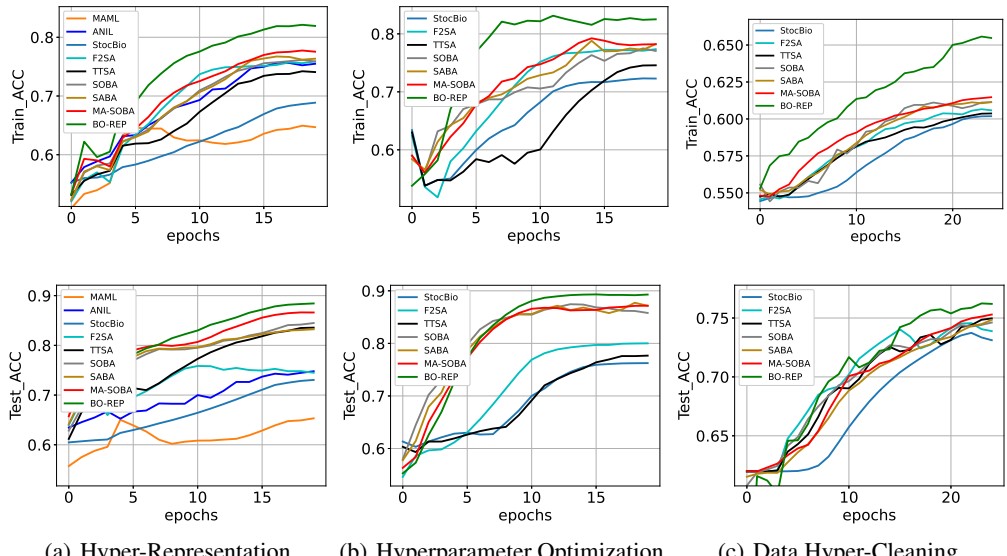

(a) Hyper-Representation    (b) Hyperparameter Optimization    (c) Data Hyper-Cleaning

Figure 2: Training and testing accuracy results for different algorithms. (a) Results of hyper-representation on Amazon Review Dataset. (b) Results of hyperparameter optimization on Amazon Review Dataset. (c) Results of data hyper-cleaning on Sentiment140 dataset with corruption rate $p = 0.3$.

MAML, ANIL, StocBio), we tune the number of iterations in the inner-loop from $\{5, 10, 20\}$ and the best value is $10$. For SOBA, SABA, MA-SOBA, and BO-REP, the step sizes for solving linear system variable $z$ are all chosen as $0.01$, which is the best tuned value from $\{0.001, 0.01, 0.1\}$. For F$^2$SA, since it is an fully first-order method and have different three variables, we tune the learning rate for these three decision variables from the range $\{0.001, 0.005, 0.01, 0.05, 0.1, 0.5\}$ and the best tuned combination of value is $(0.05, 0.05, 0.01)$. The momentum parameter $\beta$ for MA-SOBA and BO-REP is fixed as $0.9$. We increase the lagrangian multipler of F$^2$SA (denoted as $\lambda$ in F$^2$SA) by $0.01$ for every meta update.

In particular, BO-REP updates lower-level variable $\theta_i$ periodically with interval $I = 2$, which means there is one update for $y$ every two outer loops. The number of iterations $(N)$ for each periodic update in Algorithm 2 is set as $3$, the ball radius $R$ for projection is $0.5$ (ablation study for $R$ can be found in Section J.2). In addition, for simplicity, BO-REP just adopts SGD in the stage of initialization refinement, which can be regarded as an special case of epoch SGD with only one stage.

## F.2 EXPERIMENTAL DETAILS OF HYPERPARAMETER OPTIMIZATION

We conduct hyperparameter optimization on the Amazon Review dataset. We randomly sample $20000$ training samples and $2000$ testing samples from the training and testing set, respectively. A regularization parameter $\lambda$ in (4) is the upper-level variable, which is initialized as $0.0$. $w$ is the lower-level variable, the model parameter of a 2-layer RNN with input dimension=300, hidden-layer dimension=4096, and output dimension=2. The lower-level variable $w$ is initialized uniformly from the $(-1.0, 1.0)$ range.

**Parameter selection for the experiments in Figure 1(b) and Figure 2(b):** We compare our proposed algorithm BO-REP, with other baseline algorithms. We conduct a grid search for lower-level and upper-level learning rates in the range of $\{0.0001, 0.0005, 0.001, 0.005, 0.01, 0.05, 0.1\}$ and find the best parameter setting for all the baseline algorithms. Specifically, we choose the best lower-level learning rates as $0.001$ for StocBio, $0.01$ for TTSA, $0.05$ for SOBA and SABA, $0.05$ for MA-SOBA, and $0.001$ for BO-REP. The upper-level step size $0.0001$ is applied to all the algorithms. For SOBA, SABA, MA-SOBA, and BO-REP, the best learning rates for solving the linear system are $0.05$, $0.05$, and $0.01$ respectively, which are searched in the range of $\{0.001, 0.005, 0.01, 0.05, 0.1\}$. For F$^2$SA, the best combination for step sizes of its three variables is $(0.01, 0.01, 0.0001)$, which is tuned in

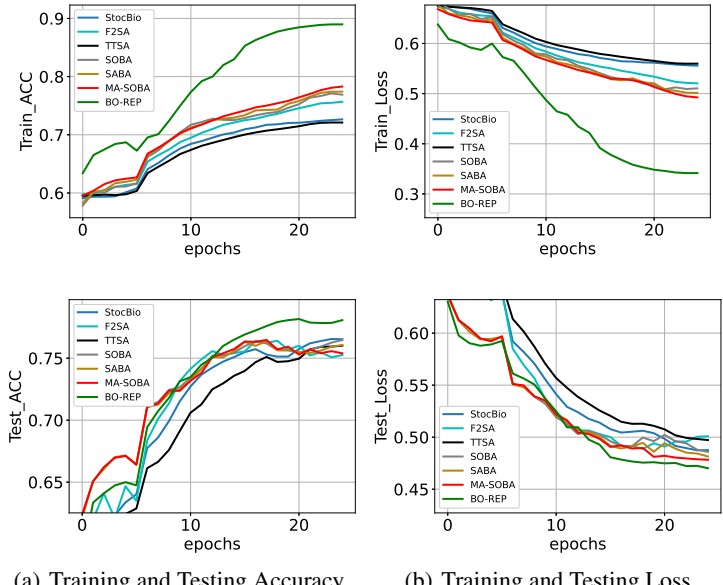

(a) Training and Testing Accuracy    (b) Training and Testing Loss

Figure 3: (a) Accuracy of data hyper-cleaning on Sentiment140 with corruption rate $p = 0.1$. (b) Loss of data hyper-cleaning on Sentiment140 with corruption rate $p = 0.1$.

the range of $\{0.0001, 0.0005, 0.001, 0.005, 0.01, 0.05, 0.1\}$. In addition, the double-loop algorithm StocBio fixes inner loops as 5 for lower-level updates.

For BO-REP, the updating interval $I$ for the lower-level variable $\boldsymbol{\theta}_i$ is set as 2, and the number of iterations ($N$) for each periodical update is 3, the ball radius $R$ for projection is $0.5$. All algorithms fix batch size as $64$. Other hyperparameter settings, including momentum parameter (for MA-SOBA and BO-REP), step size for solving linear system (for SOBA, MA-SOBA, BO-REP), and the lagrangian multiplier setting (for F$^2$SA) keep the same as Section F.1.

## F.3   EXPERIMENTAL DETAILS OF DATA HYPER-CLEANING

We conduct the experiments of the data hyper-cleaning task on Sentiment140 (Go et al., 2009) for binary text classification. Since data labels consist of two classes of emotions, positive and negative, we flip each label in the training set to its opposite class with probability $p$ (set as $0.1$ and $0.3$, respectively).

A two-layer RNN with the same architecture as that in Section F.2 is adopted as the classifier, whose parameters $\boldsymbol{w}$ are lower-level variables. The upper-level variable $\boldsymbol{\lambda}$ is the weight vector corresponding to each training sample. In practice, we initialize each sample weight $\lambda_i = 1.0$.

**Parameter selection for the experiments in Figure 1(c) and Figure 2(c):** We use a grid search for all algorithms to choose the lower-level and upper-level step size in the $\{0.01, 0.05, 0.1\}$. The best combinations of lower-level and upper-level step size are $(0.05, 0.05)$ for StocBio, $(0.05, 0.01)$ for TTSA, $(0.1, 0.05)$ for SOBA and SABA, $(0.1, 0.05)$ for MA-SOBA, and $(0.05, 0.05)$ for BO-REP. F$^2$SA chooses $(0.05, 0.05, 0.01)$ for updating its three decision variables. In addition, the learning rates for solving linear system $\boldsymbol{z}$ in SOBA, SABA, MA-SOBA, and BO-REP are all fixed as $0.05$ chosen from the range of $\{0.01, 0.05, 0.1\}$. The number of inner loops for StocBio is set as 5, which is chosen from $\{3, 5, 10\}$. For BO-REP, The updating interval $I$ and the iterations $N$ for lower-level variable $\boldsymbol{w}$ are fixed as 2 and 3, respectively, the ball radius $R$ for projection is $0.5$. Batch size is fixed to $512$ for all the algorithms. Other experimental hyperparameters, including momentum parameters (for MA-SOBA, BO-REP) and the lagrangian multiplier setting (F$^2$SA) remain the same as Section F.1.

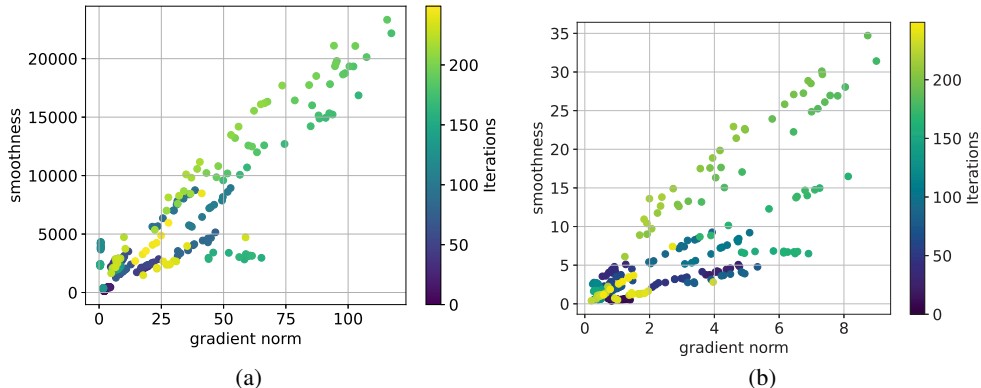

(a)                                                                 (b)

Figure 4: (a) Local gradient Lipschitz constant of the upper-level variable vs. its gradient norm along the training iterations for an RNN in the experiment of hyper-representation. (b) Local gradient Lipschitz constant of the lower-level variable vs. its gradient norm along the training iterations for an RNN in the experiment of hyper-representation.

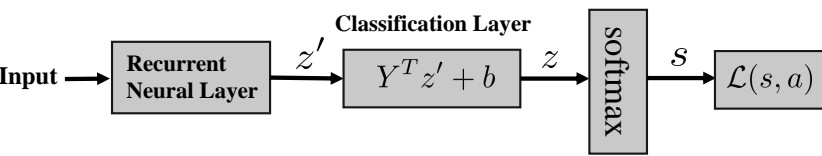

Figure 5: The model structure for Hyper-representation. The upper-level variable is the parameter of the recurrent neural layer, and the lower-level variable is the parameter of the classification layer.

## G   VERIFICATION OF RELAXED SMOOTHNESS (ASSUMPTION 1) FOR RECURRENT NEURAL NETWORKS

In this section, we empirically verified that the Recurrent Neural Network model satisfies Assumption 1. In Figure 4, we plot the estimated smoothness at different iterations during training neural networks. In particular, we conduct the hyper-representation experiments to verify Assumption 1. We adopt the same recurrent neural network as that in Section 4.1. The lower-level variable is the parameter of the last linear layer, and the upper-level variable is the parameter of the previous layers (2 hidden layers). In each training iteration, we calculate the gradient norm w.r.t. the upper-level variable $\|\nabla_{\boldsymbol{x}} f(\boldsymbol{u})\|$ and the lower-level variable $\|\nabla_{\boldsymbol{y}} f(\boldsymbol{u})\|$, and use the same method as described in Appendix H.3 in Zhang et al. (2020b) to estimate the smoothness constants of $\boldsymbol{x}$ and $\boldsymbol{y}$. From Figure 4, we can find that the smoothness parameter scales linearly in terms of gradient norm for both layers. This verifies the Assumption 1 empirically. Note that these results are consistent with the results in the literature, e.g., Figure 1 in Zhang et al. (2020b) and Figure 1 in Crawshaw et al. (2022).

## H   VERIFICATION OF ASSUMPTION 2(i)

In this section, we theoretically prove that Assumption 2(i), i.e. $\|\nabla_y f(x, y^*(x))\| \leq M$, holds in the practical case of hyper-representation. In this case, we define the cross entropy loss as

$$\mathcal{L}(a, s) = -\sum_{i=1}^{C} a_i \log(s_i),$$

where $C$ denotes the number of class, $a = (a_1, a_2, \ldots, a_C)$ is the one-hot encoded label and $s = (s_1, s_2, \ldots, s_C)$ is the probability distribution generated by the softmax layer. We also define $Y$ (we can view notation $y$ as $y = \text{vec}(Y)$ in the main text) and $b$ as the weight and bias, $z'$ and $z$ as the input and output of the last layer (classification layer). So we have $z = Y^\top z' + b$. Figure 5 illustrates the model structure and the meanning of symbols.

First we calculate $\frac{\partial s_i}{\partial z_j}$. By chain rule, we have

$$\frac{\partial s_i}{\partial z_j} = s_i \frac{\partial}{\partial z_j} \log(s_i) = s_i \frac{\partial}{\partial z_j} \log \left( \frac{e^{z_i}}{\sum_{l=1}^n e^{z_l}} \right) = s_i \frac{\partial}{\partial z_j} \left( z_i - \log \left( \sum_{l=1}^n e^{z_l} \right) \right)$$

$$= s_i \left( \frac{\partial z_i}{\partial z_j} - \frac{\partial}{\partial z_j} \log \left( \sum_{l=1}^n e^{z_l} \right) \right) = s_i \left( \mathbb{1}_{\{i=j\}} - \frac{1}{\sum_{l=1}^n e^{z_l}} \left( \frac{\partial}{\partial z_j} \sum_{l=1}^n e^{z_l} \right) \right)$$

$$= s_i \left( \mathbb{1}_{\{i=j\}} - \frac{e^{z_j}}{\sum_{l=1}^n e^{z_l}} \right) = s_i \left( \mathbb{1}_{\{i=j\}} - s_j \right).$$

Next we calculate $\frac{\partial \mathcal{L}}{\partial z_j}$. By chain rule, we have

$$\frac{\partial \mathcal{L}}{\partial z_j} = -\frac{\partial}{\partial z_j} \sum_{i=1}^C a_i \log(s_i) = -\sum_{i=1}^C a_i \frac{\partial}{\partial z_j} \log(s_i) = -\sum_{i=1}^C \frac{a_i}{s_i} \frac{\partial s_i}{\partial z_j}$$

$$= -\sum_{i=1}^C \frac{a_i}{s_i} s_i \left( \mathbb{1}_{\{i=j\}} - s_j \right) = -\sum_{i=1}^C a_i \left( \mathbb{1}_{\{i=j\}} - s_j \right) = \sum_{i=1}^C a_i s_j - \sum_{i=1}^C a_i \mathbb{1}_{\{i=j\}}$$

$$= s_j \sum_{i=1}^C a_i - a_j = s_j - a_j,$$

where we use $\sum_{i=1}^C a_i = 1$ in the last line. Hence we have

$$\frac{\partial \mathcal{L}}{\partial z} = s - a.$$

Again, by chain rule we have

$$\frac{\partial \mathcal{L}}{\partial Y} = \frac{\partial \mathcal{L}}{\partial z} \frac{\partial z}{\partial Y} = (s - a) z'^\top$$

Therefore, we conclude that

$$\left\| \frac{\partial \mathcal{L}}{\partial Y} \right\| \le \|s - a\| \|z'\| \le (\|s\| + \|a\|) \|z'\| \le 2\|z'\|,$$

where we use $\|s\| \le 1$ and $\|a\| = 1$ for any $s$ and $a$.

# I  PERFORMANCE COMPARISON IN TERMS OF RUNNING TIME

For a fair comparison of performance, we compare our proposed algorithm with other single-loop and double-loop algorithms in terms of running time, the result is shown in Figure 6. To accurately evaluate each algorithm, we use the machine learning framework PyTorch 1.13 to run each algorithm individually on an NVIDIA RTX A6000 graphics card, and record its training and test loss. As we can observe from the figure, our algorithm (BO-REP) is much faster than all other baselines in the experiment of Hyper-representation (Figure 6(a)) and Data Hyper-Cleaning (Figure 6(c)). For the hyperparameter optimization experiment (Figure 6(b)), our algorithm is slightly slower at the very beginning, but quickly outperforms all other baselines. This means that our algorithm indeed has better runtime performance than the existing baselines in bilevel optimization.

# J  ABLATION STUDY FOR HYPERPARAMETERS

## J.1  ABLATION STUDY FOR LOWER-LEVEL UPDATE PERIOD $I$ AND ITERATIONS $N$

We conduct careful ablation studies to explore the impact of hyperparameter $I$ (the update period of the lower-level variable) and $N$ (the number of iterations for updating the lower-level variable

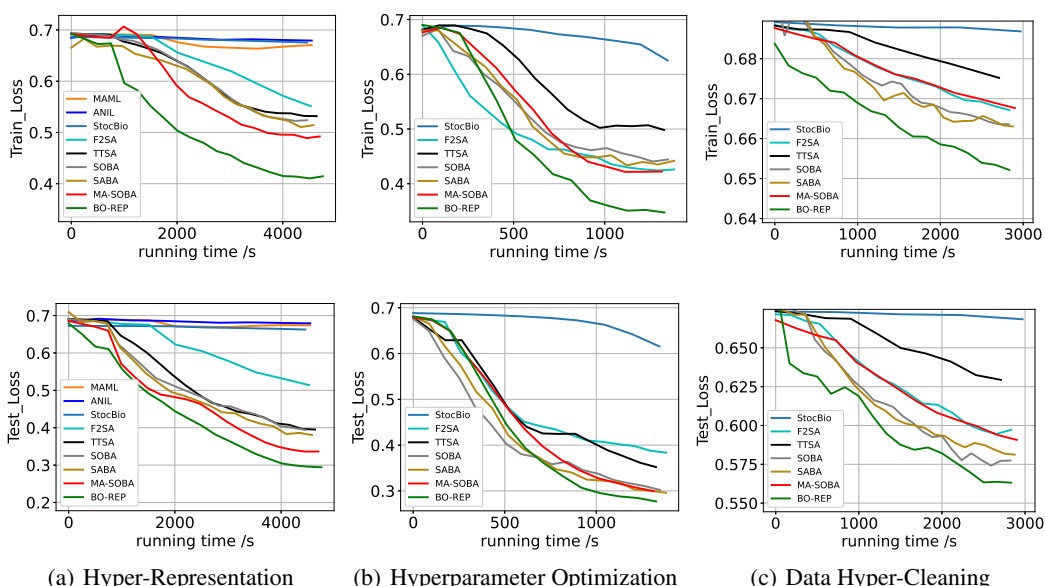

(a) Hyper-Representation     (b) Hyperparameter Optimization     (c) Data Hyper-Cleaning

Figure 6: Comparison of various bilevel optimization algorithms w.r.t running time (s): (a) results of Hyper-representation on Amazon Review Dataset. (b) results of hyperparameter optimization on Amazon Review Dataset. (c) results of data hyper-cleaning on Sentiment140 Dataset with noise rate $p = 0.3$.

Table 3: Test accuracy vs. Projection radius $R$

| Radius | $R$=0.001 | $R$=0.005 | $R$=0.01 | $R$=0.05 | $R$=0.10 | $R$=0.50 | $R$=1.00 | $R$=5.00 |
|---|---|---|---|---|---|---|---|---|
| Test Accuracy | 65.06% | 67.64% | 70.76% | 81.22% | 86.84% | 88.84% | 88.84% | 88.84% |

during each period). The experimental results in Figure 7(a) show that the performance of BO-REP algorithm decreases slightly when increasing the update period $I$ from $2$ to $8$ while fixing inner iterations $N$. That demonstrates empirically our algorithm is not sensitive to the update period hyperparameter $I$. When the value of $I$ is too large ($I \geq 16$), we observe a significant performance degradation. In our experiments in the main text, we choose $I = 2$ for all experiments and get good performance universally. The ablation result for inner iterations $N$ is shown in Figure 7(b), where the update period $I$ is fixed. The figure shows that the performance of BO-REP algorithm would increase as the number of inner iterations increases. The algorithm can achieve the best performance when $N \geq 5$. In our experiments, it is good enough to choose $N = 3$ to achieve good performance universally for all tasks.

Due to these ablation studies, it means that the algorithm does not need lots of tuning efforts despite these hyperparameters (e.g., $I$ and $N$): some default values of $I$ and $N$ (e.g., $I = 2$, $N = 3$) work very well for a wide range of tasks in practice.

## J.2   ABLATION STUDY FOR THE RADIUS $R$ OF PROJECTION BALL

We experimentally explore how the ball radius $R$ affects the algorithm's performance. We set the ball radius as $\{0.001, 0.005, 0.01, 0.05, 0.10, 0.50, 1.00, 5.00\}$ respectively, and then conduct the bilevel optimization on Hyper-representation tasks. We keep all the other hyperparameters the same as in Section 4.1. The result is shown in Table 3. When the ball radius $R$ is too small (i.e., $R < 0.10$), the performance significantly drops, possibly due to the overly restricted search space for the lower-level variable. When the ball radius $R \geq 0.50$, the performance becomes good and stable. In our experiments in the main text, we choose $R = 0.5$.

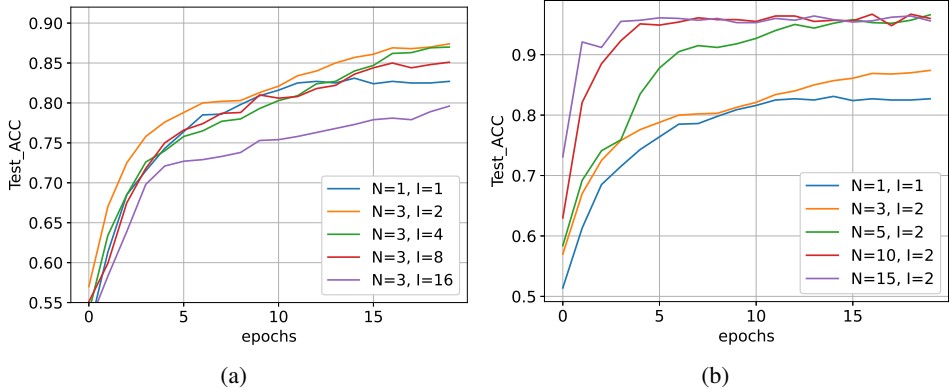

(a)                                                        (b)

Figure 7: Ablation study of the update period $I$ and the number of iterations for updating lower-level variable during each period $N$ (a) The performance of hyper-representation with different update period $I$. (b) The performance of hyper-representation with different numbers of update iterations $N$.

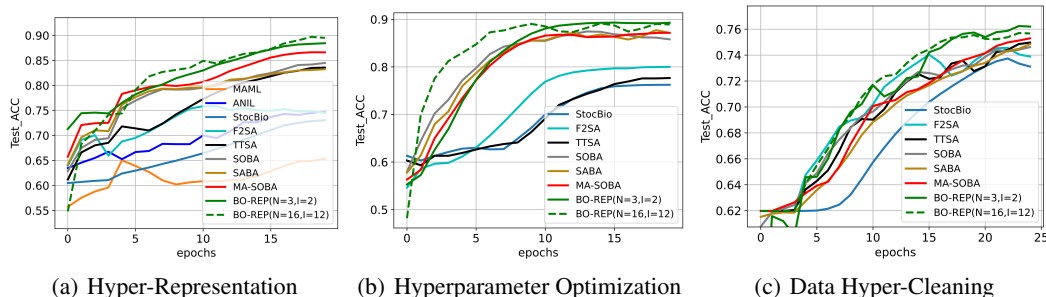

(a) Hyper-Representation       (b) Hyperparameter Optimization       (c) Data Hyper-Cleaning

Figure 8: Comparison results with the larger value of update iterations $N$ and update period $I$ on Hyper-Representation. (b) Comparison results with the larger value of update iterations $N$ and update period $I$ on Hyperparameter Optimization. (c) Comparison results with the larger value of update iterations $N$ and update period $I$ on Data Hyper-Cleaning.

## K    EXPERIMENTAL RESULTS WITH LARGE $I$ AND LARGE $N$

In this section, we further explore the setting of hyperparameters $N$ and $I$. In particular, we evaluate the algorithm performance on the larger value of $N$ and $I$ (i.e., $N = 16, I = 12$) compared with the value we used in the experiments described in main text (i.e., $N = 3, I = 2$), which may better fit our theory. The results are presented in Figure 8, where Figure 8(a), (b), (c) show that the compared test accuracy on three different tasks, respectively. We can observe that the algorithm performance with the large value of $N$ and $I$ (green dash line) is almost the same as (or even better than) the original setting (green solid line). The larger number of $N$ can compensate for performance degradation induced by a long update period $I$. In particular, the new results in Figure 8(b), (c) (green dash lines) are obtained with a slightly smaller lower-level learning rate ($8 \times 10^{-4}$ in (b) and $5 \times 10^{-3}$ in (c)) than the original setting (green solid lines with $1 \times 10^{-3}$ in (b) and $5 \times 10^{-2}$ in (c)). However, the setting of $N = 3, I = 2$ is good enough in our experiments to achieve good performance for all the tasks.

## L    OPTIMALITY OF OUR COMPLEXITY RESULTS

In this section, we demonstrate why our proposed algorithm is optimal up to logarithmic factors if no additional assumptions are imposed. We first introduce the definition of mean-squared smoothness (Arjevani et al., 2023) and individual smoothness (Cutkosky & Orabona, 2019) for single-level optimization problems, and then we discuss how these assumptions are being used in bilevel opti-

mization literature to achieve $\widetilde{O}(1/\epsilon^3)$ oracle complexity, and at last we demonstrate our proposed algorithm with $\widetilde{O}(1/\epsilon^4)$ is indeed optimal up to logarithmic factors under current assumptions in this paper.

For single-level problems, given differentiable objective function $\Psi : \mathbb{R}^d \to \mathbb{R}$, we say the function $\Psi$ satisfies *mean-squared smoothness* property (formula (4) in Arjevani et al. (2023)) if for any $x, y \in \mathbb{R}^d$ and any $\xi \sim P_\xi$,

$$\mathbb{E}_\xi[\|\nabla\Psi(x;\xi) - \nabla\Psi(y;\xi)\|^2] \leq L^2\|x-y\|^2, \tag{85}$$

where we use $\xi$, $\nabla\Psi(x;\xi)$ and $L$ (the corresponding notations in Arjevani et al. (2023) are $z$, $g(x, z)$ and $\bar{L}$, please check formula (4) in Arjevani et al. (2023) for details) to denote the random data sample, the stochastic gradient estimator and the mean-squared smoothness constant.

A slightly stronger condition than mean-squared smoothness is the *individual smoothness* property (please check the statement *"We assume that $f(x, \xi_t)$ is differentiable, and $L$-smooth as a function of $x$ with probability 1."* in Section 3 of Cutkosky & Orabona (2019) for details). We say function $\Psi$ has individual smoothness property if for any $x, y \in \mathbb{R}^d$ and any $\xi \sim P_\xi$,

$$\|\nabla\Psi(x;\xi) - \nabla\Psi(y;\xi)\| \leq L\|x-y\|. \tag{86}$$

For bilevel problems, in order to obtain $\widetilde{O}(1/\epsilon^3)$ oracle complexity bound, Yang et al. (2021), Guo et al. (2021) and Khanduri et al. (2021) actually require individual smoothness assumption (Assumption (86)) jointly in $(x, y) \in \mathbb{R}^{d_x} \times \mathbb{R}^{d_y}$ for both upper-level (i.e., outer function $f(x, y)$) and lower-level problems (i.e., inner function $g(x, y)$). To be more specific, for upper-level problem they require for any $u = (x, y), u' = (x', y') \in \mathbb{R}^{d_x} \times \mathbb{R}^{d_y}$ and any $\xi$,

$$\|\nabla_x f(u;\xi) - \nabla_x f(u;\xi)\| \leq L_{f_x}\|u-u'\|,$$
$$\|\nabla_y f(u;\xi) - \nabla_y f(u;\xi)\| \leq L_{f_y}\|u-u'\|,$$

and for lower-level problem they require for any $u = (x, y), u' = (x', y') \in \mathbb{R}^{d_x} \times \mathbb{R}^{d_y}$ and any $\zeta$,

$$\|\nabla_y g(u;\zeta) - \nabla_y g(u;\zeta)\| \leq L_{g_y}\|u-u'\|.$$

These three inequalities above can be viewed as definition of individual smoothness property under bilevel optimization setting. For more details, please check Assumption 2 in Section 3.1 of Yang et al. (2021), Assumption 2 in Section 2 and Assumption 4 in Section 4 of Guo et al. (2021), Assumption 2 in Section 2 of Khanduri et al. (2021). *Please note that our paper does not assume any of these assumptions.*

Notably, all the bilevel optimization literature (Yang et al., 2021; Guo et al., 2021; Khanduri et al., 2021) with $\widetilde{O}(1/\epsilon^3)$ complexity use individual smoothness assumption, which is stronger than the mean-squared smoothness. Also, their algorithm design depends on the STORM technique (Cutkosky & Orabona, 2019). In addition, it is shown in Arjevani et al. (2023) that without this assumption, $\Omega(1/\epsilon^4)$ complexity is necessary for stochastic first-order optimization algorithms for single problems. This indicates that our complexity is optimal up to logarithmic factors for the bilevel problems we considered in this paper.

Let us give more details to explain this. We consider potentially non-convex single-level optimization problems as considered in (Arjevani et al., 2023). Given differentiable objective function $\Psi : \mathbb{R}^d \to \mathbb{R}$, our goal is to find an $\epsilon$-stationary point using stochastic first-order algorithms. Assume the algorithms access the function $\Psi$ through a stochastic first-order oracle consisting of a noisy gradient estimator $\nabla\Psi : \mathbb{R}^d \times \Xi \to \mathbb{R}^d$ and distribution $P_\xi$ on $\Xi$ satisfying

$$\mathbb{E}_{\xi \sim P_\xi}[\nabla\Psi(x;\xi)] = \nabla\Psi(x) \quad \text{and} \quad \mathbb{E}_{\xi \sim P_\xi}[\|\nabla\Psi(x;\xi) - \nabla\Psi(x)\|^2] \leq \sigma^2 \tag{87}$$

Also we make the standard assumption that the objective $\Psi$ has bounded initial suboptimality and $L$-smoothness property,

$$\Psi(x_0) - \inf_{x \in \mathbb{R}^d} \Psi(x) \leq \Delta \quad \text{and} \quad \|\nabla\Psi(x) - \nabla\Psi(y)\| \leq L\|x-y\|, \tag{88}$$

where $x_0$ is the initialization point for the algorithm.

For potentially non-convex single-level optimization problems, Theorem 3 in Arjevani et al. (2023) states that

- Under Assumptions (87) and (88), any stochastic first-order methods requires $\Omega(1/\epsilon^4)$ queries to find an $\epsilon$-stationary point in the worst case (please check Contribution 1 in Section 1.1, and formula (23) in Theorem 3 of Arjevani et al. (2023) for details);

- Under Assumptions (87) and (88), plus mean-squared smoothness assumption, any stochastic first-order methods require $\Omega(1/\epsilon^3)$ queries to find an $\epsilon$-stationary point in the worst case (please check Contribution 2 in Section 1.1, and formula (24) in Theorem 3 of Arjevani et al. (2023) for details).

Note that our problem class is more expressive than the function class considered in Arjevani et al. (2023) and hence our problem is harder. This is because standard smoothness is a special case of relaxed smoothness and single-level optimization is a special case of bilevel optimization. For example, if we consider an easy case where the upper-level function does not depend on the lower-level problem (e.g., $\Phi(x) = f(x)$ such that $\Phi$ is independent of $y^*(x)$) and does not have mean-squared smoothness, then the $\Omega(1/\epsilon^4)$ lower bound in Arjevani et al. (2023) can be applied in our setting. Therefore the $\widetilde{O}(1/\epsilon^4)$ complexity achieved in this paper is already optimal up to logarithmic factors.

