# OpenReview forum: "Bilevel Optimization under Unbounded Smoothness: A New Algorithm and Convergence Analysis"
_ICLR.cc/2024/Conference — ICLR 2024 spotlight_

### Official Review · Reviewer_gGYZ · 2023-10-27

**Soundness:** 3 good
**Presentation:** 2 fair
**Contribution:** 2 fair
**Rating:** 6
**Confidence:** 4

**Summary:**

This paper considers the bilevel optimization under unbounded smoothness and proposes a new algorithm based on the SOBA method. This paper gives the convergence analysis under the unbounded smoothness and the experimental results demonstrate the superiority of the proposed method.

**Strengths:**

1. Under the unbounded smoothness, this paper obtains a similar convergence rate as that in other papers.
2. The experimental results demonstrate the superiority of the proposed method.

**Weaknesses:**

1. Too many loops in Algorithm 1,2 which may lead to time-consuming.
2. Some symbols are confusing. Both Algorithms 1, and 2 use K. In lemmas, what is $K_0$?
3. Too many hyperparameters in algorithms. More experimental results of different hyperparameters are needed.

**Questions:**

1. How will the iteration numbers affect the convergence performance? Is the inner loop necessary? Because this method seems to be a momentum method based on SOBA, why author use an inner loop? why not just update $y$ for a single step?
2. Why only use the SOBA in [1]. SABA also proposed in [1]. I think it should be compared.
3. How to ensure the problem considered in experiments is unbounded smooth? Because the applications considered in this paper are usually considered in other papers with bound smoothness assumption.


[1] Dagréou M, Ablin P, Vaiter S, et al. A framework for bilevel optimization that enables stochastic and global variance reduction algorithms[J]. Advances in Neural Information Processing Systems, 2022, 35: 26698-26710.

---

> ### Author Response · Authors · 2023-11-16
> **Thank you for taking the time to review our paper and provide insightful comments. We have replied to each of your thoughts below.**
>
> **Weakness 1. Too many loops in Algorithm 1,2 which may lead to time-consuming.**
>
> **A**: There is a minor issue in the pseudo-code for Algorithm 2 in the original version of the paper and we are sorry for the ambiguity and confusion brought to the readers. The outer loop in Algorithm 2 with the iteration variable $k$ spanning from $0$ to $K-1$ is indeed repeatedly defined, and we already removed it in the new version of our paper. Algorithm 1 is actually a double-loop algorithm instead of a triple-loop one. Notably, we only need to execute the inner loop occasionally since updating for the lower-level variable is periodic rather than in each iteration, which helps reduce the running time.
>
> **Weakness 2. Some symbols are confusing. Both Algorithms 1, and 2 use $K$. In lemmas, what is $K_0$?**
>
> **A**: Please check the algorithm framework in new version of our paper, we fix a typo in the pseudo-code for Algorithm 2 (remove the unnecessary loop in line 1). As illustrated before, we only need $K$ in Algorithm 1, and we do not need it in Algorithm 2. As for the notation $K_0$ in lemmas, please check "Remark" paragraph below Assumption 2 in the main text, we mentioned that $K_0$ and $K_1$ are relaxed smoothness parameters for the function $\Phi(x)$. For the definition of $K_0$ and $K_1$, please check Lemma 9 in Appendix C for details, where we define $K_0$ and $K_1$ in equation (16) and we prove that function $\Phi(x)$ satisfies the standard relaxed smoothness condition.
>
> **Weakness 3. Too many hyperparameters in algorithms. More experimental results of different hyperparameters are needed.**
>
> **A**:  Nearly all papers about bilevel optimization have many hyperparameters, and our problem setting is more difficult to solve due to Assumption 1, i.e., $(L_{x,0},L_{x,1}, L_{y,0}, L_{y,1})$-smoothness assumption. We conduct careful ablation studies to explore the impact of hyperparameter $I$ (the update period of the lower-level variable) and $N$ (the number of iterations for updating the lower-level variable during each period). The experimental results in Figure 7 (a) (in Appendix J) show that the performance of BO-REP algorithm decreases slightly when increasing the update period $I$ from $2$ to $8$ while fixing the number of iterations in each update period $N$. That demonstrates empirically our algorithm is not sensitive to the update period hyperparameter $I$. When the value of $I$ is too large ($I \geq 16$), we observe a significant performance degradation.  In our experiments in the main text, we choose $I=2$ for all experiments and get good
> performance universally. The ablation result for $N$ is shown in Figure 6 (b), where the update period $I$ is fixed.  The figure shows that the performance of BO-REP algorithm would be better as $N$ increases.  The algorithm can achieve the best performance when $N\geq 5$. In our experiments, it is good enough to choose $N=3$ to achieve good performance universally for all tasks.
>
> Due to these ablation studies, it means that the algorithm does not need lots of tuning efforts despite these hyperparameters (e.g., $I$ and $N$ ): some default values of $I$ and $N$ (e.g., $I = 2, N = 3$) work very well for a wide range of tasks in practice.

---

> > ### Author Response · Authors · 2023-11-16
> > **Thank you for taking the time to review our paper and provide insightful comments. We have replied to each of your thoughts below. (cont'd)**
> >
> > **Question 1. How will the iteration numbers affect the convergence performance? Is the inner loop necessary? Because this method seems to be a momentum method based on SOBA, why author use an inner loop? why not just update $y$ for a single step?**
> >
> > **A**: Thank you for your insightful question. The inner loop for periodic updates is quite important for convergence analysis. We will explain why we cannot use single-loop based on our current proof strategy. The main difficulty of the paper comes from the bias term $\\|\mathbb{E}\_k[\widehat{\nabla}\Phi(x_k)]-\nabla\Phi(x_k)\\|$ of the hypergradient estimator, whose upper bound depends on both $L_{x,1}\\|y_k-y_k^*\\|\\|\nabla\Phi(x_k)\\|$ and $\\|y_k-y_k^*\\|$. Please check the proof of Lemma 4 in Appendix D.6 (in the middle of page 33, between inequality (ii) and (iii)) for details. Due to the fact that both $\\|y_k-y_k^*\\|$ and $\\|\nabla \Phi(x_k)\\|$ are measurable with respect to $\mathcal{F}_{k-1}$, we have to use high probability analysis for $y_k$ to disentangle these two terms.
> >
> > If we use single-loop to solve our algorithm, then for every iteration we update $y_k$ just via one step of stochastic gradient descent. By the high probability analysis of SGD for the lower-level problem (i.e., Lemma 12 in Appendix D.2), if we choose $
> > \lambda = \max\left(\sqrt{3\ln\left(\frac{2K}{\delta}\right)}, \ln\left(\frac{2K}{\delta}\right)\right)$, $\alpha_s=\gamma$ and $T_s=1$ (i.e., single loop) in formula (19) and (20) in Lemma 12, then with probability at least $1-\delta/K$, we have
> >
> > $$
> >  \frac{\mu}{2}\\|y_1-y_0^*\\|^2 \leq g(x_0,y_1)-g(x_0,y_0^*)  \leq \frac{1}{\gamma}D^2+2\sigma_{g,1}^2\gamma+\lambda\left(2\sqrt{2}D\sigma_{g,1}+2\sigma_{g,1}^2\gamma\right) \\
> > $$
> > $$
> > =\frac{1}{\gamma}D^2+2\sigma_{g,1}^2(\lambda+1)\gamma+2\sqrt{2}\lambda\sigma_{g,1}D = 2\sqrt{2(\lambda+1)}\sigma_{g,1}D+2\sqrt{2}\lambda\sigma_{g,1}D \\
> > $$
> > $$
> > = 2\sqrt{2}(\lambda+\sqrt{\lambda+1})\sigma_{g,1}D,
> > $$
> > where $D=\epsilon/8K_0$ is the upper bound for $\|y_0-y_0^*\|$, and we choose $\gamma=\frac{D}{\sigma\sqrt{2(\lambda+1)}}$ that minimizes the value of the right-hand-side of the inequality. Then we have
> > $$
> > \\|y_1-y_1^*\\| \leq \\|y_1-y_0^*\\| + \\|y_0^*-y_1^*\\| \leq \sqrt{4\sqrt{2}(\lambda+\sqrt{\lambda+1})\sigma_{g,1}/\mu}\sqrt{D} + \frac{C_{g_{xy}}}{\mu}\eta
> > $$
> > $$
> >  = \sqrt{\sqrt{2}(\lambda+\sqrt{\lambda+1})\sigma_{g,1}/2K_0\mu}\sqrt{\epsilon} + \frac{C_{g_{xy}}}{\mu}\eta = \widetilde{\mathcal{O}}(\epsilon^{1/2}),
> > $$
> > where the last equation holds because $\eta=\mathcal{O}(\epsilon^3)$ and hence the second term is dominated by the first term when $\epsilon$ is small. Then repeat the same process, with probability at least $1-2\delta/K$, we have $\\|y_2-y_2^*\\|=\widetilde{\mathcal{O}}(\epsilon^{1/4})$. Generally speaking, we have with probability at least $1-k\delta/K$, $\\|y_k-y_k^*\\|=\widetilde{\mathcal{O}}(\epsilon^{1/2^k})$. Since for $\epsilon<1$, we have $\epsilon < \epsilon^{1/2} < \epsilon^{1/2^2} \leq \cdots \leq \epsilon^{1/2^K}$, this means that with high probability, $\frac{1}{K}\sum_{k=0}^{K-1}\\|y_k-y_k^*\\| = \frac{1}{K}\sum_{k=0}^{K-1}\mathcal{O}(\epsilon^{1/2^k})$, and we cannot conclude that $\frac{1}{K}\sum_{k=0}^{K-1}\\|y_k-y_k^*\\| = \mathcal{O}(\epsilon)$ when $K$ is large (note that $\mathcal{O}(\epsilon)$ bound for this quantity is required in our proof to obtain the $\widetilde{\mathcal{O}}(\epsilon^{-4})$ complexity). Therefore the single loop algorithm does not work if we follow our current proof strategy.
> >
> > We also conduct the ablation study to figure out how the iteration numbers affect the performance. The empirical result is shown in Figure 6 (b), where $N$ is the iteration numbers and $I$ is the update period. The ablation experiment is implemented on the hyper-representation task. The blue line (with $N=1, I=1$) denotes the result of the single update for $y$. Then we fix the update period $I=2$ (which is fixed as $2$ in Section 4) and increase the number of inner loops. The result shows that the performance of BO-REP algorithm would increase as the number of inner loops. The algorithm can achieve the best performance when $N\geq 5$. In our experiments, it is good enough to choose $N=3$ to achieve good performance universally for all tasks.

---

> > > ### Author Response · Authors · 2023-11-16
> > > **Thank you for taking the time to review our paper and provide insightful comments. We have replied to each of your thoughts below. (cont'd)**
> > >
> > > **Question 2. Why only use the SOBA in [1]. SABA also proposed in [1]. I think it should be compared.**
> > >
> > > **A**:
> > > We have added the SABA baseline in our experiments, please refer to Figure 1 in the new version. SABA slightly improves over SOBA but it is still much worse than our algorithm BO-REP.
> > >
> > >
> > > **Question 3. How to ensure the problem considered in experiments is unbounded smooth? Because the applications considered in this paper are usually considered in other papers with bound smoothness assumption.**
> > >
> > > **A**: First, we want to emphasize that the unbounded smoothness assumptions were empirically observed in many prior works [3, 2] for RNNs, LSTMs and Transformers. Second, we have empirically verified that the Recurrent Neural Network model satisfies Assumption 1 in the revised version. In Figure 4, we plot the estimated smoothness at different iterations during training neural networks. In particular, we conduct the hyper-representation experiments to verify Assumption 1. We adopt the same recurrent neural network as that in section 4.1. The lower-level variable is the model parameter of the last linear layer, and the upper-level variable is the model parameter of the previous layers (2 hidden layers). In each training iteration, we calculate the gradient norm w.r.t. the upper variable $\\|\nabla_x f(x, y)\\|$ and the lower variable $\\|\nabla_y f(x, y)\\|$, and use same the method as described in Appendix H.3 in [3] to estimate the smoothness of $x$ and $y$. We can get consistent results which are shown in Figure 1 in [3] and Figure 1 in [2]. In particular, we can find that the smoothness parameter scales linearly in terms of gradient norm for both layers. This verifies the Assumption 1 empirically.
> > >
> > >
> > > [1] *Mathieu Dagréou, Pierre Ablin, Samuel Vaiter, and Thomas Moreau. A framework for bilevel optimization that enables stochastic and global variance reduction algorithms. Advances in Neural Information Processing Systems, 35:26698-26710, 2022.*
> > >
> > > [2] *Michael Crawshaw, Mingrui Liu, Francesco Orabona, Wei Zhang, and Zhenxun Zhuang. Robustness to unbounded smoothness of generalized signsgd. Advances in neural information processing systems, 2022.*
> > >
> > > [3] *Jingzhao Zhang, Tianxing He, Suvrit Sra, and Ali Jadbabaie. Why gradient clipping accelerates training: A theoretical justification for adaptivity. International Conference on Learning Representations, 2020b.*

---

> ### Author Response · Authors · 2023-11-21
> **looking forward to post-rebuttal feedback!**
>
> Dear Reviewer gGYZ,
>
> Thank you for reviewing our paper. We have carefully answered your concerns on the nested loops, single-loop, hyperparameters, the new baseline SABA, and verification of unbounded smoothness.
>
> Please let us know if our answers accurately address your concerns. If our response resolves your concerns, we kindly ask you to consider raising the rating of our work. Thank you very much for your time and efforts! We would like to discuss any additional questions you may have.
>
> Best,
> Authors

---

> > ### Comment · Reviewer_gGYZ · 2023-11-22
> > **Response**
> >
> > Thanks for your further responses.  I will increase my score to 6.

---

> > > ### Author Response · Authors · 2023-11-22
> > > **Thank you!**
> > >
> > > Dear Reviewer gGYZ,
> > >
> > > We are glad that our responses addressed your concerns. Thank you for reviewing our paper.
> > >
> > > Best,
> > > Authors

---

### Official Review · Reviewer_ZUWL · 2023-10-30

**Soundness:** 2 fair
**Presentation:** 3 good
**Contribution:** 3 good
**Rating:** 6
**Confidence:** 3

**Summary:**

This paper investigates a bilevel stochastic optimization problem in which the gradient of the upper-level objective function is potentially unbounded, and the lower-level objective function is strongly convex.
The authors introduce a new algorithm, which integrates the normalized momentum technique for updating the upper-level variable and employs initialization refinement as well as periodic updates techniques for updating the lower-level variable. The authors also offer a theoretical complexity analysis. To substantiate the efficacy of the proposed algorithm, the paper includes a set of numerical experiments.

**Strengths:**

1. The paper is clear, well organized, and easy to follow.

2. The incorporation of the normalized momentum update for the upper-level variable, as well as the utilization of initialization refinement and periodic updates for the lower-level variable within the proposed algorithm weaken the bounded smoothness requirement of the upper-level objective function.

3. Simulations have shown the empirical advantage of the proposed algorithm.

**Weaknesses:**

1. The theoretical complexity result of the proposed algorithm, Theorem 1, lacks persuasiveness. Specifically, the choice of the iteration number $N$ in Algorithm 2 for lower-level variable updates is left unspecified. Furthermore, the selection of the parameter $I$ appears improper, particularly in comparison to the selection of the parameter $K$. This peculiarity arises because, according to Theorem 1, $I = O(1/\epsilon^2)$, while $K = O(1/\epsilon)$. Consequently, for sufficiently small values of $\epsilon$, $I$ is significantly larger than $K$. However, both the implementation of the proposed algorithm and the theoretical analysis, as exemplified by Lemma 2, necessitate $I$ to be smaller than $K$.

2. Despite the authors' assertion that they do not require the boundedness of the norm of the gradient of the upper-level objective function, $ || \nabla_y f(x, y) ||$, as stipulated in Assumption 2 (i), it is worth noting that they still demand the boundedness of $|| \nabla_y f(x, y^*(x)) ||$. The rationale behind the relative ease of achieving this condition in practical scenarios, as opposed to the former requirement, remains unclear. Moreover, it remains unaddressed whether the problems examined in the numerical experiments exhibit unbounded smoothness yet satisfy Assumption 2 (i).

3. The selection of the parameters $I$ and $N$ in the implementation of the proposed algorithm in the numerical experiments deviates from the parameters required by the theoretical analysis. Specifically, the theoretical analysis mandates that both $I$ and $N$ should be large as $\epsilon$ becomes small. However, in the context of the numerical experiments, the authors have set the values for $I$ and $N$ as merely 2 or 3, which contradicts the theoretical analysis.

**Questions:**

1. The proposed algorithm incorporates an initialization refinement procedure for the lower-level variable. I am curious about whether the authors have taken into account the computational cost associated with this initialization refinement procedure when presenting the numerical results in Figure 1.

2. While the proposed algorithm exhibits advantages over its competitors in terms of the number of epochs required, how about its performance concerning computation time?

---

> ### Author Response · Authors · 2023-11-16
> **Thank you for taking the time to review our paper and provide insightful comments. We have replied to each of your thoughts below.**
>
> **Weakness 1. The theoretical complexity result of the proposed algorithm, Theorem 1, lacks persuasiveness. Specifically, the choice of the iteration number $N$ in Algorithm 2 for lower-level variable updates is left unspecified. Furthermore, the selection of the parameter $I$ appears improper, particularly in comparison to the selection of the parameter $K$. This peculiarity arises because, according to Theorem 1, $I=\mathcal{O}(1/\epsilon^2)$, while $K=\mathcal{O}(1/\epsilon)$. Consequently, for sufficiently small values of $\epsilon$, $I$ is significantly larger than $K$. However, both the implementation of the proposed algorithm and the theoretical analysis, as exemplified by Lemma 2, necessitate $I$ to be smaller than $K$.**
>
> **A**:  Thank you for pointing it out. We actually state the choice of the iteration number $N$ in the proof of Lemma 2, but without explicitly pointing it out in statement of Lemma 2.  We have fixed it in the new version of the paper.
> For the exact value of $N$, you can check equation (45) and (50) in proof of Lemma 2 in Appendix D.3. The iteration complexity $N$ in Algorithm 2 for lower-level variable updates (within a single subroutine) should be (by eq. (45) and (50)) $N = 64^2\sigma_{g,1}^2K_0^2\left(\lambda+\sqrt{\lambda+1}\right)^2/\mu^2\epsilon^2$, where $\lambda$ is some logarithmic factors. Thus we conclude $N=\widetilde{\mathcal{O}}(\sigma_{g,1}^2K_0^2/\mu^2\epsilon^2)$ in Algorithm 2.
>
>  For Theorem 1, $I$ is indeed much smaller than $K$ for small enough $\epsilon$. Note that we choose update period $I$ to be $I = \sigma_{g,1}^2K_0^2/\mu^2\epsilon^2 = \mathcal{O}(1/\epsilon^2)$, and there is a learning rate dependency (i.e. $1/\eta$) in $K$: $K=\frac{4\Delta}{\eta\epsilon}=\mathcal{O}(1/\epsilon^4)$ because $\eta=\mathcal{O}(\epsilon^3)$. In this case, $I$ is much smaller than $K$ for small enough $\epsilon$.
>
> **Weakness 2. Despite the authors' assertion that they do not require the boundedness of the norm of the gradient of the upper-level objective function,** $\\|\nabla_yf(x,y)\\|$**, as stipulated in Assumption 2 (i), it is worth noting that they still demand the boundedness of **$\\|\nabla_y f(x, y^\*(x))\\|$**. The rationale behind the relative ease of achieving this condition in practical scenarios, as opposed to the former requirement, remains unclear. Moreover, it remains unaddressed whether the problems examined in the numerical experiments exhibit unbounded smoothness yet satisfy Assumption 2 (i).**
>
> **A**: Thank you for this insightful question. First of all, to the best of our knowledge, this is an assumption (i.e., $\\|\nabla_yf(x,y^*(x))\\|\leq M$) which is required by all bilevel optimization papers [1, 2, 3]. Also, the boundedness of $\\|\nabla_y f(x, y^*(x))\\|$ does not contradict with relaxed smoothness assumption, i.e., Assumption 1. We will illustrate it in the following. As can be seen from Figure 4 in Appendix G, we show that the smoothness parameter scales linearly in terms of the gradient norm of $x$ (parameter of the recurrent layers) and $y$ (parameter of the last linear layer).
>
> In addition, we show that if we denote $x$ as previous layers of recurrent neural networks and $y$ as the last layer (i.e., classification layer), the condition
> $\\|\nabla_yf(x,y^*(x))\\|\leq M$ indeed holds both theoretically and experimentally. Figure 4 (b) in Appendix G shows that the gradient norm in terms of $y$ is bounded, i.e., the maximal value of the horizontal axis can be bounded by $M=10$. Besides, when considering the setting of hyper-representation, we have included a proof in Appendix H, indicating that this condition holds true for any $(x, y)$ and thus also holds for $(x, y^*(x))$.
>
> [1] *Saeed Ghadimi and Mengdi Wang. Approximation methods for bilevel programming. arXiv preprint arXiv:1802.02246, 2018.*
>
> [2] *Kaiyi Ji, Junjie Yang, and Yingbin Liang. Bilevel optimization: Convergence analysis and enhanced design. In International conference on machine learning, pp. 4882–4892. PMLR, 2021.*
>
> [3] *Jeongyeol Kwon, Dohyun Kwon, Stephen Wright, and Robert D Nowak. A fully first-order method for stochastic bilevel optimization. In International Conference on Machine Learning, pp. 18083-18113. PMLR, 2023.*

---

> > ### Author Response · Authors · 2023-11-16
> > **Thank you for taking the time to review our paper and provide insightful comments. We have replied to each of your thoughts below. (cont'd)**
> >
> > **Weakness 3. The selection of the parameters $I$ and $N$ in the implementation of the proposed algorithm in the numerical experiments deviates from the parameters required by the theoretical analysis. Specifically, the theoretical analysis mandates that both $I$ and $N$ should be large as $\epsilon$ becomes small. However, in the context of the numerical experiments, the authors have set the values for $I$ and $N$ as merely 2 or 3, which contradicts the theoretical analysis.**
> >
> > **A**:  We respectfully disagree. Indeed, by Lemma 2 and Theorem 1, we know that $I = \sigma_{g,1}^2K_0^2/\mu^2\epsilon^2 = \mathcal{O}(1/\epsilon^2)$ and $N = 64^2\sigma_{g,1}^2K_0^2\left(\lambda+\sqrt{\lambda+1}\right)^2/\mu^2\epsilon^2 = \widetilde{\mathcal{O}}(1/\epsilon^2)$. However, we do not think the practical values of $I$ and $N$ contradict the theoretical analysis. Big $\mathcal{O}$ notation only gives a conservative upper bound which makes sure that the algorithm can converge even we are given the worst function instance from certain function class, but in practice, the function is never adversarial such that we can choose smaller $I$ and $N$ and hence the algorithm in fact enjoys low complexity. This is not uncommon in the literature of computer science. For example, the simplex method is very efficient in solving linear programming in practice, but it takes exponential time in the worst case from complexity point of view: this is not a contradiction, and it also inspires the seminal smoothed analysis framework [4].
> >
> > **Question 1. The proposed algorithm incorporates an initialization refinement procedure for the lower-level variable. I am curious about whether the authors have taken into account the computational cost associated with this initialization refinement procedure when presenting the numerical results in Figure 1.**
> >
> > **A**: For a fair comparison between different baselines, we have included loss curves in terms of running time in Figure 6 in Appendix I, where the loss curve of BO-REP takes the time cost of initialization refinement into consideration. Our proposed BO-REP achieves the best performance among all the baselines and shows the fastest convergence in various experiments, including Hyper-Representation, hyperparameter optimization and Data Hyper-Cleaning.
> >
> > **Question 2. While the proposed algorithm exhibits advantages over its competitors in terms of the number of epochs required, how about its performance concerning computation time?**
> >
> > **A**:   We compare our proposed algorithm with other single-loop and double-loop algorithms in terms of running time, the result is shown in Figure 6 in Appendix I. To accurately evaluate each algorithm, we use the machine learning framework PyTorch 1.13 to run each algorithm individually on an NVIDIA RTX A6000 graphics card, and record its training and test loss. As we can observe from the figure, our algorithm (BO-REP) is much faster than all other baselines in the experiment of Hyper-representation (Figure 5 (a)) and Data Hyper-Cleaning (Figure 5 (c)). For the hyperparameter optimization experiment (Figure 5 (b)), our algorithm is slightly slower at the very beginning, but quickly outperforms all other baselines. This means that our algorithm indeed has better runtime performance than the existing bilevel baselines.
> >
> >
> > [4] *Spielman, Daniel A., and Shang-Hua Teng. "Smoothed analysis of algorithms: Why the simplex algorithm usually takes polynomial time." Journal of the ACM (JACM) 51, no. 3 (2004): 385-463.*

---

> > > ### Comment · Reviewer_ZUWL · 2023-11-20
> > >
> > > I appreciate the authors' efforts in providing explanations; however, some of my concerns remain unaddressed.
> > >
> > > 1. It is important for the authors to acknowledge that not all bilevel optimization papers require the assumption $|| \nabla_y f(x,y^*(x)) || \le M$. For instance, the work
> > >
> > > Han Shen and Tianyi Chen, On penalty-based bilevel gradient descent method, ICML 2023.
> > >
> > > 2. The validation of Assumption 2(i) in Appendix H does not substantiate one of the motivation underlying this work, which asserts that Assumption 2(i) is more practical to satisfy than the requirement that the norm of the gradient of the upper-level objective function, $|| \nabla_y f(x,y) ||$, is bounded. Appendix H demonstrates that the latter condition holds for the specific hyper-representation problem.
> > >
> > > 3. The explanations regarding the concern about the selection of parameters $I$ and $N$ in the implementation of the proposed algorithm in numerical experiments are not convincing. The authors assert in their paper that the periodic updates technique is a crucial component of their proposed algorithm, with $I$ and $N$ serving as hyperparameters for this periodic updates procedure. The theoretical analysis in the paper specifies that the values of $I$ and $N$ should increase as the accuracy $\epsilon$ decreases, which can be understood as the values of $I$ and $N$ needing to grow as the iteration number $K$ increases.
> > > However, in the numerical implementation of the proposed algorithm, the authors set the values of $I$ and $N$ to a small number 2 or 3 throughout the entire iteration process. This discrepancy indicates that the algorithm implemented in the numerical experiments diverges from the one theoretically analyzed. Consequently, the algorithm employed in the numerical experiments lacks any theoretical justification. To enhance the credibility of their work, the authors should provide a theoretical analysis of the proposed algorithm with $I$ and $N$ set as fixed small numbers.

---

> ### Author Response · Authors · 2023-11-21
> **Thank you for your feedback. Below we have addressed each of your concerns**
>
> **Question 1. It is important for the authors to acknowledge that not all bilevel optimization papers require the assumption** $\\|\nabla\_yf(x,y^*(x)\\|\leq M$. **For instance, the work [Shen & Chen (2023)](https://arxiv.org/pdf/2302.05185.pdf).**
>
> **A:** The work [Shen & Chen (2023)](https://arxiv.org/pdf/2302.05185.pdf) indeed makes this assumption, but with a different notation. It's worth noting that for Assumption 1 in [Shen & Chen (2023)](https://arxiv.org/pdf/2302.05185.pdf), the author assumes that *'' There exists $L>0$ that given any $x\in\mathcal{C}$, $f(x,\cdot)$ is $L$-Lipschitz continuous on $\mathcal{U}(x)$"*, which actually implies $\\|\nabla\_yf(x,y)\\|\leq L$ for any given $x\in\mathcal{C}$ under their notation. The difference is that [Shen & Chen (2023)](https://arxiv.org/pdf/2302.05185.pdf) does not assume strong convexity of the lower-level problem, so the optimal solution set $\mathcal{S}(x)$ for the lower-level problem may not be unique. But since for any given $x\in\mathcal{C}$,  $\\|\nabla\_yf(x,y)\\|\leq L$ holds for all $y\in\mathcal{U}(x)$, then this condition also holds for any $y\in\mathcal{S}(x)$.
>
> In fact, this condition is quite important for the analysis in [Shen & Chen (2023)](https://arxiv.org/pdf/2302.05185.pdf). For example,
> - Formula (A.2) for proof of Theorem 1 in Appendix A.1, on page 21
> - Formula (A.8) and (A.10) for proof of Theorem 6 in Appendix A.2, on pages 23 and 24
> - $L$-dependency for parameter choice $\epsilon=L^2\mu/\gamma^2$, $\gamma\geq L\sqrt{3\mu\delta^{-1}}$ in part (a),  and $\epsilon=L^2/(4\gamma^2\sigma^2)$, $\gamma\geq\max\\{L\sqrt{2\mu\delta^{-1}}, L\sqrt{\delta^{-1}}/\sigma\\}$ in part (b) for proof of Proposition 2 in Appendix B.2, on page 25
> - Formula (B.5) for proof of Theorem 7 in Appendix B.3, on page 26
> - Formula (B.33) for proof of Theorem 8 in Appendix B.4, on page 32
> - Formula (C.2) for proof of Proposition 3 in Appendix C.1, on page 33
> - $L$-dependency for parameter choice $\gamma^*=L\rho$ in formula (D.3) for proof of Theorem 9 in Appendix D.1, on page 37
> - Formula (D.6) for proof of Theorem 10 in Appendix D.1, on page 38
>
> **Question 2. The validation of Assumption 2(i) in Appendix H does not substantiate one of the motivation underlying this work, which asserts that Assumption 2(i) is more practical to satisfy than the requirement that the norm of the gradient of the upper-level objective function**, $\\|\nabla\_yf(x,y)\\|$, **is bounded. Appendix H demonstrates that the latter condition holds for the specific hyper-representation problem.**
>
> **A:** Please note that the assumption "$\\|\nabla\_yf(x,y^*(x))\\|\leq M$ *for any $x\in\mathbb{R}^{d\_x}$"* is weaker than the assumption *"$\\|\nabla\_yf(x,y)\\|\leq M$ for any $x\in\mathbb{R}^{d\_x}$ and $y\in\mathbb{R}^{d\_y}$"*, that is, the latter assumption implies the former assumption. In theory, we only need the former (i.e., weaker) assumption to obtain the convergence result, and in Appendix H we demonstrate that the latter (i.e., stronger) condition holds for the hyper-representation problem, thus the former condition in the theoretical part also holds naturally.
>
> In addition, we do not assert Assumption 2(i) (i.e., "$\\|\nabla\_yf(x,y^*(x))\\|\leq M$ *for any $x\in\mathbb{R}^{d\_x}$"*) is one of the motivation in our paper. Our motivation focuses on addressing unbounded smoothness phenomenon for specific problem settings, and then we introduce the new $(L\_{x,0}, L\_{x,1}, L\_{y,0}, L\_{y,1})$-smoothness assumption under bilevel setting, but existing bilevel optimization algorithms do not have any convergence guarantees in the presence of unbounded smoothness (please read our second paragraph in Section 1 for details). We also do not claim that the assumption "$\\|\nabla\_yf(x,y^*(x))\\|\leq M$ *for any $x\in\mathbb{R}^{d\_x}$"* is more practical to satisfy than the assumption "$\\|\nabla\_yf(x,y)\\|\leq M$ *for any $x\in\mathbb{R}^{d\_x}$ and $y\in\mathbb{R}^{d\_y}$"*. We would like to emphasize that we only need a weaker assumption for theoretical analysis although we can show that stronger assumption indeed holds in practice.

---

> > ### Author Response · Authors · 2023-11-21
> > **Thank you for your feedback. Below we have addressed each of your concerns (cont'd).**
> >
> > **Question 3. The explanations regarding the concern about the selection of parameters $I$ and $N$ in the implementation of the proposed algorithm in numerical experiments are not convincing. The authors assert in their paper that the periodic updates technique is a crucial component of their proposed algorithm, with $I$ and $N$ serving as hyperparameters for this periodic updates procedure. The theoretical analysis in the paper specifies that the values of $I$ and $N$ should increase as the accuracy $\epsilon$ decreases, which can be understood as the values of $I$ and $N$ needing to grow as the iteration number $K$ increases. However, in the numerical implementation of the proposed algorithm, the authors set the values of $I$ and $N$ to a small number 2 or 3 throughout the entire iteration process. This discrepancy indicates that the algorithm implemented in the numerical experiments diverges from the one theoretically analyzed. Consequently, the algorithm employed in the numerical experiments lacks any theoretical justification. To enhance the credibility of their work, the authors should provide a theoretical analysis of the proposed algorithm with $I$ and $N$ set as fixed small numbers.**
> >
> > **A:** We want to emphasize that our empirical study is not a contradiction to our theoretical analysis. To further explore the setting of hyperparameters $N$ and $I$, we evaluate the algorithm performance on the larger value of $N$ and $I$ (i.e., $N=16, I=12$), which may better fit our theory. The results are presented in Figure 8 in Appendix K, where Figure 8 (a), (b), (c) show that the compared test accuracy on three different tasks, respectively. We can observe that the algorithm performance with the large value of $N$ and $I$ (green dash line) is almost the same as (or even better than) the original setting (green solid line). The larger number of $N$ can compensate for performance degradation induced by a long update period $I$. In particular, the new results in Figure 8 (b) (c) (green dash lines) are obtained with a slightly smaller lower-level learning rate ($8\times 10^{-4}$ in (b) and $5\times 10^{-3}$ in (c)) than the original setting (green solid lines with $1\times10^{-3}$ in (b) and $5\times 10^{-2}$ in (c)). However, the setting of $N=3, I=2$ is good enough in our experiments to achieve good performance for all the tasks.

---

> > > ### Comment · Reviewer_ZUWL · 2023-11-22
> > >
> > > Thank the authors for the explanations. I have increased my score to 6.

---

> > > > ### Author Response · Authors · 2023-11-22
> > > > **Thank you for your review.**
> > > >
> > > > Dear Reviewer ZUWL,
> > > >
> > > > Thank you for providing constructive feedback for our paper. We are glad that our responses addressed your concerns.
> > > >
> > > > Best,
> > > > Authors

---

### Official Review · Reviewer_enk4 · 2023-10-30

**Soundness:** 2 fair
**Presentation:** 2 fair
**Contribution:** 2 fair
**Rating:** 6
**Confidence:** 4

**Summary:**

This paper proposes bilevel algorithm, BO-REQ to address unbounded smoothness challenge. Specifically, the authors introduce the normalized momentum and initialization refinement techniques, and proposed algorithm achieves the convergence rate of $\mathcal{\widetilde O}(1/\epsilon^4)$. The experiments across various settings also demonstrate the effectiveness of the proposed algorithm.

**Strengths:**

The proposed algorithm outperforms other bilevel baselines in experiments across various contexts.

**Weaknesses:**

1. This paper relaxes the bounded assumptions but is not based unbounded assumption as stated in the abstract. The example pf RNN and LSTMs are misleading unless the authors are able to prove that these networks satisfy the Assumption 1.

2. The second stated contribution in introduction section is not valid. As the author also mentions in the footnote 3, the SOTA complexity results under bounded smoothness is $\mathcal{\widetilde O}(1/\epsilon^3)$. The proposed method does not achieve the SOTA complexity rate, even after considering the logarithmic factors.

3. Momentum technique has been incorporated in the bilevel optimization area before. For example, MRBO [1] achieves the $\mathcal{\widetilde O}(1/\epsilon^3)$ after momentum introduction. The authors are expected to improve the convergence rate of proposed algorithm.

4. The authors are suggested to show the experimental results w.r.t. time rather than only epochs. The proposed algorithm involves three loops per epoch, which might introduce extra computational cost. It will be more fair to compare proposed algorithm with other single-loop or double-loop based algorithms in terms of running time.

[1] Provably Faster Algorithms for Bilevel Optimization

**Questions:**

Check the weakness.

---

> ### Author Response · Authors · 2023-11-16
> **Thank you for taking the time to review our paper. Below we have addressed each of your concerns.**
>
> **Weakness 1. This paper relaxes the bounded assumptions but is not based unbounded assumption as stated in the abstract. The example pf RNN and LSTMs are misleading unless the authors are able to prove that these networks satisfy the Assumption 1.**
>
> **A**: Thank you for your suggestion. First, we want to emphasize that the unbounded smoothness assumptions were empirically observed in many prior works ([1, 2]) for RNNs, LSTMs and Transformers. Second, we have empirically verified that the Recurrent Neural Network model satisfies Assumption 1 in the revised version. In Figure 4 in Appendix G, we plot the estimated smoothness at different iterations during training neural networks. In particular, we conduct the hyper-representation experiments to verify Assumption 1. We adopt the same recurrent neural network as that in Section 4.1. The lower-level variable is the parameter of the last linear layer, and the upper-level variable is the parameter of the previous layers (2 hidden layers). In each training iteration, we calculate the gradient norm w.r.t. the upper variable $\\|\nabla_x f(x,y)\\|$ and the lower variable $\\|\nabla_y f(x,y)\\|$, and use the same method as described in Appendix H.3 in [1] to estimate the smoothness of $x$ and $y$. From Figure 4 in Appendix G, we can find that the smoothness parameter scales linearly in terms of gradient norm for both layers. This verifies the Assumption 1 empirically. Note that these results are consistent with the results in the literature, e.g., Figure 1 in [1] and Figure 1 in [2].
>
> **Weakness 2. The second stated contribution in introduction section is not valid. As the author also mentions in the footnote 3, the SOTA complexity results under bounded smoothness is
> $\widetilde{\mathcal{O}}(1/\epsilon^3)$. The proposed method does not achieve the SOTA complexity rate, even after considering the logarithmic factors.**
>
> **A**: As stated in footnote 3, there are a few works which achieve $\mathcal{O}(\epsilon^{-3})$ oracle complexity when the stochastic function is mean-squared smooth [3, 4, 5], but our paper does not have such an assumption. Without mean-squared smoothness assumption, our proposed method is optimal up to logarithmic factors as proved in [6]. We have made it more clear in the abstract as well.
>
> **Weakness 3. Momentum technique has been incorporated in the bilevel optimization area before. For example, MRBO [3] achieves the $\widetilde{\mathcal{O}}(1/\epsilon^3)$ after momentum introduction. The authors are expected to improve the convergence rate of the proposed algorithm.**
>
> **A**: Please note that these two momentum techniques are completely different. Our momentum is simply doing moving-average estimate for history stochastic gradients, but the momentum used in [3] is a type of recursive momentum which is based on the STORM estimator [7]. The STORM estimator requires the mean-squared smoothness assumption while our momentum does not. Without this assumption, our complexity result is already optimal up to logarithmic factors and it is impossible to achieve the $\widetilde{\mathcal{O}}(1/\epsilon^3)$ rate as proved in [6]. We could also improve our complexity result to $\widetilde{\mathcal{O}}(1/\epsilon^3)$ by imposing the mean-squared smoothness assumption and adopting the STORM technique, but it is out of the scope of our current paper.
>
> [1] *Jingzhao Zhang, Tianxing He, Suvrit Sra, and Ali Jadbabaie. Why gradient clipping accelerates training: A theoretical justification for adaptivity. International Conference on Learning Representations, 2020b.*
>
> [2] *Michael Crawshaw, Mingrui Liu, Francesco Orabona, Wei Zhang, and Zhenxun Zhuang. Robustness to unbounded smoothness of generalized signsgd. Advances in neural information processing systems, 2022.*
>
> [3] *Junjie Yang, Kaiyi Ji, and Yingbin Liang. Provably faster algorithms for bilevel optimization. Advances in Neural Information Processing Systems, 34:13670-13682, 2021.*
>
> [4] *Zhishuai Guo, Quanqi Hu, Lijun Zhang, and Tianbao Yang. Randomized stochastic variance-reduced methods for multi-task stochastic bilevel optimization. arXiv preprint arXiv:2105.02266, 2021.*
>
> [5] *Prashant Khanduri, Siliang Zeng, Mingyi Hong, Hoi-To Wai, Zhaoran Wang, and Zhuoran Yang. A near-optimal algorithm for stochastic bilevel optimization via double-momentum. Advances in neural information processing systems, 34:30271–30283, 2021.*
>
> [6] *Yossi Arjevani, Yair Carmon, John C Duchi, Dylan J Foster, Nathan Srebro, and Blake Woodworth. Lower bounds for non-convex stochastic optimization. Mathematical Programming, 199(1-2): 165–214, 2023.*
>
> [7] *Ashok Cutkosky and Francesco Orabona. Momentum-based variance reduction in non-convex sgd. Advances in neural information processing systems, 32, 2019.*

---

> > ### Author Response · Authors · 2023-11-16
> > **Thank you for taking the time to review our paper. Below we have addressed each of your concerns (cont'd).**
> >
> > **Weakness 4. The authors are suggested to show the experimental results w.r.t. time rather than only epochs. The proposed algorithm involves three loops per epoch, which might introduce extra computational cost. It will be more fair to compare the proposed algorithm with other single-loop or double-loop based algorithms in terms of running time.**
> >
> > **A**: We have to clarify a typo in our manuscript. Please check the algorithm framework in the new version of our paper, we fix the typo in the pseudo-code for Algorithm 2 (remove the unnecessary loop in line 1). Therefore our algorithm is a double-loop algorithm instead of a triple-loop one. Notably, we only need to execute the second loop occasionally since updating for the lower-level variable is periodic rather than in each iteration.
> >
> > We compare our proposed algorithm with other single-loop and double-loop algorithms in terms of running time, the result is shown in Figure 6 in Appendix I. To accurately evaluate each algorithm, we use the machine learning framework PyTorch 1.13 to run each algorithm individually on an NVIDIA RTX A6000 graphics card, and record its training and test loss. As we can observe from the figure, our algorithm (BO-REP) is much faster than all other baselines in the experiment of Hyper-representation (Figure 5 (a)) and Data Hyper-Cleaning (Figure 5 (c)). For the hyperparameter optimization experiment (Figure 5 (b)), our algorithm is slightly slower at the very beginning, but quickly outperforms all other baselines. This means that our algorithm indeed has better runtime performance than the existing bilevel baselines.

---

> > > ### Comment · Reviewer_enk4 · 2023-11-20
> > >
> > > Thanks the author's efforts. Still some concerns remain.
> > >
> > > 1. What is the definition of mean-squared smooth you are referring?  Is it related to outer function or inner function? Is it with regard to $x$ or $y$?
> > >
> > > 2. Is the achievement of $\mathcal{\widetilde O}(1/\epsilon^3)$ convergence rate is always based on such mean-squared smooth or have to be completed with STORM based technique? If not, the contribution is still not valid.
> > >
> > > I appreciate author's further explaining about unboundedness and extra experiments.

---

> ### Author Response · Authors · 2023-11-21
> **Thank you for your feedback. Below we have addressed each of your concerns.**
>
> **Question 1. What is the definition of mean-squared smooth you are referring? Is it related to outer function or inner function? Is it with regard to $x$ or $y$?**
>
> **A:** For single level problems, given differentiable objective function $F: \mathbb{R}^d \rightarrow \mathbb{R}$, we say the function $F$ satisfies *mean-squared smoothness* property (formula (4) on page 4 in [Arjevani et al. (2023)](https://arxiv.org/pdf/1912.02365.pdf)) if for any $x,y\in\mathbb{R}^d$ and any $\xi\sim P_{\xi}$,
> $$
> \mathbb{E}_\xi[\\|\nabla F(x;\xi)-\nabla F(y;\xi)\\|^2] \leq L^2\\|x-y\\|^2,  \tag{1}
> $$
> where we use $\xi$, $\nabla F(x;\xi)$ and $L$ to denote the random data sample, the stochastic gradient estimator and the mean-squared smoothness constant (the corresponding notations in [Arjevani et al. (2023)](https://arxiv.org/pdf/1912.02365.pdf) are $z$, $g(x,z)$ and $\bar{L}$, please check formula (4) on page 4 in [Arjevani et al. (2023)](https://arxiv.org/pdf/1912.02365.pdf) for details).
>
> A slightly stronger condition than mean-squared smoothness is the *individual smoothness* property (please check the statement *"We assume that $f(x,\xi_t)$ is differentiable, and $L$-smooth as a function of $x$ with probability 1."* in Section 3 on page 3 in [Cutkosky & Orabona (2019)](https://arxiv.org/pdf/1905.10018.pdf) for details). We say function $F$ has individual smoothness property if for any $x,y\in\mathbb{R}^d$ and any $\xi\sim P_{\xi}$,
> $$
> \\|\nabla F(x;\xi)-\nabla F(y;\xi)\\| \leq L\\|x-y\\|. \tag{2}
> $$
>
> For bilevel problems, in order to obtain $\widetilde{\mathcal{O}}(1/\epsilon^3)$ oracle complexity bound, [Yang et al. (2021)](https://arxiv.org/pdf/2106.04692.pdf), [Guo et al. (2021)](https://arxiv.org/pdf/2105.02266.pdf) and [Khanduri et al. (2021)](https://arxiv.org/pdf/2102.07367.pdf) actually require individual smoothness assumption jointly in $(x,y)\in\mathbb{R}^{d_x}\times\mathbb{R}^{d_y}$ for both upper-level (i.e., outer function $f(x,y)$) and lower-level problems (i.e., inner function $g(x,y)$). To be more specific, for upper level problem they require for any $u=(x,y), u'=(x',y')\in\mathbb{R}^{d_x}\times\mathbb{R}^{d_y}$ and any $\xi$,
> $$
> \\|\nabla_x f(u;\xi)-\nabla_x f(u;\xi)\\| \leq L_{f_{x}}\\|u-u'\\|, \quad \\|\nabla_y f(u;\xi)-\nabla_y f(u;\xi)\\| \leq L_{f_{y}}\\|u-u'\\|, \tag{3}
> $$
> and for lower level problem they require for any $u=(x,y), u'=(x',y')\in\mathbb{R}^{d_x}\times\mathbb{R}^{d_y}$ and any $\zeta$,
> $$
> \\|\nabla_y g(u;\zeta)-\nabla_y g(u;\zeta)\\| \leq L_{g_{y}}\\|u-u'\\|. \tag{4}
> $$
> These three inequalities above, namely formulas (3) and (4), can be viewed as definition of individual smoothness property under bilevel optimization setting.
>
> For more details, please check Assumption 2 in Section 3.1 on page 6 in [Yang et al. (2021)](https://arxiv.org/pdf/2106.04692.pdf), Assumption 2 in Section 2 on page 5 and Assumption 4 in Section 4 on page 8 in [Guo et al. (2021)](https://arxiv.org/pdf/2105.02266.pdf), Assumption 2 in Section 2 on page 4 in [Khanduri et al. (2021)](https://arxiv.org/pdf/2102.07367.pdf). *Please note that our paper does not assume any of these assumptions.*

---

> ### Author Response · Authors · 2023-11-21
> **Thank you for your feedback. Below we have addressed each of your concerns (cont'd).**
>
> **Question 2. Is the achievement of $\widetilde{\mathcal{O}}(1/\epsilon^3)$ convergence rate is always based on such mean-squared smooth or have to be completed with STORM based technique? If not, the contribution is still not valid.**
>
> **A:** Yes. All the bilevel optimization literature ([Yang et al., 2021](https://arxiv.org/pdf/2106.04692.pdf); [Guo et al., 2021](https://arxiv.org/pdf/2105.02266.pdf); [Khanduri et al., 2021](https://arxiv.org/pdf/2102.07367.pdf)) with $\widetilde{\mathcal{O}}(1/\epsilon^3)$ complexity use individual smoothness assumption, which is even stronger than the mean-squared smoothness assumption as we mentioned in answer to Question 1. Also, their algorithm design depends on the STORM technique ([Cutkosky & Orabona, 2019](https://arxiv.org/pdf/1905.10018.pdf)). In addition, it is shown in [Arjevani et al. (2023)](https://arxiv.org/pdf/1912.02365.pdf) that without this assumption, $\Omega(1/\epsilon^4)$ complexity is necessary for stochastic first-order optimization algorithms for single level problems. This indicates that our complexity is optimal up to logarithmic factors for the bilevel problems we considered in this paper.
>
> Let us give more details to explain this. First we consider potentially non-convex single level optimization problems as considered in [Arjevani et al. (2023)](https://arxiv.org/pdf/1912.02365.pdf) . Given differentiable objective function $F: \mathbb{R}^d \rightarrow \mathbb{R}$, our goal is to find an $\epsilon$-stationary point using stochastic first-order algorithms. Assume the algorithms access the function $F$ through a stochastic first-order oracle consisting of a noisy gradient estimator $\nabla F: \mathbb{R}^d\times\Xi \rightarrow \mathbb{R}^d$ and distribution $P_\xi$ on $\Xi$ satisfying
> $$
> \mathbb{E}\_{\xi\sim P\_{\xi}}[\nabla F(x;\xi)] = \nabla F(x), \quad \mathbb{E}\_{\xi\sim P\_{\xi}}[\\|\nabla F(x;\xi)-\nabla F(x)\\|^2] \leq \sigma^2 \tag{5}
> $$
> For potentially non-convex single level optimization problems, Theorem 3 in [Arjevani et al. (2023)](https://arxiv.org/pdf/1912.02365.pdf) states that
> - Under Assumption (5), any stochastic first-order methods requires $\Omega(1/\epsilon^4)$ queries to find an $\epsilon$-stationary point in the worst case (please check Contribution 1 in Section 1.1 on page 2, and formula (23) in Theorem 3 on page 14 in [Arjevani et al. (2023)](https://arxiv.org/pdf/1912.02365.pdf) for details);
> - Under Assumptions (5), plus mean-squared smoothness assumption, any stochastic first-order methods require $\Omega(1/\epsilon^3)$ queries to find an $\epsilon$-stationary point in the worst case (please check Contribution 2 in Section 1.1 on page 2, and formula (24) in Theorem 3 on page 14 in [Arjevani et al. (2023)](https://arxiv.org/pdf/1912.02365.pdf) for details).
>
> Note that our problem class is more expressive than the function class considered in [Arjevani et al. (2023)](https://arxiv.org/pdf/1912.02365.pdf) and hence our problem is harder. For example, if we consider an easy function where the upper-level function does not depend on the lower-level problem and does not have mean-squared smoothness, the $\Omega(1/\epsilon^4)$ lower bound can be applied in our setting. Therefore the oracle complexity $\widetilde{\mathcal{O}}(1/\epsilon^4)$ achieved in our paper is already optimal up to logarithmic factors.

---

> ### Comment · Reviewer_enk4 · 2023-11-21
>
> Thanks for your further responses. It addressed my concern and I increase my score to 6.
>
> Note that all we discussed are not included in the paper. I strongly encourage you to write an individual paragraph to explain it.
>
> Besides, I read your mentioned works and it looks like variance reduction technique can also help to improve the convergence rate. I suggest you to discuss both STORM and variance reduction techniques in your revision.

---

> > ### Author Response · Authors · 2023-11-21
> > **Thank you for your review.**
> >
> > Dear Reviewer enk4,
> >
> > Thank you very much for your suggestions. We have updated the paper and added an appendix L to justify the optimality of our algorithm.
> >
> > Best,
> > Authors

---

### Official Review · Reviewer_fyzv · 2023-10-31

**Soundness:** 3 good
**Presentation:** 2 fair
**Contribution:** 3 good
**Rating:** 8
**Confidence:** 3

**Summary:**

This paper investigates bi-level stochastic optimization problems in which the upper-level function could have an unbounded smoothness parameter. To solve the above problem, the authors proposed to use $(L_{x,0}, L_{x,1}, L_{y,0}, L_{y,1})$-smoothness assumption for the upper-level function, which is a generalization of the relaxed smoothness assumption for single-level optimization. Then they designed a new algorithm named BO-REP for solving bilevel optimization problems with unbounded smooth upper-level functions. BO-REP used normalized momentum technique for updating the upper-level variable and initialization refinement and periodic update tecniques for lower-level variables. The experiments demonstrated the effectiveness of the proposed algorithm.

**Strengths:**

### Originality
The problem setting of this paper is novel. To the best of my knowledge, this is the first work in the literature of bilevel optimization that takes account into the unbounded smoothness assumption for the upper-level functions.

The proposed method is novel. To solve the above problem, the authors proposed two novel mechanisms for updating the lower-level variables.

### Quality and Clarity

Overall, this paper is well written, although it is not clear enough in some places.

### Significance

This article explores a new problem and designs new algorithms to solve it. I think this will inspire researchers to thin about bilevel optimization problems from a new perspective and design more efficient algorithms.

**Weaknesses:**

The main drawback of the proposed algorithm is that the structure is complex, and each subroutine contains additional hyperparameters that need to be tuned.

**Questions:**

1. In initilalization refinement step, what's the purpose to invoke a multi-epoch algorithm to obtain an accurate estimate of lower-level variables and why periodic update doesn't use such a multi-epoch strategy?

2. The initialization refinement and periodic updates for are carefully designed for obtaining an accurate estimate of the lower-level variables at each iteration. Under current unbounded smoothness assumptions, is that possible to design algorithms with inexact estimate of the lower-level variables? Since the periodic update for the lower-level variables make the proposed algorithm a nested loop one, is there any potential strategies for us to design a single-loop algorithm for solving the unbounded smoothness bilevel optimization problems?

3. In the experiments, how is the ball radius R chosen?

Minor issues:
1) caption of table: "$\mathcal{C}_L^{a, a}$ denotes a-times differentiability with Lipschitz k-th order derivatives." I think it should be $\mathcal{C}_L^{a, k}$ here.
2) In the pseudo-code for Algorithm 2, the outer loop with the iteration variable k spanning from 0 to K-1 is repeatedly defined, which may lead to ambiguity or confusion for the readers. To enhance clarity, I'd recommend passing the current iteration k as an input to the UpdateLower subroutine (which is a part of the BO-REP algorithm) and subsequently removing the loop from the algorithm.

---

> ### Author Response · Authors · 2023-11-16
> **Thank you for taking the time to review our paper and provide insightful comments. We have replied to each of your thoughts below.**
>
> **Weakness. The main drawback of the proposed algorithm is that the structure is complex, and each subroutine contains additional hyperparameters that need to be tuned.**
>
> **A**: Our problem setting is much more challenging to solve than the standard bilevel optimization for smooth objectives. The reason is that, under Assumption 1, i.e., $(L_{x,0},L_{x,1}, L_{y,0}, L_{y,1})$-smoothness assumption, we have to carefully update both upper-level and lower-level variables. Therefore our proposed algorithm has more complex structure compared to other bilevel algorithms. However, we want to emphasize that our algorithm is not sensitive to additional hyperparameters such as $I$ and $N$ (please see our ablation study in Appendix J). In fact, for each subroutine of the periodic updates, we do not need to carefully tune these values in practice:  some default values of $I$ and $N$ (e.g., $I=2, N=3$) do not seem to hurt the performance for a wide range of bilevel optimization tasks.
>
> **Question 1. In initialization refinement step, what's the purpose to invoke a multi-epoch algorithm to obtain an accurate estimate of lower-level variables and why periodic update doesn't use such a multi-epoch strategy?**
>
> **A**: For the initialization refinement step, to achieve $\\|y_0-y_0^*\\|\leq\epsilon/8K_0$ with high probability guarantee, the iteration (and also oracle) complexity for multi-epoch SGD (by detailed statement of Lemma 1 in Appendix D.2) and SGD (by Lemma 12 in Appendix D.2)  are $\widetilde{\mathcal{O}}(1/\epsilon^2)$ and $\widetilde{\mathcal{O}}(1/\epsilon^4)$, respectively. Thus multi-epoch SGD algorithm has a better convergence rate compared to SGD for initialization refinement.
>
> When $k$ is a multiple of $I$, the periodic update step performs actual updates for the lower-level variable $y$. Note that the initial distance between $y_k$ and $y_k^*$ satisfies $\\|y_k-y_k^*\\|\leq\epsilon/4K_0$ with high probability. By the detailed statements of Lemma 12 (SGD) and Lemma 1 (multi-epoch SGD)  in Appendix D.2, and Lemma 2 in Appendix D.3 (periodic updates), we can see that the high probability bound for SGD benefits from the small initial distance $\\|y_k-y_k^*\\|$ (e.g., the polynomial dependency in terms of $D$ in Lemma 12 in formula (20)), while the high probability bound for multi-epoch SGD only marginally benefits from the small initial distance (e.g., the logarithmic dependency of $\mathcal{V}\_0$ in Lemma 1 in formula (32)). Therefore, with $\\|y_k-y_k^*\\|\leq\epsilon/4K_0$, both multi-epoch SGD and SGD guarantee $\\|y_{k+1}-y_k^*\\|\leq\epsilon/8K_0$ with high probability within $\widetilde{\mathcal{O}}(1/\epsilon^2)$ iterations. So we choose SGD instead of multi-epoch SGD for the periodic update step due to its easy implementation in practice.

---

> > ### Author Response · Authors · 2023-11-16
> > **Thank you for taking the time to review our paper and provide insightful comments. We have replied to each of your thoughts below. (cont'd)**
> >
> > **Question 2. The initialization refinement and periodic updates for are carefully designed for obtaining an accurate estimate of the lower-level variables at each iteration. Under current unbounded smoothness assumptions, is that possible to design algorithms with inexact estimate of the lower-level variables? Since the periodic update for the lower-level variables make the proposed algorithm a nested loop one, is there any potential strategies for us to design a single-loop algorithm for solving the unbounded smoothness bilevel optimization problems?**
> >
> > **A**: Thank you for your insightful question. We will explain why we cannot use single-loop based on our current proof strategy. The main difficulty of the paper comes from the bias term $\\|\mathbb{E}\_k[\widehat{\nabla}\Phi(x_k)]-\nabla\Phi(x_k)\\|$ of the hypergradient estimator, whose upper bound depends on both $L_{x,1}\\|y_k-y_k^*\\|\\|\nabla\Phi(x_k)\\|$ and $\\|y_k-y_k^*\\|$. Please check the proof of Lemma 4 in Appendix D.6 (in the middle of page 33, between inequality (ii) and (iii)) for details. Due to the fact that both $\\|y_k-y_k^*\\|$ and $\\|\nabla \Phi(x_k)\\|$ are measurable with respect to $\mathcal{F}_{k-1}$, we have to use high probability analysis for $y_k$ to disentangle these two terms.
> >
> > If we use single-loop to solve our algorithm, then for every iteration we update $y_k$ just via one step of stochastic gradient descent. By the high probability analysis of SGD for the lower-level problem (i.e., Lemma 12 in Appendix D.2), if we choose $
> > \lambda = \max\left(\sqrt{3\ln\left(\frac{2K}{\delta}\right)}, \ln\left(\frac{2K}{\delta}\right)\right)$, $\alpha_s=\gamma$ and $T_s=1$ (i.e., single loop) in formula (19) and (20) in Lemma 12, then with probability at least $1-\delta/K$, we have
> >
> > $$
> >  \frac{\mu}{2}\\|y_1-y_0^*\\|^2 \leq g(x_0,y_1)-g(x_0,y_0^*)  \leq \frac{1}{\gamma}D^2+2\sigma_{g,1}^2\gamma+\lambda\left(2\sqrt{2}D\sigma_{g,1}+2\sigma_{g,1}^2\gamma\right) \\
> > $$
> > $$
> > =\frac{1}{\gamma}D^2+2\sigma_{g,1}^2(\lambda+1)\gamma+2\sqrt{2}\lambda\sigma_{g,1}D = 2\sqrt{2(\lambda+1)}\sigma_{g,1}D+2\sqrt{2}\lambda\sigma_{g,1}D \\
> > $$
> > $$
> > = 2\sqrt{2}(\lambda+\sqrt{\lambda+1})\sigma_{g,1}D,
> > $$
> > where $D=\epsilon/8K_0$ is the upper bound for $\|y_0-y_0^*\|$, and we choose $\gamma=\frac{D}{\sigma\sqrt{2(\lambda+1)}}$ that minimizes the value of the right-hand-side of the inequality. Then we have
> > $$
> > \\|y_1-y_1^*\\| \leq \\|y_1-y_0^*\\| + \\|y_0^*-y_1^*\\| \leq \sqrt{4\sqrt{2}(\lambda+\sqrt{\lambda+1})\sigma_{g,1}/\mu}\sqrt{D} + \frac{C_{g_{xy}}}{\mu}\eta
> > $$
> > $$
> >  = \sqrt{\sqrt{2}(\lambda+\sqrt{\lambda+1})\sigma_{g,1}/2K_0\mu}\sqrt{\epsilon} + \frac{C_{g_{xy}}}{\mu}\eta = \widetilde{\mathcal{O}}(\epsilon^{1/2}),
> > $$
> > where the last equation holds because $\eta=\mathcal{O}(\epsilon^3)$ and hence the second term is dominated by the first term when $\epsilon$ is small. Then repeat the same process, with probability at least $1-2\delta/K$, we have $\\|y_2-y_2^*\\|=\widetilde{\mathcal{O}}(\epsilon^{1/4})$. Generally speaking, we have with probability at least $1-k\delta/K$, $\\|y_k-y_k^*\\|=\widetilde{\mathcal{O}}(\epsilon^{1/2^k})$. Since for $\epsilon<1$, we have $\epsilon < \epsilon^{1/2} < \epsilon^{1/2^2} \leq \cdots \leq \epsilon^{1/2^K}$, this means that with high probability, $\frac{1}{K}\sum_{k=0}^{K-1}\\|y_k-y_k^*\\| = \frac{1}{K}\sum_{k=0}^{K-1}\mathcal{O}(\epsilon^{1/2^k})$, and we cannot conclude that $\frac{1}{K}\sum_{k=0}^{K-1}\\|y_k-y_k^*\\| = \mathcal{O}(\epsilon)$ when $K$ is large (note that $\mathcal{O}(\epsilon)$ bound for this quantity is required in our proof to obtain the $\widetilde{\mathcal{O}}(\epsilon^{-4})$ complexity). Therefore the single loop algorithm does not work if we follow our current proof strategy.
> >
> > In summary, both accurate estimate of the lower-level variables at each iteration (i.e., initialization refinement) and the nested loop nature of periodic updates are important for our convergence analysis.

---

> > > ### Author Response · Authors · 2023-11-16
> > > **Thank you for taking the time to review our paper and provide insightful comments. We have replied to each of your thoughts below. (cont'd)**
> > >
> > > **Question 3. In the experiments, how is the ball radius $R$ chosen?**
> > >
> > > **A**: The ball radius $R$ is set as $0.5$ in the experiments. We experimentally explore how the ball radius $R$ affects the algorithm's performance. We set the ball radius as $\{0.001, 0.005, 0.01, 0.05, 0.10, 0.50, 1.00, 5.00\}$ respectively, and then conduct the bilevel optimization on Hyper-representation tasks. We keep all the other hyperparameters the same as in Section 4.1. The result is shown in Table 1.  When the ball radius $R$ is too small (i.e., $R<0.10$), the performance significantly drops, possibly due to the overly restricted search space for the lower-level variable.  When the ball radius $R \geq 0.50$, the performance becomes good and stable. In our experiments in the main text, we choose $R=0.50$.
> > >
> > > ### **Table 1: Test accuracy vs. Projection radius $R$.**
> > > | **Radius**   | $R=0.001$ | $R=0.005$ | $R=0.01$ | $R=0.05$ | $R=0.10$ | $R=0.50$ | $R=1.00$ | $R=5.00$ |
> > > |-------------|----------------|-----------------|---------------|--------------|---------------|---------------|---------------|--------------|
> > > | **Test ACC**| $65.06\\%$ | $67.64\\%$  | $70.76\\%$| $81.22\\%$| $86.84\\%$ | $88.84\\%$ | $88.84\\%$ | $88.84\\%$ |
> > >
> > > **Minor issues 1: Caption of table: "$\mathcal{C}\_{L}^{a,a}$ denotes a-times differentiability with Lipschitz k-th order derivatives." I think should be $\mathcal{C}\_{L}^{a,k}$ here.**
> > >
> > > **A**: We have fixed this issue in the new version.
> > >
> > > **Minor issue 2: In the pseudo-code for Algorithm 2, the outer loop with the iteration variable $k$ spanning from $0$ to $K-1$ is repeatedly defined, which may lead to ambiguity or confusion for the readers. To enhance clarity, I'd recommend passing the current iteration k as an input to the UpdateLower subroutine (which is a part of the BO-REP algorithm) and subsequently removing the loop from the algorithm.**
> > >
> > > **A**: Thank you for catching this! It is indeed a typo in our original manuscript. Please check the algorithm framework in the new version of our paper, we fix the typo in the pseudo-code for Algorithm 2 (remove the unnecessary loop in line 1).

---

> ### Comment · Reviewer_fyzv · 2023-11-18
>
> I would like to thank the authors' detailed responses and explanations which have addressed my previous concerns. So I will increase my score to 8.

---

> > ### Author Response · Authors · 2023-11-18
> > **Thank you for your review.**
> >
> > Dear Reviewer fyzv,
> >
> > Thank you very much for your constructive comments to help us improve our paper! We are glad that our responses addressed your concerns.
> >
> > Best,
> > Authors

---

### Official Review · Reviewer_7Cii · 2023-11-01

**Soundness:** 4 excellent
**Presentation:** 4 excellent
**Contribution:** 3 good
**Rating:** 8
**Confidence:** 4

**Summary:**

The authors present a novel algorithm for bilevel optimization called BO-REP. This method achieves $\mathcal{O}(1/\epsilon^4)$ convergence rates to find $\epsilon$-stationary point in the stochastic non-convex setup assuming only a relaxed smoothness assumption. Moreover, this result matches the state-of-the-art complexity results under the more restrictive types of smoothness up to logarithmic factors. The key two technical ingredients - initialization refinement and periodic updates - are easily implementable. Finally, authors demonstrate the practical efficiency of the proposed approach on hyper-representation learning, hyperparameter optimization and data hyper-clearning for text classification.

**Strengths:**

- The paper is very well-written and easy to follow. Moreover, it clearly demonstrate all the novelties of the proposed approach.
- The complexity result matches the existing state-of-the-art methods with less restrictive smoothness assumption, that is critical for presented applications on RNNs.
- The experimental validation shows the superiority of the proposed method over other methods for bilevel optimization.

**Weaknesses:**

- Lack of lower bounds does not allow to understand optimality of the proposed method under given assumption;
- Proposed guarantees holds only for finding an $\epsilon$-stationary point in expectation.

**Questions:**

- Is it possible to relax the strong convexity assumption of the lower function to log-concavity?
- What are challenges to provide high-probability guarantees for $\epsilon$-stationary point for the given algorithm?

---

> ### Author Response · Authors · 2023-11-16
> **Thank you for taking the time to review our paper. Below we have addressed each of your concerns.**
>
> **Weakness 1. Lack of lower bounds does not allow to understand optimality of the proposed method under given assumption.**
>
> **A**: Thank you for your insightful question. We want to emphasize that our oracle complexity of finding an $\epsilon$-stationary point is indeed optimal up to logarithmic factors by directly applying the lower bound results in [1]. Let us give more details here. For the upper-level problem, we assume the stochastic gradient estimator is unbiased and has bounded variance. Under this stochastic gradient oracle, the lower bound for finding $\epsilon$-stationary points in non-convex smooth single-level stochastic optimization with first-order oracle is at least $\Omega(1/\epsilon^4)$ (Theorem 1 of [1]). Since the single-level optimization is a special case of bilevel optimization (e.g., bilevel optimization becomes a single-level optimization when the upper-level function does not depend on the optimal solution of the lower-level problem) and standard smoothness is a special case of relaxed smoothness (e.g., relaxed smoothness with $L_1=0$ correspond to standard smoothness), our $\widetilde{\mathcal{O}}(1/\epsilon^4)$ complexity result is optimal up to logarithmic factors. We have added descriptions in the main paper to make it more clear.
>
>
> **Weakness 2. Proposed guarantees holds only for finding an $\epsilon$-stationary point in expectation.**
>
> **A**: Please note that we do not assume light tail assumption for upper-level problems in our paper, so our convergence guarantees hold only in expectation. Note that the high probability result even in the single-level stochastic nonconvex optimization setting requires the light tail assumption (e.g., Corollary 2.5 in [2]).
>
> **Question 1. Is it possible to relax the strong convexity assumption of the lower function to log-concavity?**
>
> **A**: Strong convexity guarantees that the optimal solution $y^*(x)$ is unique for the lower-level problem: this is very important for our analysis. However, this property does not always hold for log-concave functions. For example, let $g(x,y)=(\sin{x}+2)\exp(-y^2/2)$, then $g(x,y)$ is log-concave in $y$ for any $x$, and $g(x,y)$ also satisfies other assumptions in Assumption 2. For this specific function, $y^*(x)$ does not even exist, since for any fixed $x$, we could not find a minimum for $y$ if $y\in\mathbb{R}^{d_y}$.
>
> **Question 2. What are challenges to provide high-probability guarantees for $\epsilon$-stationary point for the given algorithm?**
>
> **A**: For Corollary 2.5 in [2], it requires stochastic estimators to be unbiased and have bounded variance, and also the light-tail assumption for stochastic gradient noise to provide high probability bound for single-level non-convex problems. The main challenge for bilevel optimization is the following: even if we have all of these assumptions in bilevel problems, the hypergradient estimator is still biased  (check Lemma 4 in our paper for details) and therefore the techniques in the single-level stochastic optimization may not apply.
>
>
> [1] *Yossi Arjevani, Yair Carmon, John C Duchi, Dylan J Foster, Nathan Srebro, and Blake Woodworth. Lower bounds for non-convex stochastic optimization. Mathematical Programming, 199(1-2): 165–214, 2023.*
>
> [2] *Saeed Ghadimi and Guanghui Lan. Stochastic first and zeroth-order methods for nonconvex stochastic programming. SIAM Journal on Optimization, 23(4):2341–2368, 2013.*

---

### Author Response · Authors · 2023-11-16
**General Response**

Thank you to all of the reviewers for taking the time to review our paper and provide many useful feedback. We have responded to each of your reviews individually, and here we provide a general summary of the main changes we made to the paper. The major changes of our paper are marked in red.

1. We have fixed a typo in Algorithm 2, which gets rid of one repeatedly defined loop.

2. We have emphasized that our algorithm achieves optimal oracle complexity when there is no mean-squared smooth condition of the stochastic gradient oracle.

3. In Appendix G, we empirically verified that the Recurrent Neural Network (RNN) model satisfies Assumption 1. In particular, we show that the smoothness parameter scales linearly in terms of the gradient norm of RNN parameters both for recurrent layers and also for the last linear layer.

4. In Appendix H, we theoretically and empirically verify that Assumption 2(i), i.e. $\\|\nabla_yf(x,y^*(x))\\|\leq M$, holds in practice when considering the setting of hyper-representation.

5. In Appendix I, we conduct experiments to compare our proposed algorithm with other single-loop
and double-loop algorithms in terms of running time. We have also added a new baseline SABA for comparison. The results show our algorithm (BO-REP) converges much faster than all other baselines in most cases and always achieves the best performance.

6. In Appendix J, we conduct careful ablation studies to explore the impact of hyperparameter $I$ (the update period of the lower-level variable), $N$ (the number of iterations for updating the lower-level variable during each period), and $R$ (the ball radius of projection). The results show that our algorithm is not sensitive to the choices of $I$ and $N$ and default values (e.g., $I=2$, $N=3$) work well for all of our experiments. We also show how the value of the project ball radius $R$ affects the performance and we find that $R=0.5$ works well for all experiments we have tried.

7. In Appendix K, we added experiments for our algorithm with large values of $I$ and $N$, which may better fit our theory. It shows that our algorithm can also outperform other bilevel methods in three different tasks.

8. In Appendix L, we illustrated why our proposed algorithm is optimal if no additional assumptions are imposed.

---

### Meta-Review · Area_Chair_PMEv · 2023-12-05

**Metareview:**

This paper proposes a new algorithm for bilevel optimization under unbounded smoothness assumption. All reviewers have a positive opinions on this work. The rebuttal period helped to clarify some remaining points.

**Justification For Why Not Higher Score:**

Lack a deeper impact on the bilevel optimization community

**Justification For Why Not Lower Score:**

only positive opinions from the referees.

---

### Decision · Program_Chairs · 2024-01-16

Accept (spotlight)